# Ensembling Pruned Attention Heads For Uncertainty-Aware Efficient Transformers

**Firas Gabetni**[1,*]**, Giuseppe Curci**[1,2,*]**, Andrea Pilzer**[3]**, Subhankar Roy**[4]**, Elisa Ricci**[2,5]**, Gianni Franchi**[1,6]

[1]U2IS, ENSTA, Institut Polytechnique de Paris    [2]University of Trento    [3]NVIDIA
[4]University of Bergamo, Italy    [5]Fondazione Bruno Kessler (FBK)    [6]AMIAD, Pôle Recherche, Palaiseau

## Abstract

Uncertainty quantification (UQ) is essential for deploying deep neural networks in safety-critical settings. Although methods like Deep Ensembles achieve strong UQ performance, their high computational and memory costs hinder scalability to large models. We introduce *Hydra Ensembles*, an efficient transformer-based ensemble that prunes attention heads to create diverse members and merges them via a new multi-head attention with grouped fully-connected layers. This yields a compact model with inference speed close to a single network, matching or surpassing Deep Ensembles in UQ performance without retraining from scratch. We also provide an in-depth analysis of pruning, showing that naive approaches can harm calibration, whereas Hydra Ensembles preserves robust uncertainty. Experiments on image and text classification tasks, with various architectures, show consistent gains over Deep Ensembles. Remarkably, in zero-shot classification on ImageNet-1k, our approach surpasses state of the art methods, even without requiring additional training.

## 1 Introduction

Deep neural networks (DNNs) excel in vision, language, and multimodal tasks, yet are prone to making overconfident errors (Hein et al., 2019). This unreliability in predictions is especially concerning in safety-critical domains such as healthcare and autonomous driving, underscoring the importance of studying *Uncertainty Quantification (UQ)* in DNNs (Gawlikowski et al., 2023).

Several UQ approaches have been explored, including evidential methods (Sensoy et al., 2018), conformal prediction (Mollaali et al., 2025), Bayesian inference (Kendall & Gal, 2017), and ensemble methods (Lakshminarayanan et al., 2017; Laurent et al., 2023). Currently, the most reliable method for UQ is *Deep Ensembles* (Lakshminarayanan et al., 2017) that aggregates predictions from multiple independently trained models. Although highly accurate, Deep Ensembles is extremely costly, as it requires multiple rounds of pre-training and fine-tuning, storing several checkpoints, and performing as many number of forward passes as the number of models, making it both slow and memory-intensive, especially for foundation models (e.g, CLIP (Radford et al., 2021), BERT (Devlin et al., 2019)). Several efficient alternatives exist, such as MC Dropout (Gal & Ghahramani, 2016), MIMO (Havasi et al., 2020), BatchEnsemble (Wen et al., 2020), and Packed Ensembles (Laurent et al., 2023), which lower computational costs by reusing weights or sharing parameters. However, they still require full pre-training, and scaling them to transformer-based (Vaswani et al., 2017; Dosovitskiy et al., 2020) large-scale foundation models is computationally expensive.

To still leverage the power of Deep Ensembles for UQ while reducing the inference cost, a naive approach is pruning or discarding unimportant weights of each constituent model. Contrary to conventional wisdom, in this work we show that commonly used pruning methods (Molchanov et al., 2019; He et al., 2020), despite their ability to preserve accuracy, can, in fact, harm calibration and lead to unreliable predictions. We prove theoretically and empirically under which conditions pruning degrades performance (predictive uncertainty), and consequently raise a pertinent question: *How can we leverage ensembles of transformers for efficient and effective UQ?*

---

[*]Equal contribution.

Motivated by this challenge, we develop a framework for ensembling pruned transformers for efficient and effective UQ, which is amenable to large-scale models like CLIP and BERT. In detail, our framework, *Hydra Ensembles*[1], aims to preserve the diversity of multiple models, like Deep Ensembles, while maintaining the inference cost close to a single model. Unique to our approach, we generate diverse models via pruning attention heads from a single pre-trained transformer model. These pruned *subnetworks* are then merged into a *single model* using Grouped Fully Connected (GFC) layers (Lafage et al., 2025a). Importantly, unlike previous approaches (Liu et al., 2021; Le et al., 2020), Hydra Ensembles does not require training each member from scratch, and can even operate without fine-tuning, offering computational advantages both at training and inference.

We evaluate our proposed Hydra Ensembles on three different classification tasks: image classification with ViT (Dosovitskiy et al., 2020), text classification with BERT (Devlin et al., 2019), and zero-shot classification with OpenCLIP-ViT (Ilharco et al., 2021). Our experimental results demonstrate that Hydra Ensembles is competitive to Deep Ensembles both in terms of accuracy and calibration metrics, while greatly reducing the inference costs. Notably, on zero-shot ImageNet-1K OOD benchmarks, Hydra Ensembles surpasses the state of the art ViLU (Lafon et al., 2025) method (which requires training) by +1.3 AUROC, –3.5 FPR95, and +4 AUPR (Table 3).

Our main **contributions** are: *(i)* We investigate the impact of pruning transformer-based models on UQ, and demonstrate both theoretically and empirically that naive pruning can lead to poorly calibrated uncertainty (Section 3.2). *(ii)* We introduce *Hydra Ensembles*, the first pruning framework specifically designed for UQ in transformer-based large-scale models (Section 4). *(iii)* We show that Hydra Ensembles delivers uncertainty estimates that are comparable to Deep Ensembles, while significantly reducing computational costs both during training and inference (Section 5).

## 2 RELATED WORK

**Transformers and Uncertainty Quantification.** Estimating epistemic uncertainty in DNN is challenging. A common approach is to approximate the intractable posterior distribution over the model's weights. Bayesian methods such as Variational Inference (Graves, 2011; Ranganath et al., 2014) or Markov Chain Monte Carlo (MCMC) (Chen et al., 2014; Neal, 2012) are theoretically sound but often too computationally expensive to scale to large transformer models. Deep Ensembles (Lakshminarayanan et al., 2017) remain the gold standard for accuracy and calibration, but their cost grows linearly with the number of models, making them impractical for very large architectures. Lighter alternatives such as MC Dropout (Gal & Ghahramani, 2016), MIMO (Havasi et al., 2020), BatchEnsemble (Wen et al., 2020), MaskEnsembles (Durasov et al., 2021), and LoRA-Ensemble (Mühlematter et al., 2024; Wang et al., 2023) reduce computation by sharing most of the backbone, but this limits the independence of ensemble members. Packed-Ensembles (Laurent et al., 2023) maintain stricter independence but reduce per-member representation capacity. For CLIP (Radford et al., 2021) and other Large vision-language models, post-hoc methods like BayesVLM (Baumann et al., 2024) (Laplace approximation on the last layers) and ViLU (Lafon et al., 2025) (adding a lightweight error-prediction head) have been explored. While convenient, post-hoc approaches generally yield weaker uncertainty estimates. Overall, ensemble-style methods provide stronger uncertainty but either require costly retraining or sacrifice diversity for efficiency. Hydra Ensembles offers a balance: it preserves member diversity, avoids retraining, and achieves near single-model inference cost.

**Network Pruning.** Neural network pruning is widely used to reduce model size and computational cost while retaining predictive accuracy. Pruning removes less important weights to preserve accuracy (Chauvin, 1988; Molchanov et al., 2019; He et al., 2020; Sun et al., 2025), but it often degrades robustness to noise and impairs uncertainty estimation (Liebenwein et al., 2021). Several prior works leverage pruning to construct ensembles, including unstructured pruning with complementary subnetworks (Whitaker & Whitley, 2022), stochastic masking (Whitaker & Whitley, 2024), and sparse training from scratch of pruned subnetworks (Liu et al., 2021; Le et al., 2020). While these methods can improve accuracy, they typically require retraining full models, rely on slow iterative pruning, or use unstructured pruning that provides little to no speedup. In contrast, *Hydra Ensembles* lever-

---

[1]In Greek mythology, the Hydra is a serpent-like creature with multiple heads, and it is famously known for its regenerative ability: when one head is severed, two more grow in its place.

ages pre-trained models and structured head-level pruning to build diverse subnetworks efficiently, avoiding retraining while improving accuracy and UQ.

# 3 ON PRUNING AND UNCERTAINTY QUANTIFICATION

In this section, we first introduce some preliminaries on UQ and pruning, and then present our initial theoretical result examining the effect of pruning on performance under noisy data.

## 3.1 PRELIMINARIES

**Uncertainty Quantification.** Consider a deep neural network (DNN) $f_{\boldsymbol{\theta}}(\cdot)$ with parameters $\boldsymbol{\theta} \in \mathbb{R}^D$, trained on data $\mathcal{D} = \{(\mathbf{x}_i, y_i)\}_{i=1}^n$. Uncertainty in predictions is usually split into two types (Hüllermeier & Waegeman, 2021): *(i) Aleatoric uncertainty*: noise or ambiguity in the data, *(i) Epistemic uncertainty*: limited knowledge of the model parameters.

Following Blundell et al. (2015), the prediction of a DNN on input $\boldsymbol{x}$ can be interpreted as a conditional likelihood: $f_{\boldsymbol{\theta}}(\boldsymbol{x}) = P(\boldsymbol{y} \mid \boldsymbol{\theta}, \boldsymbol{x})$. In practice, a simple UQ technique is the MSP, maximum softmax probability (Hendrycks & Gimpel, 2017). A more reliable method is *Deep Ensembles*, i.e. training several networks with the same architecture but different initializations:

$$P(\boldsymbol{y} \mid \boldsymbol{x}) = \frac{1}{M} \sum_{m=1}^{M} P(\boldsymbol{y} \mid \boldsymbol{\theta}^{(m)}, \boldsymbol{x}).$$

Ensembles capture both types of uncertainty but are computationally expensive, since each network must be trained fully. In this work, we explore pruning as a way to alleviate these costs while retaining the benefits of ensembles.

**Pruning.** Pruning reduces the number of weights in a DNN while keeping accuracy close to the original model. Formally, it seeks a smaller parameter set $\tilde{\boldsymbol{\theta}} \in \mathbb{R}^d$ $(d < D)$ such that the loss $\mathcal{L}(\cdot)$ of the DNN on dataset $\mathcal{D}$ is equal if we reduce the number of parameters:

$$\mathcal{L}_{\mathcal{D}}(\tilde{\boldsymbol{\theta}}) \approx \mathcal{L}_{\mathcal{D}}(\boldsymbol{\theta}).$$

Classical methods such as Optimal Brain Damage (LeCun et al., 1989) and Optimal Brain Surgeon (Hassibi & Stork, 1992) analyze the sensitivity of weights using a Taylor expansion of the loss and prune the least important ones, please refer to A.1 for more details.

## 3.2 PRUNING UNDER NOISY DATA CONDITIONS

Pruning methods are effective on clean test datasets $\mathcal{D}^t$, as demonstrated by numerous studies on both structured (Kurtić et al., 2023; Kwon et al., 2022; Park et al., 2023) and unstructured (Kurtic et al., 2022; Yao et al., 2021; Xu et al., 2022) pruning approaches. However, in the presence of a noisy or corrupted test dataset $\mathcal{D}^n$, pruning may degrade performance (Liebenwein et al., 2021).

*Assumption* 1 (Clean vs. Noisy Datasets). Let $\mathcal{D}^t$ denote a clean test dataset and $\mathcal{D}^n$ a noisy (or corrupted) test dataset. Define the loss gap between the two as

$$\Delta\mathcal{L}(\boldsymbol{\theta}) \coloneqq \mathcal{L}_{\mathcal{D}^n}(\boldsymbol{\theta}) - \mathcal{L}_{\mathcal{D}^t}(\boldsymbol{\theta}).$$

We assume that $\boldsymbol{\theta}$ minimizes $\mathcal{L}_{\mathcal{D}}$ (training set) and $\mathcal{L}_{\mathcal{D}^t}$ (clean test set), i.e.

$$\nabla\mathcal{L}_{\mathcal{D}}(\boldsymbol{\theta}) = 0, \text{ and } \nabla\mathcal{L}_{\mathcal{D}^t}(\boldsymbol{\theta}) = 0,$$

and that pruning induces a small perturbation $\delta\boldsymbol{\theta}$. Also we assume that the pruning perturbation $\delta\boldsymbol{\theta}$ is aligned (non-negatively correlated) with the gradient of the noisy loss :

$$\left(\nabla\mathcal{L}_{\mathcal{D}^n}(\boldsymbol{\theta})\right)^{\top} \delta\boldsymbol{\theta} \geq 0,$$

**Proposition 1** (Pruning is Worse under Noise). *Suppose Assumption 1 holds. Let $\mathbf{H}^t$ and $\mathbf{H}^n$ denote the Hessians of the loss on $\mathcal{D}^t$ and $\mathcal{D}^n$, respectively. If $\mathbf{H}^n - \mathbf{H}^t \succ 0$, then the loss gap after pruning satisfies $\Delta\mathcal{L}(\boldsymbol{\theta}) \leq \Delta\mathcal{L}(\boldsymbol{\theta} + \delta\boldsymbol{\theta})$, i.e., pruning degrades performance more severely on the noisy dataset than on the clean one.*

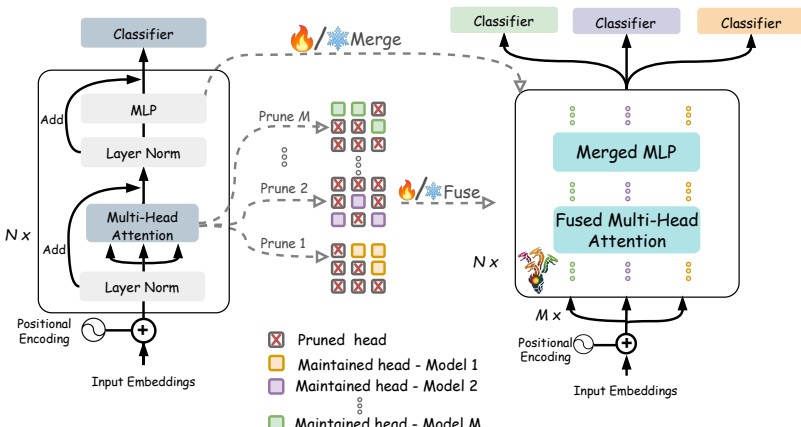

Figure 1: **Hydra Ensembles.** We start from a single transformer backbone and prune its attention heads to create multiple diverse subnetworks. These subnetworks are then combined at the head level into a Fused Multi-Head Attention (MHA), and then also merged at the MLP level, as described in Section 4.2. The pruned heads and MLPs can either be fine-tuned or kept frozen. Transformer heads are shown in matrix form for illustration only.

The proof of the proposition 1 is present in Appendix A.2. This shows that classically pruned DNNs suffer from a stronger degradation of performance than unpruned models when evaluated on noisy data. Appendix B.1 examines whether the hypothesis required for Proposition 1 holds. Hence, such pruned networks cannot be directly relied upon for uncertainty quantification. This raises a natural question: *how can we leverage ensembles of pruned models in a way that remains effective for uncertainty estimation?*

A possible improvement is to finetune the pruned models, but for some foundation models this may expensive. A promising alternative comes from *circuits* (Olah et al., 2020), developed in mechanistic interpretability. A circuit is a subgraph of the network that performs a semantically meaningful subcomputation (e.g., a feature or an attention head). Instead of removing weights blindly, one extracts subnetworks that preserve useful functionality. Recent methods, such as the *Headmap algorithm* (Wang et al., 2025), allow systematic extraction of circuits that remain stable under noise. In this paper we propose to build ensembles of such structured subnetworks, combining the benefits of pruning with robustness for UQ. Our approach is described in the following Section.

In the supplementary material, we provide additional discussions (Appendix B) that help deepen the understanding of our method. In particular, Appendix B.2 explores the circuit-level representations involved in UQ. We show that certain attention heads are highly specialized for this task, and pruning them can be particularly harmful. This specialization is also illustrated in Figure 4, making this section a unique contribution that we invite the reader to examine closely.

# 4    EFFICIENT ENSEMBLES WITH HYDRA ENSEMBLES

Hydra Ensembles (Fig. 1) is based on the idea of building an efficient *ensemble of pruned Transformers*. More specifically, we perform *structured head pruning* on multiple copies of the same backbone (each pruned differently) and then merge the remaining attention heads into a single Transformer. This design preserves ensemble-like diversity and ensures that inference remains *fast and efficient*, since it does not require to run one forward pass for each model sequentially. In the following, we present the details of our approach. Subsection 4 describes how we efficiently transform an ensemble of pruned models into a single Hydra Ensembles model, while Subsection 4.3 illustrates how the individual pruned members of the ensemble are obtained.

## 4.1 TRANSFORMER PRELIMINARIES.

A layer $l$ of a pre-norm Transformer with $L$ layers, hidden size $d$, $H$ attention heads (head dim. $d_k = d/H$) and sequence length $T$, processes an input $X_{i,\ell} \in \mathbb{R}^{T \times d}$ as follows:

$$\widehat{X}_{i,\ell} = \text{LN}(X_{i,\ell}), \qquad Y_{i,\ell} = X_{i,\ell} + \text{MHA}(\widehat{X}_{i,\ell}), \qquad X_{i,\ell+1} = Y_{i,\ell} + \text{MLP}(\text{LN}(Y_{i,\ell})). \quad (1)$$

In multi-head attention (MHA), the input $\widehat{X}_\ell$ is linearly projected into $H$ sets of queries $Q^{(h)}$, keys $K^{(h)}$, and values $V^{(h)}$, one for each head $h$. The output of head $h$ is computed as $Z^{(h)} = \text{softmax}\left(\frac{Q^{(h)}(K^{(h)})^\top}{\sqrt{d_k}}\right)V^{(h)}$, and concatenates all heads. The Multi Layer Perceptron (MLP) is a two-layer feed-forward network with nonlinearity $\sigma$.

## 4.2 PROPOSED METHOD

We introduce Hydra Ensembles, an efficient ensemble method that prunes at the attention-head level.

**Why prune attention heads?** Each Transformer layer has two main parts: MHA and MLP. MoE methods (Fedus et al., 2022) usually operate pruning on the MLPs, since they contain more parameters (e.g., in ViT-B/16: MLP = 4.7M vs. MHA = 2.3M). In this paper instead, we propose to prune attention heads because: *(i)* MoE already exploits MLP specialization, so pruning MLP will turn them less compatible with MoE architectures; *(ii)* Head-level pruning is simpler for model merging, since adding heads is cheaper than merging full MLPs and better fits circuit extraction.

**Hydra Ensembles Setup.** Assume we have $M$ pruned models $\{f_{\tilde{\boldsymbol{\theta}}^{(m)}}\}_{m=1}^M$, all variants of the same original trained model $f_{\boldsymbol{\theta}}$. Each differs only in the set of surviving heads after pruning.

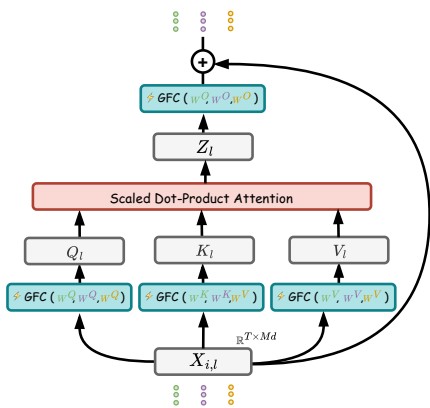

Figure 2: Illustration of **Fused MHA.**

For layer $\ell$, the input is the concatenation: $X_{i,\ell} = \begin{bmatrix} X_{i,\ell}^{(1)} & X_{i,\ell}^{(2)} & \dots & X_{i,\ell}^{(M)} \end{bmatrix}^\top \in \mathbb{R}^{MT \times d}$, where $X_{i,\ell}^{(m)}$ is the input for model $m$. Our goal is to build a single Transformer that fuses these $M$ models, performing ensemble inference in a *single forward pass*. This new transformer architecture consists of two principal components: the Fused MLP and the Fused MHA, which we describe below.

**Merged MLP.** At each MLP layer, we average the weights and biases across the $M$ models:

$$\overline{W}_\ell^{(j)} = \tfrac{1}{M} \sum_m W_\ell^{(j)(m)}, \quad \overline{b}_\ell^{(j)} = \tfrac{1}{M} \sum_m \boldsymbol{b}_\ell^{(j)(m)},$$

Here, the superscript $j$ on the weights and biases indicates the $j$-th fully-connected layer of the MLP. This produces a merged MLP $\text{MLP}_\ell^{(\text{merge})}$.

**Fused MHA.** The input at layer $\ell$ is first reshaped: $X_{i,\ell} \in \mathbb{R}^{MT \times d} \quad \mapsto \quad \tilde{X}_{i,\ell} \in \mathbb{R}^{T \times Md}$.

Each pruned model $m$ retains its own set of active heads, with a total dimension $d_\ell = H_\ell d_k$, where $H_\ell$ denotes the number of heads remaining after pruning. For each model $m$, the projection matrices are:

$$W_\ell^{Q(m)} \in \mathbb{R}^{d \times d_\ell}, \quad W_\ell^{K(m)} \in \mathbb{R}^{d \times d_\ell}, \quad W_\ell^{V(m)} \in \mathbb{R}^{d \times d_\ell}.$$

Using Grouped Fully-Connected (GFC) layers (Xie et al., 2017; Laurent et al., 2023; Lafage et al., 2025a), queries, keys, and values are computed jointly:

$$Q_\ell = \text{GFC}(\tilde{X}_\ell; W_\ell^{Q(1)}, W_\ell^{Q(2)}, \dots, W_\ell^{Q(M)}) \in \mathbb{R}^{T \times Md_\ell},$$

and similarly for $K_\ell$ and $V_\ell$.

The attention heads are then computed as:

$$A_\ell^{(h)} = \mathrm{softmax}\left( \tfrac{1}{\sqrt{d_k}} Q_\ell^{(h)} (K_\ell^{(h)})^\top \right) \in \mathbb{R}^{T \times T}, \tag{2}$$

$$Z_\ell^{(h)} = A_\ell^{(h)} V_\ell^{(h)} \in \mathbb{R}^{T \times d_k}. \tag{3}$$

Let $Z_\ell \in \mathbb{R}^{T \times 3d_\ell}$ be the concatenation of all heads. The fused MHA is then:

$$\mathrm{MHA}_\ell^{(\mathrm{fuse})}(X_{i,\ell}) = \mathrm{GFC}(Z_\ell; W_\ell^{O(1)}, W_\ell^{O(2)}, \dots, W_\ell^{O(M)}) \in \mathbb{R}^{T \times Md},$$

where $W_\ell^{O(m)} \in \mathbb{R}^{d_\ell \times d}$ are the output projection weights. Please refer to Fig.2 for an illustration.

**Final fused Transformer layer.** The fused Transformer layer is therefore:

$$\widehat{X}_{i,\ell} = \mathrm{LN}_\ell^{(1)}(X_{i,\ell}) \in \mathbb{R}^{MT \times d}, \tag{4}$$

$$Y_{i,\ell} = \mathrm{reshape}_{MT \times d \,\mapsto\, T \times Md}(X_{i,\ell}) + \mathrm{MHA}_\ell^{(\mathrm{fuse})}\left( \mathrm{reshape}_{MT \times d \,\mapsto\, T \times Md}(\widehat{X}_{i,\ell}) \right) \in \mathbb{R}^{T \times Md}, \tag{5}$$

$$\widehat{Y}_{i,\ell} = \mathrm{LN}_\ell^{(2)}\left( \mathrm{reshape}_{T \times Md \,\mapsto\, MT \times d}(Y_{i,\ell}) \right) \in \mathbb{R}^{MT \times d}, \tag{6}$$

$$X_{i,\ell+1} = \mathrm{reshape}_{T \times Md \,\mapsto\, MT \times d}(Y_{i,\ell}) + \mathrm{MLP}_\ell^{(\mathrm{merge})}(\widehat{Y}_{i,\ell}) \in \mathbb{R}^{MT \times d}. \tag{7}$$

In summary, the final fused Transformer block first normalizes the inputs from different models, then applies fused MHA on a reshaped representation where model-specific inputs are stacked along the depth dimension, reducing the token load for attention. The model-specific outputs are then reshaped back to the token dimension, passed through an MLP with residual connections, producing the next-layer representation. This final layer with reshaping ensures that $\mathrm{MHA}^{(\mathrm{fuse})}$ processes the same number of tokens as a standard MHA, while $\mathrm{MLP}^{(\mathrm{merge})}$ uses the same number of features, allowing it to be used without fine-tuning. This greatly improve efficiency (see Appendix B.4).

### 4.3 How to construct ensemble members

We have illustrated the procedure for merging multiple models into a single ensemble. We now describe our approach for creating the individual ensemble members, proposing two strategies depending on the available data.

**Strategy 1: No access to an uncertainty validation set.** If no validation set is available, we propose to use a classical structured pruning with the Taylor method (Molchanov et al., 2019). This prunes the least important heads while keeping the architecture intact. This approach is computationally efficient and well-suited for structured pruning at the head level, making it possible to remove the least important heads while keeping the overall model architecture. We denote this technique Hydra Ensembles (Taylor).

**Strategy 2: Access to an uncertainty validation set.** If noisy or uncertainty-focused validation data exists, we use circuits. In particular, the *Headmap* method (Wang et al., 2025) identifies which heads matter most for uncertainty, and removes the rest. This strategy is more targeted than standard Taylor pruning, as it explicitly considers extracting circuits for a given task. We denote this technique Hydra Ensembles (Circuit).

**Fine-tuning or zero-shot usage?** Pruned models can either be fine-tuned to recover accuracy, or used directly in a zero-shot way. Based on Proposition 1, we argue that when pruning is performed without circuit-based strategies (e.g., using only Taylor pruning), it is generally preferable to fine-tune the models. This is because zero-shot pruned models may lack robustness when applied to uncertainty quantification tasks.

When fine-tuning is chosen, each pruned model is trained independently on the same dataset. For example, in the case of an ensemble with $M$ members, we obtain $M$ distinct pruned models $f_{\tilde{\theta}^{(m)}}$. These models are then fine-tuned on the same training dataset by solving the following optimization problem: $\theta^{(m)\star} = \arg\min_{\theta^{(m)}} \mathcal{L}_\mathcal{D}(\theta^{(m)})$. This procedure produces $M$ specialized models. Appendix B.5 analyzes the diversity introduced by this fine-tuning strategy.

For supervised image and text classification, our method is fine-tuned; for zero-shot image classification, it is not. All experiments use $M = 3$ for ensembling, a standard choice in Deep Ensembles.

## 5 EXPERIMENTAL RESULTS

We evaluate our method on three tasks: supervised image classification, zero-shot image classification, and text classification. For supervised tasks, models are fine-tuned; for zero-shot, they are not. All experiments use $M = 3$ for ensembling, a standard choice, see e.g. (Lakshminarayanan et al., 2017). The average number of heads per-layer is indicated in each section. Full implementation details of our experiments are in Appendix C. We also provide a short study on cost/benefit of using more than 3 members in Hydra Ensembles in Appendix B.6

### 5.1 SUPERVISED IMAGE CLASSIFICATION

**Datasets, metrics and architecture.** We evaluate our approach considering both the ImageNet-1K (Russakovsky et al., 2015) and CIFAR-100 (Krizhevsky, 2009) datasets. For ImageNet-1K, we use the standard train/validation split, while for CIFAR-100 we consider the official train/test split. We perform our evaluation considering ViT-B/16 as backbone (Dosovitskiy et al., 2020). Unless stated otherwise, ViT backbones are pre-trained on ImageNet-21K (Ridnik et al., 2021) and fine-tuned on the target dataset. Following previous works (Lafage et al., 2025b), for In-Distribution (ID) evaluation, we report Top-1 Accuracy along with standard calibration metrics: Brier score, negative log-likelihood (NLL), expected calibration error (ECE) (Guo et al., 2017), and adaptive ECE (aECE). For out-of-distribution (OOD), we report AUROC, FPR95 and AUPR (Hendrycks & Gimpel, 2017) and follow the *OpenOOD* benchmark (Yang et al., 2022) splits for both datasets.

**Baselines.** We compare several methods: *(i)* a single transformer architecture (SINGLE), acting as a reference; *(ii)* the state-of-the-art approach DEEP ENSEMBLES (Lakshminarayanan et al., 2017), built from *independently pretrained and trained*[2] ViT-B/16 models with different random seeds; *(iii)* efficient ensemble methods such as PACKED-ENSEMBLES (Laurent et al., 2023), BATCH ENSEMBLES (Wen et al., 2020) and MIMO (Havasi et al., 2020); *(iv)* methods based on parameter-efficient adapters or dropout (LORA ENSEMBLES (Mühlematter et al., 2024) and MC DROPOUT (Gal & Ghahramani, 2016)) *(v)* previous pruning baselines such as the unstructured pruning method OBA[3] (Sun et al., 2025), the structured Taylor-based method in (Molchanov et al., 2019) (TAYLOR), and CIRCAVG, which extracts circuits using OOD and accuracy losses (see Appendix D.1). We also test two variants of our method: HYDRA ENSEMBLES (TAYLOR) and HYDRA ENSEMBLES (CIRC). Details about post pruning fine-tuning in Appendix D.2

**Results.** Table 1 shows the results on classification and OOD detection. On ImageNet-1K, Hydra Ensembles surpasses mostly all baselines and can narrow the gap with Deep Ensembles in terms of accuracy ($-1.3\%$) while improving slightly average OOD performance (AUROC $+0.8\%$ and AUPR $+0.4\%$). Compared with the original transformer model (SINGLE), adapter-based methods and other efficient ensembles, Hydra Ensembles achieves better robustness. A similar trend can be observed for CIFAR-100 dataset. Compared to Deep Ensembles, Hydra Ensembles shows its largest advantage in UQ, consistently improving on average OOD metrics (AUROC $+3.4\%$, FPR95 $-2.2\%$, AUPR $+1.2\%$). Additional results including experiments on distribution shift and results reporting mean/std across multiple seeds for Hydra Ensembles are reported in Appendix E.1

Beyond accuracy and robustness, we also examine the computational cost of Hydra Ensembles in Figure 3. Interestingly, in `bfloat16`, Hydra Ensembles achieves nearly the same inference cost as the single model, with a ratio Hydra Ensembles/Single of only $1.07\times$ for both whole-test runtime and per-batch inference. By contrast, the Deep Ensembles approach is almost three times slower under the same conditions and have more than twice as many parameters (see also Appendix B.4).

Although MLP fusion could in principle reduce ensemble diversity when applied post training, but in Hydra Ensembles this does not occur in practice because diversity is already assured by different attention heads representations learned during separate member training, and we observe no degradation in ID or OOD metrics (see Appendix B.7)

---

[2] Each ensemble member uses the same architecture (ViT-B/16) but different random seeds, and is pretrained and fine-tuned separately, yielding diverse weights.

[3] OBA (Sun et al., 2025) prunes weights on the entire model. To match attention-head pruning, we retain 89.1% of parameters for ImageNet-1K (8 heads per MHA block) and 94% for CIFAR-100 ($\approx$ 10 heads per block).

Table 1: **Results on ImageNet-1K and CIFAR-100 datasets**. Metrics evaluate accuracy, calibration performance and OOD detection. Best in bold, second-best underlined.

| Method | | | | | IMAGENET-1K | | | | | | | | | | CIFAR-100 | | | | | |
|---|---|---|---|---|---|---|---|---|---|---|---|---|---|---|---|---|---|---|---|---|
| | Heads | Acc↑ | Brier↓ | NLL↓ | ECE↓ | aECE↓ | OOD Avg | | | Heads | Acc↑ | Brier↓ | NLL↓ | ECE↓ | aECE↓ | OOD Avg | | |
| | | | | | | | AUROC↑ | FPR95↓ | AUPR↑ | | | | | | | AUROC↑ | FPR95↓ | AUPR↑ |
| SINGLE | 12 | 80.67 | 0.27 | 0.71 | **0.01** | **0.01** | 84.40 | 50.25 | 60.91 | 12 | 92.15 | 0.11 | 0.25 | **0.007** | **0.005** | 85.46 | 40.27 | 93.55 |
| DEEP ENSEMBLES | 12 | **82.19** | 0.25 | **0.65** | 0.01 | 0.01 | 85.48 | **46.93** | 62.76 | 12 | **93.52** | **0.09** | **0.22** | 0.01 | 0.01 | 86.08 | 38.67 | 94.26 |
| PACKED ENSEMBLES | 12 | 79.23 | 0.29 | 0.78 | 0.01 | 0.01 | 83.26 | 51.65 | 58.17 | 12 | 90.63 | 0.13 | 0.31 | 0.01 | 0.008 | 86.99 | 38.54 | 93.48 |
| MIMO | 12 | 80.59 | 0.27 | 0.72 | 0.01 | 0.01 | 83.63 | 52.64 | 59.14 | 12 | 92.62 | 0.10 | 0.23 | 0.009 | 0.008 | 88.08 | 37.00 | 95.15 |
| BATCH ENSEMBLES | 12 | 80.53 | 0.27 | 0.72 | 0.01 | 0.01 | 84.34 | 50.38 | 60.71 | 12 | 92.19 | 0.11 | 0.26 | 0.008 | 0.006 | 86.27 | 39.66 | 93.66 |
| MC DROPOUT | 12 | 80.3 | 0.28 | 0.73 | 0.02 | 0.02 | 83.7 | 51.44 | 58.9 | 12 | 92.04 | 0.11 | 0.25 | 0.01 | 0.01 | 84.74 | 42.53 | 93.12 |
| LORA ENSEMBLES | 12 | 80.68 | 0.27 | 0.71 | 0.01 | 0.01 | 84.24 | 50.59 | 60.35 | 12 | 92.14 | 0.11 | 0.26 | 0.007 | 0.006 | 85.77 | 40.18 | 93.53 |
| OBA | – | 78.52 | 0.30 | 0.85 | 0.03 | 0.03 | 82.61 | 54.90 | 54.89 | – | 91.88 | 0.11 | 0.26 | **0.007** | **0.005** | 85.85 | 40.48 | 93.41 |
| TAYLOR | 8 | 80.68 | 0.28 | 0.79 | 0.02 | 0.02 | 84.38 | 54.51 | 59.46 | 10 | 91.59 | 0.12 | 0.31 | 0.01 | 0.01 | 88.79 | 40.35 | 95.03 |
| CIRCAVG | 8 | 80.22 | 0.28 | 0.77 | 0.02 | 0.02 | 85.71 | 50.23 | 62.41 | 10 | 91.67 | 0.12 | 0.30 | 0.01 | 0.01 | **89.77** | 37.79 | 95.21 |
| HYDRA ENS (TAYLOR) | 8x3 | 81.20 | 0.26 | 0.75 | **0.01** | **0.01** | 85.36 | 50.50 | 60.75 | 10x3 | 92.00 | 0.11 | 0.28 | 0.008 | 0.007 | 88.89 | 39.57 | **95.46** |
| HYDRA ENS (CIRC) | 8x3 | 80.88 | 0.27 | 0.74 | **0.01** | **0.01** | **86.29** | 47.62 | **63.15** | 10x3 | 92.11 | 0.12 | 0.28 | 0.008 | 0.006 | 89.43 | **36.44** | 95.17 |

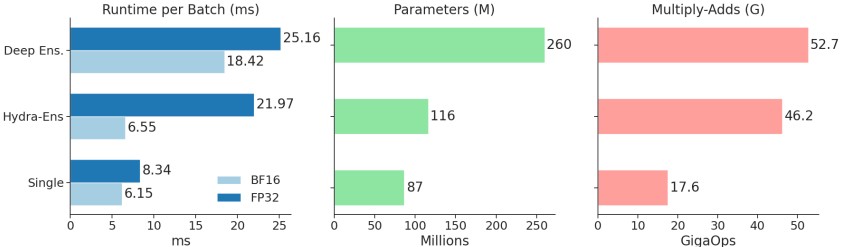

Figure 3: **Inference costs** of Deep Ensembles, Single, and Hydra Ensembles with 8 heads on ImageNet-1k. We report runtime per batch under BF16/FP32 (left), as well as parameter count (middle) and multiply-adds (right).

## 5.2 SUPERVISED TEXT CLASSIFICATION

**Datasets and architecture.** We also test on SST-2 (Socher et al., 2013), a sentiment analysis benchmark with binary labels, using a `bert-base-uncased` model (Devlin et al., 2019) fine-tuned on the SST-2 training set (Socher et al., 2013).

**Baselines.** We consider similar baselines than in the image classification setting. However, for the Deep Ensembles baseline, we avoid training three completely independent BERT (Devlin et al., 2019) models from scratch to reduce computational cost. Instead, we fine-tune a shared pre-trained BERT backbone and train three separate classifiers with different random seeds on top of it (we denote it as DEEP ENSEMBLES (D)). More generally, in the supervised text classification setting, we only include those baselines from the image classification experiments that do not require re-training large models from scratch, due to the computational complexity of BERT. For all pruning approaches, we keep 6 heads per attention block. Please refer to Appendix D.2 for details on fine-tuning. We consider both variants of our method.

**Results.** Table 2 reports ID accuracy, calibration, and OOD detection performance for different uncertainty methods. Deep Ensembles achieves the best overall performance but at high computational cost—about three times a single model, while Hydra Ensembles in float16 they match single-model time. Compared to Deep Ensembles (D), classical lightweight approaches such as MC Dropout and LoRA Ensembles are close in accuracy but show weaker OOD detection. Our proposed Hydra Ensembles performs on par with it in terms of accuracy and calibration, while significantly improving OOD detection: AUROC ($+2.8\%$), FPR95 ($-7.6\%$) and AUPR ($+2.2\%$).

## 5.3 ZERO-SHOT IMAGE CLASSIFICATION

**Datasets and architecture.** We consider the OpenCLIP-ViT/B-32 model (Ilharco et al., 2021) and nine standard datasets covering diverse domains: ImageNet-1k, CIFAR-100 and CIFAR-10, Food101 (Bossard et al., 2014), SUN397 (Xiao et al., 2010), Oxford Pets (Parkhi et al., 2012), DTD (Cimpoi et al., 2014), EuroSAT (Helber et al., 2018) and Caltech101 (Fei-Fei et al., 2004).

Table 2: **Results on text classification.** Metrics evaluate accuracy, calibration performance and OOD detection. Best in bold, second-best underlined.

| Method | Heads | Acc↑ | Brier↓ | NLL↓ | ECE↓ | aECE↓ | OOD Average | | |
|--------|-------|------|--------|------|------|-------|------|------|------|
| | | | | | | | AUROC↑ | FPR95↓ | AUPR↑ |
| SINGLE | 12 | 92.55 | 0.12 | 0.27 | 0.05 | **0.04** | 70.16 | 70.62 | 81.93 |
| DEEP ENSEMBLES (D) | 12 | **93** | **0.11** | 0.24 | **0.04** | **0.04** | 74.81 | 62.69 | **84.9** |
| MC DROPOUT | 12 | 92.55 | 0.13 | 0.31 | 0.05 | 0.04 | 72.23 | 67.36 | 81.96 |
| LORA ENSEMBLES | 12 | 92.89 | 0.12 | 0.28 | 0.05 | **0.04** | 70.83 | 68.7 | 82.27 |
| TAYLOR | 6 | 92.43 | 0.13 | 0.35 | 0.06 | 0.06 | 70.95 | 68.57 | 82.11 |
| CIRCAVG | 6 | 92.78 | 0.13 | 0.33 | 0.05 | 0.05 | 75.04 | 56.22 | 84.18 |
| HYDRA-ENS (TAYLOR) | 6x3 | **93** | 0.12 | 0.29 | 0.05 | **0.04** | 71.85 | 62.59 | 82.44 |
| HYDRA-ENS (CIRC) | 6x3 | 92.55 | 0.12 | **0.24** | **0.04** | **0.04** | **77.6** | **55.06** | 84.16 |

Table 3: **Results on zero-shot classification**. Metrics evaluate accuracy, calibration performance and on the ImageNet-1k OOD set. Best in bold, second-best underlined.

| Method | Heads | Train | Acc↑ | Brier↓ | NLL↓ | ECE↓ | aECE↓ | OOD Average | | |
|--------|-------|-------|------|--------|------|------|-------|------|------|------|
| | | | | | | | | AUROC↑ | FPR95↓ | AUPR↑ |
| SINGLE | 12 | - | 73.65 | 0.36 | 0.98 | 8.68 | 8.57 | 70.76 | 75.20 | 37.73 |
| TEMP. SCALING | 12 | - | 73.65 | 0.36 | **0.90** | 4.03 | 3.80 | 70.76 | 75.20 | 37.73 |
| BAYESVLM | 12 | ✓ | 73.64 | **0.35** | 0.93 | 5.95 | 5.86 | 71.84 | 74.65 | 39.08 |
| VILU | 12 | ✓ | - | - | - | 9.50 | 8.86 | 75.38 | 71.59 | 43.81 |
| TAYLOR | 10 | - | 60.75 | 0.53 | 1.65 | 13.76 | 13.67 | 67.44 | 79.41 | 33.64 |
| CIRCAVG | 10 | - | 71.91 | 0.38 | 0.99 | 4.00 | 4.11 | **76.88** | 68.26 | 47.64 |
| HYDRA-ENS (CIRC) | 10x3 | - | **74.00** | 0.36 | 0.93 | **3.49** | **3.35** | 76.82 | **68.05** | **47.85** |

All datasets but SUN397 come from `torchvision`, whose source instead is Hugging Face. Following Radford et al. (2021), we use `"A photo of a {label}."` as prompt template. See Appendix C.3 for additional details.

**Baselines.** We compare our method against recent UQ baselines which care developed based on CLIP[4]: *(i)* BAYESVLM (Baumann et al., 2024), which improves uncertainty estimation via a Laplace approximation using a subset of LAION-400M (Schuhmann et al., 2021); *(ii)* VILU (Lafon et al., 2025), which trains an uncertainty predictor as a dataset-specific binary classifier to distinguish correct from incorrect predictions, thereby affecting only ECE and OOD performance; and *(iii)* TEMPERATURE SCALING (Guo et al., 2017), applied on a small portion of the training set of ImageNet-1k following the setup described in Baumann et al. (2024). As in previous experiments, we also report the results obtained with pruning-based approaches. For our method, we propose two setups: one with pruning only on the vision encoder, and another with both encoders pruned. Moreover, we test the effect of fine-tuning on a small subset of LAION-400M for the first one. Here we present the results with both encoders pruned. See Appendix D.3 for the details about the implementation and Appendix E.3.2 for the results with the pruning only on the vision encoder in zero-shot and fine-tuned settings. Here, we only report the performance of the best-performing version of our method, Hydra Ensembles (Circ).

**Results.** Table 3 presents our results for classification averaged across the datasets, and for the ImageNet-1k OOD set, following the same protocol as in the supervised setting. Disaggregated results and on the OOD of CIFAR100, where Hydra Ensembles improves on the baselines as well, are reported in Appendix E.3.1. Hydra Ensembles achieves the best accuracy, ECE, aECE and the second-best NLL, *all without any training*. For OOD detection (see also disaggregated results in the Appendix), the only competitive baseline is ViLU, but our method surpasses it with substantially better AUROC $(+1.3\%)$, FPR95 $(-3.5\%)$ and AUPR $(+4.0\%)$.

## 6 CONCLUSIONS

In this work, we introduced *Hydra Ensembles*, a structured pruning approach for building efficient transformer ensembles. By aggregating diverse pruned heads into a single model, Hydra Ensembles provides uncertainty estimates comparable to, and in some cases surpassing, Deep Ensembles,

---

[4]Many UQ methods cannot be transferred directly to CLIP.

while maintaining near-single-model inference speed. Experiments on both large-scale and small-scale image classification datasets, text classification, and zero-shot image classification demonstrate substantial improvements in OOD detection and calibration, occasionally even without retraining. We further show, both empirically and theoretically, that structured pruning of attention heads can meaningfully affect UQ, enhancing the separation between ID and OOD representations when heads are carefully selected, whereas naive or unstructured pruning may be detrimental. These results indicate that careful head selection preserves both accuracy and reliable uncertainty, positioning Hydra Ensembles as a scalable, memory-efficient, and robust framework for uncertainty quantification in modern deep learning models.

### REPRODUCIBILITY STATEMENT

We have taken multiple steps to ensure the reproducibility of our work. All experimental setups are described in detail in Section 5, including datasets, evaluation metrics, and architectures. Additional implementation details, such as dataset splits, hyperparameter settings, pruning strategies, and fine-tuning protocols, are provided in the Appendix (see in particular Appendix C.1.5, C, and D.2). For zero-shot experiments, we describe our CLIP baselines and pruning variants in Appendix C.3 and D.3, along with extended results. All theoretical assumptions and proofs are presented in the main text and appendix. We will release the complete source code, training scripts, and pretrained checkpoints upon publication to further facilitate reproducibility and extension of our results.

### ACKNOWLEDGMENTS

This work was granted access to the HPC resources of IDRIS under the allocation AD011015965R1 , AD011016078 , and the allocation AD011016325 made by GENCI. The authors also acknowledge the CINECA award under the ISCRA initiative for the availability of high-performance computing resources and support.

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

TABLE OF CONTENTS - SUPPLEMENTARY MATERIAL

# A ON PRUNING AND UNCERTAINTY QUANTIFICATION

In this section we first recall the classical pruning framework, including its derivation via second-order loss approximations, to set the stage for later results. We then extend this analysis to noisy data conditions, illustrating why conventional pruning can be especially harmful when test distributions are corrupted. Together, these sections clarify both the foundations and the limitations of pruning, motivating our proposed ensemble-based approach.

## A.1 PRELIMINARIES

Let us still consider the DNN $f_{\boldsymbol{\theta}}(\cdot)$. A pruning algorithm aims to find a subset of weights $\tilde{\boldsymbol{\theta}} \in \mathbb{R}^d$ with $d < D$ such that $\mathcal{L}_{\mathcal{D}}(\tilde{\boldsymbol{\theta}}) = \mathcal{L}_{\mathcal{D}}(\boldsymbol{\theta})$, so that the pruned model is lighter while maintaining accuracy on the test set: $\mathcal{L}_{\mathcal{D}^t}(\tilde{\boldsymbol{\theta}}) = \mathcal{L}_{\mathcal{D}^t}(\boldsymbol{\theta})$.

Following the Optimal Brain Damage (OBD) method (LeCun et al., 1989), one studies the effect of a small perturbation $\delta\boldsymbol{\theta}$ on the loss via a second-order Taylor expansion:

$$\delta\mathcal{L}_{\mathcal{D}}(\boldsymbol{\theta}) \triangleq \mathcal{L}_{\mathcal{D}}(\boldsymbol{\theta} + \delta\boldsymbol{\theta}) - \mathcal{L}_{\mathcal{D}}(\boldsymbol{\theta}) = \underbrace{\left(\nabla\mathcal{L}_{\mathcal{D}}(\boldsymbol{\theta})\right)^{\top}\delta\boldsymbol{\theta}}_{\text{first order}} + \tfrac{1}{2}\delta\boldsymbol{\theta}^{\top}\underbrace{\nabla^2\mathcal{L}_{\mathcal{D}}(\boldsymbol{\theta})}_{\mathbf{H}}\delta\boldsymbol{\theta} + o(\|\delta\boldsymbol{\theta}\|^3), \quad (8)$$

where $\mathbf{H} \in \mathbb{R}^{n \times n}$ is the Hessian of the loss.

Similar to Optimal Brain Surgeon (OBS) (Hassibi & Stork, 1992), the classical assumption is that the DNN is trained to a local minimum, so that the gradient term vanishes, and higher-order terms are neglected. Hence, the pruning problem can be formulated as

$$\min_q \left\{ \min_{\delta\boldsymbol{\theta}} \tfrac{1}{2}\delta\boldsymbol{\theta}^{\top}\mathbf{H}\,\delta\boldsymbol{\theta} \ \text{ s.t. } \ \mathbf{e}_q^{\top}\delta\boldsymbol{\theta} + \theta_q^* = 0 \right\},$$

where the constraint enforces elimination of the weight $\theta_q^*$, while the quadratic term controls the increase in loss.

## A.2 PRUNING UNDER NOISY DATA CONDITIONS

This strategy is effective on a clean test dataset $\mathcal{D}^t$, leading to a wide range of structured (Kurtić et al., 2023; Kwon et al., 2022; Park et al., 2023) and unstructured pruning approaches (Kurtic et al., 2022; Yao et al., 2021; Xu et al., 2022). However, in the presence of a noisy or corrupted test dataset $\mathcal{D}^n$, pruning may degrade performance (Liebenwein et al., 2021).

*Assumption* 2 (Clean vs. Noisy Datasets). Let $\mathcal{D}^t$ denote a clean test dataset and $\mathcal{D}^n$ a noisy (or corrupted) test dataset. Define the loss gap between the two as

$$\Delta\mathcal{L}(\boldsymbol{\theta}) := \mathcal{L}_{\mathcal{D}^n}(\boldsymbol{\theta}) - \mathcal{L}_{\mathcal{D}^t}(\boldsymbol{\theta}).$$

We assume that $\boldsymbol{\theta}$ minimizes $\mathcal{L}_{\mathcal{D}}$ (training set) and $\mathcal{L}_{\mathcal{D}^t}$ (clean test set), i.e.

$$\nabla\mathcal{L}_{\mathcal{D}}(\boldsymbol{\theta}) = 0, \text{ and } \nabla\mathcal{L}_{\mathcal{D}^t}(\boldsymbol{\theta}) = 0,$$

and that pruning induces a small perturbation $\delta\boldsymbol{\theta}$. Also we assume that the pruning perturbation $\delta\boldsymbol{\theta}$ is aligned (non-negatively correlated) with the gradient of the noisy loss :

$$\left(\nabla\mathcal{L}_{\mathcal{D}^n}(\boldsymbol{\theta})\right)^{\top}\delta\boldsymbol{\theta} \geq 0,$$

**Proposition 1** (Pruning is Worse under Noise) Suppose Assumption 1 holds. Let $\mathbf{H}^t$ and $\mathbf{H}^n$ denote the Hessians of the loss on $\mathcal{D}^t$ and $\mathcal{D}^n$, respectively. If

$$\mathbf{H}^n - \mathbf{H}^t \succ 0,$$

then the loss gap after pruning satisfies

$$\Delta\mathcal{L}(\boldsymbol{\theta}) \leq \Delta\mathcal{L}(\boldsymbol{\theta} + \delta\boldsymbol{\theta}),$$

i.e., pruning degrades performance more severely on the noisy dataset than on the clean one.

*Proof.* By the second-order Taylor expansion (cf. Eq. equation 8), the difference in loss gaps before and after pruning is

$$\Delta\mathcal{L}(\boldsymbol{\theta} + \delta\boldsymbol{\theta}) - \Delta\mathcal{L}(\boldsymbol{\theta}) = \delta\mathcal{L}_{\mathcal{D}^n}(\boldsymbol{\theta}) - \delta\mathcal{L}_{\mathcal{D}^t}(\boldsymbol{\theta}).$$

Expanding both terms yields

$$= \left(\nabla\mathcal{L}_{\mathcal{D}^n}(\boldsymbol{\theta}) - \nabla\mathcal{L}_{\mathcal{D}^t}(\boldsymbol{\theta})\right)^\top \delta\boldsymbol{\theta} + \tfrac{1}{2}\delta\boldsymbol{\theta}^\top\left(\mathbf{H}^n - \mathbf{H}^t\right)\delta\boldsymbol{\theta} + o(\|\delta\boldsymbol{\theta}\|^3).$$

Since $\nabla\mathcal{L}_{\mathcal{D}^t}(\boldsymbol{\theta}) = 0$ by Assumption 1, the first-order term vanishes. Thus,

$$\Delta\mathcal{L}(\boldsymbol{\theta} + \delta\boldsymbol{\theta}) - \Delta\mathcal{L}(\boldsymbol{\theta}) = \left(\nabla\mathcal{L}_{\mathcal{D}^n}(\boldsymbol{\theta})\right)^\top \delta\boldsymbol{\theta} + \tfrac{1}{2}\delta\boldsymbol{\theta}^\top\left(\mathbf{H}^n - \mathbf{H}^t\right)\delta\boldsymbol{\theta} + o(\|\delta\boldsymbol{\theta}\|^3).$$

If $\mathbf{H}^n - \mathbf{H}^t \succ 0$ and $\left(\nabla\mathcal{L}_{\mathcal{D}^n}(\boldsymbol{\theta})\right)^\top \delta\boldsymbol{\theta} \geq 0$, the quadratic form is positive for any nonzero $\delta\boldsymbol{\theta}$, which implies

$$\Delta\mathcal{L}(\boldsymbol{\theta}) \leq \Delta\mathcal{L}(\boldsymbol{\theta} + \delta\boldsymbol{\theta}) \qquad and \qquad \delta\mathcal{L}_{\mathcal{D}^t}(\boldsymbol{\theta}) \leq \delta\mathcal{L}_{\mathcal{D}^n}(\boldsymbol{\theta}).$$

showing that pruning amplifies the performance gap under noise. □

This shows that classically pruned DNNs suffer from a stronger degradation of performance than unpruned models when evaluated on noisy data. Consequently, such pruned networks cannot be directly relied upon for uncertainty quantification. This raises a natural question: *how can we leverage ensembles of pruned models in a way that remains effective for uncertainty estimation?*

## B  DISCUSSIONS

### B.1  THE INFLUENCE OF ZEROS SHOT PRUNING AND UNCERTAINTY QUANTIFICATION

Proposition 1 states that pruning on uncertainty data is valid if the following conditions hold: $\left(\nabla\mathcal{L}_{\mathcal{D}^n}(\boldsymbol{\theta})\right)^\top \delta\boldsymbol{\theta} \geq 0$, and $\mathbf{H}^n - \mathbf{H}^t \succ 0$, where the first condition requires that the pruning perturbation $\delta\boldsymbol{\theta}$ is aligned with the gradient, and the second condition requires that the difference between the Hessians is positive-definite.

To verify these assumptions in practice, we apply the TAYLOR pruning strategy to a ViT-B/16 model, evaluating it on ImageNet-1K (Deng et al., 2009) and CIFAR-100 (Krizhevsky, 2009), as described in Section 5.1. For robustness testing, we additionally use their corrupted counterparts, ImageNet-1K-C and CIFAR-100-C (Hendrycks & Dietterich, 2019a), which introduce common synthetic corruptions (e.g., noise, blur, weather effects) to the original test images.

To test these conditions empirically, we first examine the alignment between the gradient and the pruning perturbation. On CIFAR-100, the inner product between the gradient and the pruning perturbation, $\left(\nabla\mathcal{L}_{\mathcal{D}^n}(\boldsymbol{\theta})\right)^\top \delta\boldsymbol{\theta}$, is equal to 1.19, while on ImageNet-1K it is 0.37. In both cases the values are positive, confirming that the perturbation is indeed aligned with the gradient, as required.

For the Hessian condition, we approximate the Hessian by its diagonal due to the computational cost of the full matrix. The difference between the two diagonals is strictly positive across both datasets, with average values of 0.45 for CIFAR-100 and 0.17 for ImageNet-1K. This suggests that the positive-definiteness assumption holds approximately in practice.

To further support the assumptions of Proposition 1, we also evaluate zero-shot pruning on both text and image classification using BERTDevlin et al. (2019) on SST-2Socher et al. (2013) and ViT-B/16 on the previously mentioned datasets (Table 4). In this setting, the Taylor strategy shows mixed behavior: it causes clear larger degradation on some benchmarks and OOD splits (ImageNet-1k), while occasionally yielding mild or even favorable effects in others (CIFAR100). By contrast, CIRCAVG remains consistently stable and overall stronger, providing additional empirical evidence in favor of Proposition 1.

### B.2  STUDYING THE PRUNED HEADS AND UNCERTAINTY QUANTIFICATION

In this section we propose to study the effect of pruned heads on uncertainty quantification. For example, we aim to understand how using Wang et al. (2025) pruning changes and improve the internal representation of OOD. All results are obtained using ViT-B/16 model.

Table 4: Zero-shot Results across Models and Datasets. While TAYLOR pruning shows inconsistent behavior—sometimes degrading OOD performance and other times mildly improving it—CIRCAVG remains more stable across settings.

| Method | ID Dataset | Model | Acc↑ | ECE↓ | Near-OOD Avg | | | Far-OOD Avg | | |
|---|---|---|---|---|---|---|---|---|---|---|
| | | | | | AUROC↑ | FPR95↓ | AUPR↑ | AUROC↑ | FPR95↓ | AUPR↑ |
| SINGLE | ImageNet-1k | ViT-B/16 | 80.67 | 0.01 | 77.96 | 63.08 | 55.29 | 90.84 | 37.42 | 66.53 |
| TAYLOR | ImageNet-1k | ViT-B/16 | 66.46 | 0.01 | 68.14 | 77.88 | 42.87 | 85.96 | 48.29 | 50.02 |
| CIRCAVG | ImageNet-1k | ViT-B/16 | 73.61 | 0.01 | 73.81 | 70.51 | 50.14 | 89.90 | 41.16 | 63.88 |
| SINGLE | CIFAR100 | ViT-B/16 | 92.15 | 0.007 | 89.63 | 37.96 | 89.71 | 84.41 | 40.85 | 94.51 |
| TAYLOR | CIFAR100 | ViT-B/16 | 86.52 | 0.01 | 86.52 | 46.24 | 86.35 | 86.93 | 41.27 | 94.66 |
| CIRCAVG | CIFAR100 | ViT-B/16 | 91.15 | 0.01 | 87.39 | 44.16 | 87.39 | 86.64 | 37.78 | 95.01 |
| SINGLE | SST-2 | bert base | 92.55 | 0.05 | 70.16 | 70.62 | 81.93 | 70.16 | 70.62 | 81.93 |
| TAYLOR | SST-2 | bert base | 90.6 | 0.06 | 58.05 | 88.19 | 99.03 | 73.18 | 64.15 | 73.42 |
| CIRCAVG | SST-2 | bert base | 91.74 | 0.02 | 71.95 | 66.57 | 99.40 | 79.07 | 55.57 | 78.64 |

**1. How does pruning affect model's internal representation?** To answer this question we use the dataset CIFAR-100 (Krizhevsky, 2009) and its OOD benchmark (Yang et al., 2022). To study how attention head pruning impacts internal representations, we extract the CLS token from each head in the last layer. This token is treated as a vector capturing the head's contribution to the overall representation. For each head, we compute the centroid of ID and OOD vectors. The separation between these centroids is then measured using Euclidean and Mahalanobis distances.

Let the output of the last attention layer's attention be:

$$\mathbf{Z} \in \mathbb{R}^{N \times T \times H \times d_k}$$

where $N$ is the number of samples, $T$ the length of the sequence, $H$ is the number of attention heads, and $d_k$ is the head dimension. Here, $\mathbf{Z}_{n,h} \in \mathbb{R}^{d_k}$ is the CLS token vector for sample $n$ at head $h$.

For a set of ID samples $\mathcal{X}_{\mathrm{ID}}$ and OOD samples $\mathcal{X}_{\mathrm{OOD}}$, the centroid for head $h$ is:

$$\mathbf{c}_h^{\mathrm{ID}} = \frac{1}{|\mathcal{X}_{\mathrm{ID}}|} \sum_{n \in \mathcal{X}_{\mathrm{ID}}} \mathbf{Z}_{n,h}, \quad \mathbf{c}_h^{\mathrm{OOD}} = \frac{1}{|\mathcal{X}_{\mathrm{OOD}}|} \sum_{n \in \mathcal{X}_{\mathrm{OOD}}} \mathbf{Z}_{n,h}.$$

The Euclidean distance between ID and OOD centroids for head $h$ is:

$$d_h^{\mathrm{Eucl}} = \left\| \mathbf{c}_h^{\mathrm{ID}} - \mathbf{c}_h^{\mathrm{OOD}} \right\|_2.$$

Let $\Sigma_h$ be the covariance of the ID representations for head $h$:

$$\Sigma_h = \mathrm{Cov}\big(\{\mathbf{Z}_{n,h} \mid n \in \mathcal{X}_{\mathrm{ID}}\}\big).$$

Then the Mahalanobis distance is:

$$d_h^{\mathrm{Mah}} = \sqrt{(\mathbf{c}_h^{\mathrm{ID}} - \mathbf{c}_h^{\mathrm{OOD}})\Sigma_h^{-1}(\mathbf{c}_h^{\mathrm{ID}} - \mathbf{c}_h^{\mathrm{OOD}})^\top}.$$

Finally, we average over all heads:

$$\bar{d}^{\mathrm{Eucl}} = \frac{1}{H} \sum_{h=1}^{H} d_h^{\mathrm{Eucl}}, \quad \bar{d}^{\mathrm{Mah}} = \frac{1}{H} \sum_{h=1}^{H} d_h^{\mathrm{Mah}}.$$

We report the results for CircOOD as defined in D.1.2 and the dense base model in Table 5, which shows how across heads and datasets, CircOOD consistently improves separation, achieving an average Euclidean distance of 2.26 and Mahalanobis distance of 3.32, compared to 1.75 and 3.03 for the dense model.

Additionally, in Figure 4, we apply PCA to a batch of the MNIST (Lecun et al., 2002) dataset samples to visualize the internal representations of the six most and least important heads—corresponding respectively to the last and first heads pruned during CircOOD extraction. The plot clearly shows that the most important heads produce a larger separation between ID and OOD data, whereas the least important heads can even be detrimental.

Table 5: Disaggregated Euclidean and Mahalanobis distances between `CLS` token centroids of ID and OOD samples across datasets. The green values indicate the improvement (Circuit – Dense).

| Dataset | CircuitOOD | | Dense Model | |
|---|---|---|---|---|
| | Euclidean ↑ | Mahalanobis ↑ | Euclidean ↑ | Mahalanobis ↑ |
| Texture | 1.66 (+0.34) | 2.59 (+0.23) | 1.32 | 2.36 |
| MNIST | 4.92 (+1.23) | 7.03 (+0.83) | 3.69 | 6.20 |
| SVHN | 2.10 (+0.37) | 3.21 (+0.03) | 1.73 | 3.18 |
| Places365 | 1.37 (+0.33) | 1.98 (+0.21) | 1.04 | 1.77 |
| CIFAR-10 | 1.25 (+0.29) | 1.77 (+0.12) | 0.96 | 1.65 |
| Average | 2.26 (+0.51) | 3.32 (+0.29) | 1.75 | 3.03 |

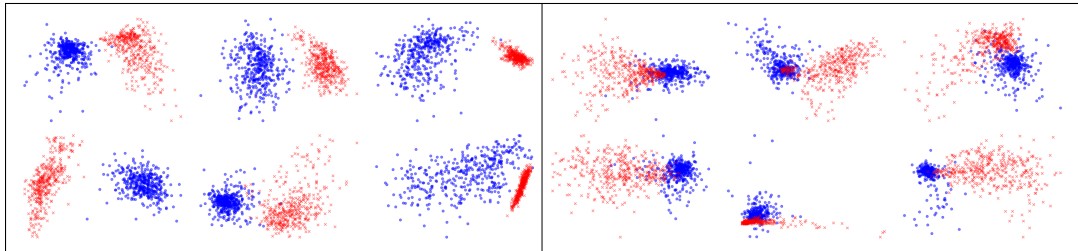

Figure 4: Internal representation of the six most important (left) and least important (right) attention heads for OOD and ID data using PCA. The representation on the left shows a clearer separation between the two distributions, while on the right it illustrates how the distributions sometimes even collapse onto each other.

Table 6: Comparison between circuits on ID classification and OOD detection on ImageNet-1k with ViT-B/16. Each circuit shows superior performance on its respective task.

| Method | Heads | Acc↑ | Near-OOD | | | Far-OOD | | | OOD Avg (all OOD) | | |
|---|---|---|---|---|---|---|---|---|---|---|---|
| | | | AUROC↑ | FPR95↓ | AUPR↑ | AUROC↑ | FPR95↓ | AUPR↑ | AUROC↑ | FPR95↓ | AUPR↑ |
| CircOOD | 8 | 71.11 | 74.26 | 70.36 | 50.75 | 89.52 | 40.11 | 61.77 | 81.89 | 55.24 | 56.26 |
| CircAcc | 8 | 73.41 | 73.46 | 71.26 | 48.91 | 88.15 | 44.11 | 58.29 | 80.80 | 57.69 | 53.60 |

**2. Why does Circuit outperform Taylor pruning?** Table 4 shows that, in a training-free setting, CircAvg consistently outperforms Taylor and in some cases even surpasses the dense model on OOD performance. To illustrate how different the two resulting sub-networks are, we examine the sets of heads they select and find that they overlap by only 25%. One might suspect this low overlap is simply due to Taylor pruning considering importance scores within each layer. To verify that this is not the case, we apply Taylor pruning globally across all layers. The overlap increases only slightly to 29%, and this global view also allows us to estimate the difference between the two sub-networks: the Jensen–Shannon divergence between the discrete distributions of selected heads is 0.37, highlighting the substantial distinction between them.

**3. Does the specific Circuit excel at its task?** As described in Appendix D.1.2, we extract three different circuits, each optimized for a different criterion. To validate this, we evaluate individually CircOOD and CircAcc on ImageNet-1K (Deng et al., 2009) and its OOD set (Yang et al., 2022). The results in Table 6 confirm the expected behavior: CircAcc achieves higher classification accuracy (+2.30%), while CircOOD provides stronger OOD detection, yielding improvements of +1.09% in AUROC, -2.45% in FPR95, and +2.66% in AUPR.

## B.3 EFFECT OF THE NUMBER OF HEADS

We conduct an ablation study on the number of attention heads removed per block (2, 4, 6, 8) and compare Taylor and CircAvg in both the zero-shot and the fine-tuning setup. We report Top-1 accuracy (ID) and OOD metrics (AUROC, FPR95, AUPR) with ViT-B/16 on ImageNet-1K (Deng et al., 2009)

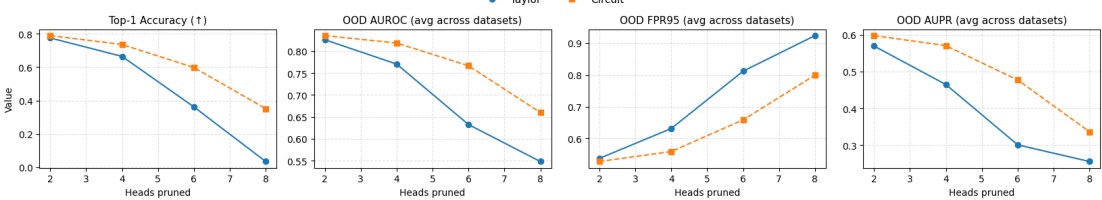

Figure 5: Effect of pruning different number of heads count using Taylor and CircAvg for ViT-B/16 on ImageNet-1K (Zero shot)

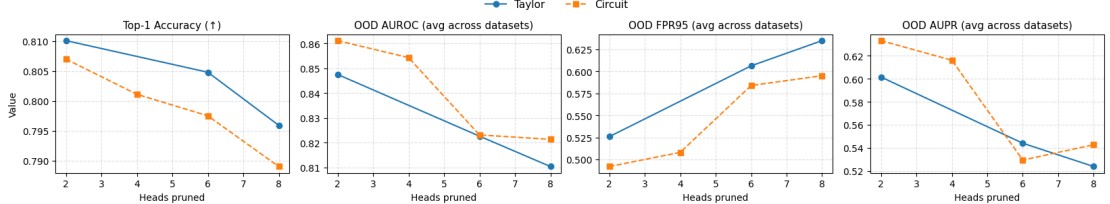

Figure 6: Effect of pruning different number of heads count using Taylor and CircAvg for ViT-B/16 on ImageNet-1K (Post training)

**Zero-shot (no training).** Figure 5 shows that pruning degrades all metrics as the number of heads pruned increases, with FPR95 rising rapidly. The circuit approach, however, degrades more slowly than Taylor across accuracy, AUROC, and AUPR.

**After training.** Fine-tuning each pruned model largely recovers performance as displayed in Figure 6. Top-1 accuracy stays close to the base dense model (1); Taylor pruning is the best at preserving accuracy and CircAvg pruning gives the best OOD performance across all metrics.

### B.4 COMPUTATIONAL COST OF HYDRA ENSEMBLES

We now compare the parameter cost of Hydra Ensembles to that of a standard Transformer. For clarity, we focus on a single Transformer block, which is composed of a multi-head self-attention (MHSA) module followed by a Multi-Layer Perceptron (MLP).

**Standard Transformer.** A Transformer with hidden dimension $d$ and $H$ attention heads splits the hidden dimension evenly across heads:

$$d_k = d_v = \frac{d}{H}.$$

Each MHSA layer consists of four projection matrices:

- Query matrix $W^Q \in \mathbb{R}^{d \times d}$,
- Key matrix $W^K \in \mathbb{R}^{d \times d}$,
- Value matrix $W^V \in \mathbb{R}^{d \times d}$,
- Output projection $W^O \in \mathbb{R}^{d \times d}$.

Thus, the attention block has $P_{\text{MHSA}}^{\text{standard}} = 4d^2$ parameters.

The MLP has two linear layers:

$$W_1 \in \mathbb{R}^{d \times d_{\text{ff}}}, \qquad W_2 \in \mathbb{R}^{d_{\text{ff}} \times d},$$

where typically $d_{\text{ff}} = 4d$. Hence: $P_{\text{MLP}}^{\text{standard}} = 2d\, d_{\text{ff}} = 8d^2$.

Therefore, a full Transformer layer has: $P_{\text{layer}}^{\text{standard}} = P_{\text{MHSA}}^{\text{standard}} + P_{\text{MLP}}^{\text{standard}} = 12d^2$.

**Hydra Ensembles.** We now consider Hydra Ensembles with $M = 3$ models. At each layer $\ell$, model $m$ keeps $H_\ell^{(m)} \leq H$ active heads. For simplicity, assume all models keep the same number of active heads $H_\ell$.

Each active head still projects into $d_k = d/H$. Therefore, the dimension of the concatenated head representation after pruning is

$$d_\ell = H_\ell d_k = \frac{H_\ell}{H}\, d.$$

Each MHSA layer of Hydra Ensembles consists of four projection matrices:

- Query matrix $W^Q \in \mathbb{R}^{d \times M d_\ell}$,
- Key matrix $W^K \in \mathbb{R}^{d \times M d_\ell}$,
- Value matrix $W^V \in \mathbb{R}^{d \times M d_\ell}$,
- Output projection $W^O \in \mathbb{R}^{M d_\ell \times d}$.

Thus, the attention block has $P_{\text{MHSA}}^{\text{P\&M}} = 4dM d_\ell$

*Multi-Layer Perceptron* In Hydra Ensembles, the Multi-Layer Perceptron parameters are shared across models (via averaging and grouping). Hence, the MLP cost remains:

$$P_{\text{MLP}}^{HydraEnsembles} = 8d^2.$$

*Total per layer.* The total parameter cost of Hydra Ensembles is:

$$P_{\text{layer}}^{\text{P\&M}} = P_{\text{MHSA}}^{\text{P\&M}} + P_{\text{MLP}}^{\text{P\&M}} = 4M\frac{H_\ell}{H}d^2 + 8d^2.$$

**Comparison.** Thus, the cost of Hydra Ensembles depends linearly on the pruning ratio $\frac{MH_\ell}{H}$. If pruning is strong (i.e. $H_\ell \ll H$), the Hydra Ensembles layer can have fewer parameters than a standard Transformer layer, even when aggregating three models.

In terms of memory cost, we have the following bound

$$P_{\text{layer}}^{\text{standard}} = 12d^2 \leq P_{\text{layer}}^{HydraEnsembles} = 4M\frac{H_\ell}{H}d^2 + 8d^2 \leq P_{\text{layer}}^{\text{DeepEns.}} = 12Md^2$$

**Inference Cost Analysis.** We compare all methods on ViT-B/16 and ImageNet-1k (Deng et al., 2009), keeping on average 8 heads per layer, under both `float32` and `bfloat16` precision. Table 7 reports runtime for the entire test set as well as per-batch inference time.

For `bfloat16`, we observe that Hydra Ensembles achieves an inference cost that is nearly identical to the single model baseline: the ratio Hydra Ensembles/Single is only $1.07\times$ for both whole-test runtime and per-batch inference. In contrast, Deep Ensembles are almost three times slower than a single model under the same setting ($\approx 2.99\times$).

For `float32`, Hydra Ensembles incurs a higher overhead compared to the single model ($\approx 2.66\times$), but still remains substantially faster than Deep Ensembles, which require roughly three times the inference cost of a single model ($\approx 3.02\times$). This increase in ratio for `float32` comes from the higher memory bandwidth and arithmetic cost of full-precision operations, which scale less favorably when multiple pruned members are executed jointly. In contrast, `bfloat16` computations benefit from specialized GPU hardware (Tensor Cores) combined with FlashAttention significantly accelerates matrix multiplications and reduces memory transfer, making Hydra Ensembles almost as efficient as a single model with no impact on performance.

Overall, these results highlight that our approach offers inference efficiency close to a single model in the `bfloat16` regime, while providing a favorable trade-off between performance and computational cost compared to traditional ensembles.

Table 7: **Comparison of inference cost between a single model, Deep Ensembles, and Hydra Ensembles.** We report runtime for a full test set and per-batch inference under BF16 and BF32, batch size 4, along with parameter count and multiply-add operations.

| Method | BFP16 | | FP32 | | Model Size | |
|---|---|---|---|---|---|---|
| | Full Test (s) | Per Batch (ms) | Full Test (s) | Per Batch (ms) | Params (M) | Mult-Adds (G) |
| SINGLE MODEL | 23.07 | 6.15 | 31.27 | 8.34 | 86.57 | 17.58 |
| TAYLOR | 13.18 | 3.51 | 28.02 | 7.47 | 77.13 | 15.48 |
| CIRCAVG | 13.20 | 3.52 | 28.47 | 7.59 | 77.13 | 15.48 |
| DEEP ENSEMBLES | 69.06 | 18.42 | 94.36 | 25.16 | 259.7 | 52.74 |
| HYDRA ENSEMBLES | 24.55 | 6.55 | 82.37 | 21.97 | 116.31 | 46.19 |

Table 8: Effect of keeping different diversity sources. v means the factor is varied across members, x means it is kept fixed.

| Variant | Diversity kept | | | Classification | | | Calibration | | OOD Average | | |
|---|---|---|---|---|---|---|---|---|---|---|---|
| | Pruning Seed | Batch | Backprop | ACC↑ | Brier↓ | NLL↓ | ECE↓ | aECE↓ | AUROC↑ | FPR95↓ | AUPR↑ |
| SINGLE | – | – | – | 80.67 | **0.27** | **0.71** | **0.01** | **0.01** | 84.40 | 50.25 | 60.91 |
| CIRCAVG | – | – | – | 80.11 | 0.30 | 0.88 | 0.10 | 0.10 | 85.43 | 50.79 | 61.60 |
| HYDRA-ENS (CIRC) | ✓ | x | x | **80.80** | 0.29 | 0.86 | 0.12 | 0.12 | 86.02 | **47.94** | 62.33 |
| HYDRA-ENS (CIRC) | x | ✓ | x | 80.13 | 0.30 | 0.88 | 0.10 | 0.10 | 85.48 | 46.93 | 62.76 |
| HYDRA-ENS (CIRC) | x | x | ✓ | 80.06 | 0.30 | 0.88 | 0.10 | 0.10 | 85.34 | 50.55 | 61.28 |
| HYDRA-ENS (CIRC) | x | ✓ | x | 80.11 | 0.30 | 0.88 | 0.10 | 0.10 | 85.38 | 50.49 | 61.28 |
| HYDRA-ENS (CIRC) | ✓ | ✓ | ✓ | 80.78 | 0.29 | 0.86 | 0.12 | 0.12 | **86.07** | 48.26 | **62.53** |
| HYDRA-ENS (CIRC) | x | x | x | 80.12 | 0.30 | 0.88 | 0.10 | 0.10 | 85.31 | 50.67 | 61.27 |

## B.5 STUDYING THE SOURCES OF DIVERSITY

To understand which factors contribute most significantly to the diversity and robustness of Hydra Ensembles, we systematically ablate different sources of randomness during pruning. In this case, we assume only one circuit to be available (*CircAvg*) and to build Hydra Ensembles we sample 48 heads to prune three times out of 100. Table 8 summarizes the impact of varying the pruning seed, batch order, and backpropagation stochasticity on both in-distribution and out-of-distribution performance. The results indicate that accuracy improves significantly only when the pruning seed is varied. Introducing *pruning-seed* diversity yields the most consistent OOD gains: AUROC increases to approximately 86 (from ∼85.3), and FPR95 decreases to the high 40s (from ∼50.7). *Batch-order* variability primarily reduces FPR95 (with a modest improvement in AUPR) but produces smaller and less consistent AUROC gains, whereas *backpropagation* stochasticity alone has limited impact. Combining these factors does not systematically surpass the seed-driven setting. We therefore conclude that the primary source of beneficial diversity is the *pruning seed*.

In addition to accuracy and OOD robustness, we directly measure ensemble diversity using mutual information (MI) and disagreement index (DI) on both ID and OOD data (Table 9). These metrics confirm the same conclusion: varying the *pruning seed* is the only factor that reliably creates non-trivial diversity (ID_MI/ID_DI ≈ 0.05/0.11 and OOD_MI/OOD_DI ≈ 0.12/0.48), reaching levels close to a standard Deep Ensembles. In contrast, changing only the batch order or only the backpropagation seed yields near-zero MI and DI, meaning the resulting members behave almost identically. Even when batch order and backpropagation randomness are combined, diversity remains negligible unless the pruning seed is also varied. This strengthens our conclusion that the beneficial diversity in Hydra Ensembles is fundamentally driven by the *pruning seed*, i.e., by the structural differences induced during pruning.

## B.6 STUDYING THE COST/BENEFIT OF HAVING MORE MEMBERS

To study how the number of members affects performance, we compare a Single model, a 3-member Deep Ensembles, and Hydra Ensembles with 3 and 5 members (Table 10). With 3 members, Hydra Ensembles keeps inference time close to the Single model (24.55 s vs. 23.07 s for the full test) while matching or slightly improving OOD performance over Deep Ensembles. Increasing Hydra

Table 9: Diversity ablations for Hydra Ensembles . v means the factor is varied across members, x means it is kept fixed.

| Method | Sources of diversity | | | Diversity metrics | | | |
|---|---|---|---|---|---|---|---|
| | pruning seed | batch order | backprop. | ID_MI ↑ | ID_DI ↑ | OOD_MI ↑ | OOD_DI ↑ |
| Single | – | – | – | – | – | – | – |
| CircAvg | – | – | – | – | – | – | – |
| Deep ensembles | – | – | – | **0.06** | **0.13** | **0.23** | **0.58** |
| Hydra Ensembles  (Circ) | v | x | x | 0.05 | 0.11 | 0.12 | 0.48 |
| Hydra Ensembles  (Circ) | x | v | x | 0.00 | 0.02 | 0.00 | 0.13 |
| Hydra Ensembles  (Circ) | x | x | v | 0.00 | 0.00 | 0.00 | 0.02 |
| Hydra Ensembles  (Circ) | x | v | v | 0.00 | 0.02 | 0.00 | 0.13 |
| Hydra Ensembles  (Circ) | v | v | v | 0.04 | 0.11 | 0.12 | 0.48 |
| Hydra Ensembles  (Circ) | x | x | x | 0.00 | 0.00 | 0.00 | 0.00 |

Table 10: Ablation on the number of members for Hydra Ensembles  (float16 inference)

| Method | Float 16 Speed | | Classification | | | Calibration | | OOD Average | | |
|---|---|---|---|---|---|---|---|---|---|---|
| | whole test (s) | 1 batch (ms) | ACC | Brier | NLL | ECE | aECE | AUROC | FPR95 | AUPR |
| Single | **23.07** | **6.15** | 80.67 | 0.27 | 0.71 | 0.01 | 0.01 | 84.40 | 50.25 | 60.91 |
| Deep Ensembles (3M) | 69.06 | 18.42 | **82.19** | **0.25** | **0.65** | **0.01** | **0.01** | 85.48 | **46.93** | 62.76 |
| Hydra Ensembles (3M) | 24.55 | 6.55 | 80.88 | 0.27 | 0.74 | **0.01** | **0.01** | 86.29 | 47.62 | 63.15 |
| Hydra Ensembles (5M) | 41.98 | 11.19 | 81.20 | 0.27 | 0.73 | **0.01** | **0.01** | 86.33 | 47.85 | **63.20** |

Ensembles  to 5 members further improves accuracy and OOD metrics at the cost of higher, but still substantially lower, latency than a 3-member Deep Ensembles.

### B.7 MLP FUSION AND DIVERSITY

We expand here on why fusing the MLP sub-blocks does *not* degrade ensemble diversity. In Hydra Ensembles , diversity mainly comes from the attention blocks being different across members: each Hydra Ensembles  member is trained separately as a pruned subnetwork with its own set of active heads. Because these attention structures are not the same, the members learn different internal features and make different predictions. We fuse only the MLP sub-blocks, and we do it *after* the separate training is finished, so this core source of difference between members stays intact.

This claim is supported empirically in Table 11. The fused and non-fused Hydra Ensembles  are essentially indistinguishable on both in-distribution and OOD criteria: they match calibration (ECE/aECE $= 0.01$), attain the same NLL ($0.74$), and yield nearly identical OOD detection (AU-ROC $86.29$ vs. $86.26$, FPR@95 $47.62$ vs. $47.98$, AUPR $63.15$ vs. $63.26$). If MLP fusion were reducing diversity, we would expect a noticeable degradation in calibration or uncertainty based OOD metrics due to increasingly correlated member outputs. Instead, OOD performance is preserved (and slightly improved relative to a non fused model), indicating that member disagreement and ensemble diversity remains intact despite MLP fusion.

## C DETAILS ON BASELINE IMPLEMENTATIONS

In these section we provide all details regarding baselines implementation and data used for each of the experimental sections: supervised image classification, zero-shot image classification and text classification.

### C.1 SUPERVISED IMAGE CLASSIFICATION

We adopt a two-stage training procedure for Vision Transformer (ViT) following (Dosovitskiy et al., 2020). Stage 1 pre-trains a ViT-B/16 on ImageNet-21k (Ridnik et al., 2021); Stage 2 fine-tunes on the target dataset (*ImageNet-1k* (Deng et al., 2009) or *CIFAR-100* (Krizhevsky, 2009)).

Table 11: Effect of MLP fusion on Hydra Ensembles . Fusing MLP sub-blocks preserves both in-distribution performance and OOD detection, indicating no loss of ensemble diversity.

| Method | Classification | | | Calibration | | OOD Average | | |
|---|---|---|---|---|---|---|---|---|
| | ACC ↑ | Brier ↓ | NLL ↓ | ECE ↓ | aECE ↓ | AUROC ↑ | FPR@95 ↓ | AUPR ↑ |
| Single | 80.67 | **0.27** | **0.71** | **0.01** | **0.01** | 84.40 | 50.25 | 60.91 |
| Hydra Ensembles (fused) | 80.88 | 0.27 | 0.74 | **0.01** | **0.01** | **86.29** | **47.62** | 63.15 |
| Hydra Ensembles (non-fused) | **81.00** | 0.27 | 0.74 | **0.01** | **0.01** | 86.26 | 47.98 | **63.26** |

### C.1.1 STAGE 1: PRE-TRAINING ON IMAGENET-21K

We train ViT-B/16 from scratch on ImageNet-21k (Ridnik et al., 2021) (Winter 2021; 13,153,500 images, 19,167 classes). Each input undergoes the following data augmentation:

- Random resized crop to $224 \times 224$ with scale sampled uniformly from $[0.08, 1.0]$,
- Random horizontal flip with probability 0.5,
- Conversion to tensor,
- Channel-wise normalization.

We use AdamW (Loshchilov & Hutter, 2017) as an optimizer with the following parameters

$$\eta_{\max} = 10^{-3}, \quad \text{dropout} = 0.1, \quad \lambda = 0.03, \quad \beta = (0.9, \ 0.999).$$

The learning rate follows a linear warm-up for the first $10,000$ steps, then decays linearly to zero. Pre-training runs for 90 epochs.

### C.1.2 STAGE 2: FINE-TUNING ON IMAGENET-1K

We load the pre-trained weights resulting from stage 1, and reinitialize the classifier part to $N{=}1000$ classes. During the training, we reuse the pre-training data augmentation pipeline. For evaluation, images are resized to $256{\times}256$, center-cropped to $224{\times}224$, and normalized. The official validation set is partitioned into a small validation subset (1%) and a larger held-out test subset (99%) to monitor convergence.

Fine-tuning uses SGD (momentum 0.9), with no weight decay and no dropout. The best learning rate is selected from $\{0.003, 0.01, 0.03, 0.06\}$. We apply a linear warm-up over $500$ steps, followed by a cosine decay over $20,000$ steps, and stop at validation convergence or when the step budget is reached.

### C.1.3 FINE-TUNING ON CIFAR-100

We load the pre-trained weights on ImageNet-21k (Ridnik et al., 2021) and reinitialize the classifier to $N{=}100$ classes. The training split reuses the pre-training data augmentation pipeline and test images are resized to $224 \times 224$ and normalized ; we use the official train/val/test splits.

Fine-tuning again uses SGD (momentum 0.9), no weight decay, and no dropout. The best learning rate is chosen from $\{0.001, 0.003, 0.01, 0.03\}$, with a linear warm-up over $500$ steps and cosine decay over $10,000$ steps, stopping on validation convergence or at the step budget.

### C.1.4 PACKED ENSEMBLES, BATCH ENSEMBLE, MIMO AND LORA ENSEMBLE

For both Packed-Ensembles (Laurent et al., 2023) and MIMO (Havasi et al., 2020), we use the same two-stage protocol as above: train each baseline from scratch on ImageNet-21k (Ridnik et al., 2021), then fine-tune on the target dataset (ImageNet-1k (Russakovsky et al., 2015) or CIFAR-100 (Krizhevsky, 2009)). For MIMO (Havasi et al., 2020), we set the number of estimators to $E{=}3$, use $\rho{=}0.5$, and `batch_repeat= 4`. For packed ensembles, we use $E{=}3$ and $\alpha{=}2$. For Batch Ensembles Wen et al. (2020), we apply it to the trained checkpoint on ImageNet-21k (Ridnik et al., 2021) due to high cost of this pretraining step, we later fine-tune this model on either ImageNet-1k (Russakovsky et al., 2015) or CIFAR-100 (Krizhevsky, 2009) . For Lora Ensemle (Mühlematter et al., 2024) (Wang et al., 2023) we attach lora modules to the attention mechanism and fine-tune the model 3 different times with different seeds, we use $r{=}4$ and $alpha{=}8$.

### C.1.5 IMAGE OOD EVALUATION DETAILS

For out-of-distribution (OOD) evaluation, we follow the **OpenOOD** benchmark (Yang et al., 2022) splits for both datasets, which define Near-OOD and Far-OOD scenarios. For ImageNet-1K, the OOD datasets include SSB (Vaze et al., 2021) , OpenImage-O (Wang et al., 2022), Ninco (Bitterwolf et al., 2023), iNaturalist (Huang & Li, 2021) and Texture (Kylberg, 2011); for CIFAR-100, they are CIFAR-10, TinyImageNet (Torralba et al., 2008), Texture, MNIST (Lecun et al., 2002), SVHN (Netzer et al., 2011) and Places365 (Zhou et al., 2017). We note that we exclude OpenImage-O for ImageNet-1K and TinyImageNet for CIFAR-100 to ensure an evaluation that is as fair as possible, since their validation sets are used for circuit extraction.

## C.2 SUPERVISED TEXT CLASSIFICATION

We fine-tune a `bert-base-uncased` (Devlin et al., 2019) classifier initialized from the HuggingFace checkpoint on SST-2 (Socher et al., 2013). Since there is no official test set, we use the validation set for testing, while setting aside part of the training set for validation.

**Tokenization** We use the `bert-base-uncased` tokenizer with `max_length`=128, truncation, and padding to max length. A deterministic split is applied: the first 3,000 rows of the GLUE (Wang et al., 2018) train split serve as validation; the remainder forms the training set. Evaluation is reported on the official GLUE validation split (872 labeled examples).

**Optimizer and schedule.** We use AdamW as optimizer with decoupled weight decay (weight_decay = 0.01) and exclude bias and LayerNorm weights from decay. The learning rate is $\eta = 8 \times 10^{-6}$. The schedule is *per step*: linear warm-up over $10\%$ of the total training steps followed by linear decay to zero. Gradient clipping is applied with $\ell_2 = 1.0$.

**Early stopping and checkpoints.** Training runs for up to 7 epochs with early stopping on validation accuracy (patience $= 2$, $\Delta = 5 \times 10^{-4}$). We checkpoint every epoch and select the model with the best validation accuracy; final numbers are reported on the held-out test split.

**OOD evaluation.** We consider two out-of-distribution (OOD) settings: (i) *Near-OOD*, where the task is still sentiment analysis but the data comes from domains other than movie reviews. (ii) *Far-OOD*, where the task is different from sentiment analysis, following recent NLP OOD protocols Liu et al. (2023)Kim et al. (2023).

## C.3 ZERO SHOT IMAGES CLASSIFICATION

**Datasets.** The complete list of datasets used for evaluation is the following: ImageNet-1K, CIFAR-100 and CIFAR-10, Food101 (Bossard et al., 2014), SUN397 (Xiao et al., 2010), Oxford Pet (Parkhi et al., 2012), DTD (Cimpoi et al., 2014), EuroSAT (Helber et al., 2018) and Caltech101 (Fei-Fei et al., 2004). For ImageNet-1K and CIFAR-100 we adopt the same evaluation splits as in supervised image classification, and the same holds for out-of-distribution benchmarks. For SUN397 we use HuggingFace while we use `torchvision` for the other ones. These datasets cover a broad spectrum, ranging from large-scale benchmarks such as ImageNet-1K to smaller fine-grained recognition tasks like Oxford Pet and Food101. They also include diverse domains such as textures (DTD) and satellite imagery (EuroSAT), ensuring a comprehensive evaluation across different levels of difficulty and granularity, just like in standard prior CLIP-based studies (Zhou et al., 2022). Following Radford et al. (2021), we use `"A photo of a {label}`. as prompt template.

**Architecture.** For all our experiments we use the backbone CLIP-ViT/B-32 model pretrained on LAION2B (Schuhmann et al., 2022). The specific model instance is `laion2b_s34b_b79k` which is made readily available along with the train and validation transform by the OpenCLIP repository (Ilharco et al., 2021).

**Baselines Implementation.** Taylor pruning (Molchanov et al., 2019) is applied only on the vision encoder removing 24 heads for simplicity. We do so because the scale of the importance scores between the two encoders differs greatly as showed in Figure 7, thus making them not directly comparable. BayesVLM (Baumann et al., 2024), a training-free method that improves uncertainty estimation using a Laplace approximation over 327k samples of LAION-400M and captures uncertainties inherent to the model itself. For its evaluation we leverage the public GitHub repository of

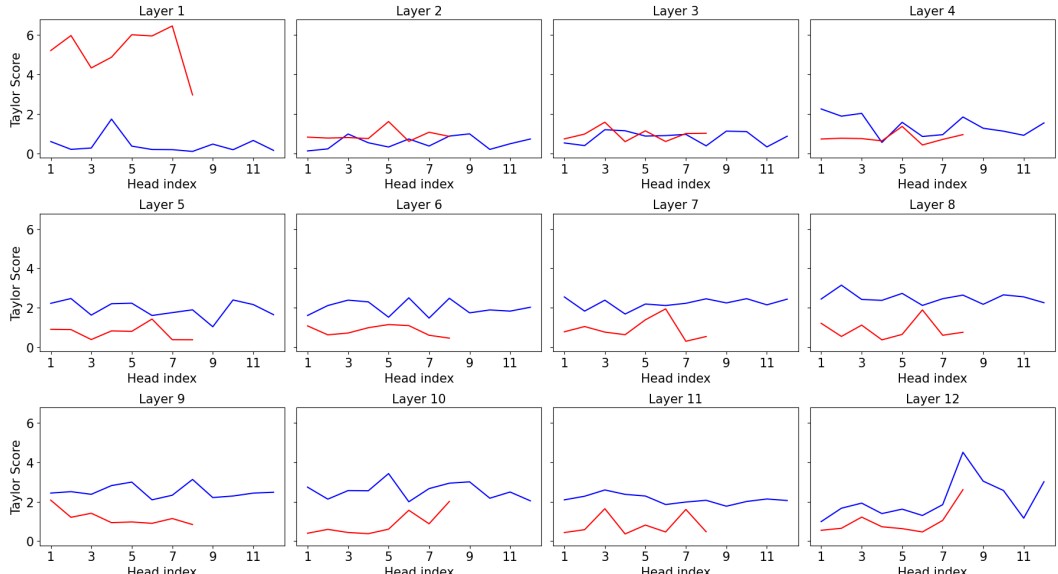

Figure 7: Taylor scores for text (8x12 heads) and vision (12x12 heads) encoders ordered by layer and head; in all layers, but the first three, the vision encoder has consistently higher scores.

the project where they also make available the hessian estimation. ViLU Lafon et al. (2025) instead adopts a very different approach by introducing additional parameters to train a binary misclassification classifier to distinguish correct from incorrect predictions. The classifier can be used for OOD detection by considering OOD samples as misclassified ones. It leverages frozen text and vision features by creating an embedding composed of reweighted text embeddings for each class, the vision embedding, and the textual embedding of the predicted class. Unlike our method, ViLU requires training on each dataset using a weighted binary cross-entropy loss. Implementation follows the original GitHub repository settings, and we train on each dataset for a number of epochs that range between 100 and 300, which for ImageNet-1k requires approximately 10 hours on 8 A100 GPUs. For Temperature Scaling (Guo et al., 2017) we use 5000 samples from the training set of ImageNet-1k following the setup used in BayesVLM. Lastly note that we don't evaluate BayesVLM and ViLU with Temperature Scaling because they are not compatible with it.

# D DETAILS ON METHOD IMPLEMENTATIONS

## D.1 PRUNING TECHNIQUES

### D.1.1 TAYLOR PRUNING

Following Molchanov et al. (2019), for each Transformer block $l$, we score every attention head $h$ using a first–order Taylor criterion computed on a small calibration loader. Let $D$ be the embedding dimension, $H$ the number of heads, and $d_h = D/H$. Let $W_q^{(l)}, W_k^{(l)}, W_v^{(l)} \in \mathbb{R}^{D \times D}$ be the input-projection weights and $G_q^{(l)}, G_k^{(l)}, G_v^{(l)} \in \mathbb{R}^{D \times D}$ their gradients $\left\{ \frac{\partial \mathcal{L}}{\partial W_q^{(l)}}, \frac{\partial \mathcal{L}}{\partial W_k^{(l)}}, \frac{\partial \mathcal{L}}{\partial W_v^{(l)}} \right\}$. For head $h \in \{1, \dots, H\}$, we define its row index set $\mathcal{R}_h = \{(h-1)d_h + 1, \dots, hd_h\}$. The final scores per

layer is computed as follows :

$$q_h^{(l)} = \frac{1}{d_h D} \sum_{r \in \mathcal{R}_h} \sum_{c=1}^{D} \left| W_{q,rc}^{(l)} G_{q,rc}^{(l)} \right|,$$

$$k_h^{(l)} = \frac{1}{d_h D} \sum_{r \in \mathcal{R}_h} \sum_{c=1}^{D} \left| W_{k,rc}^{(l)} G_{k,rc}^{(l)} \right|,$$

$$v_h^{(l)} = \frac{1}{d_h D} \sum_{r \in \mathcal{R}_h} \sum_{c=1}^{D} \left| W_{v,rc}^{(l)} G_{v,rc}^{(l)} \right|,$$

$$s_h^{(l)} = \frac{1}{3} \left( q_h^{(l)} + k_h^{(l)} + v_h^{(l)} \right).$$

Given a budget $r_l$ heads to prune per layer, we keep the top $H_l - r_l$ heads by $s_{l,h}$ among the layer heads and **structurally** rebuild the attention as a `MultiHeadAttentionPruned` module by slicing $W_q, W_k, W_v$ (and $W^O$) and their biases. After pruning a layer, gradients are recomputed again, ensuring scores reflect the updated network. This prune–recompute step is applied sequentially across layers until the pruning budget is met across all layers.

### D.1.2 CIRCUIT EXTRACTION AND PRUNING

Inspired by Wang et al. (2025), we prune the attention heads of a transformer *without* relying on gradients (as in Taylor-based approaches). Instead, we score each head by its contribution to a target task (accuracy or OOD detection), which gives a ranking from most to least useful head. Given a head budget, we then *structurally* prune the lowest-ranked heads, leaving a small set that preserves the desired behavior of the model. We consider three behaviors throughout: (i) in-distribution (ID) accuracy, (ii) out-of-distribution (OOD) separability under MSP, and (iii) a balance of both. For each target behavior we define a validation score to optimize:

$$S_{\text{acc}} = \text{Acc}(\tilde{f}; \mathcal{D}_{\text{ID}}), \qquad S_{\text{ood}} = \text{AUROC}_{\text{MSP}}(\tilde{f}; \mathcal{D}_{\text{ID}}, \mathcal{D}_{\text{OOD}}), \qquad S_{\text{avg}} = \tfrac{1}{2} \left( S_{\text{acc}} + S_{\text{ood}} \right),$$

where $\tilde{f}$ denotes the model with a proposed head temporarily turned off (i.e., ablated). Here, $S_{\text{acc}}$ targets ID accuracy, $S_{\text{ood}}$ targets OOD separability under MSP, and $S_{\text{avg}}$ trades off both. Depending on the criterion used for circuit extraction, we denote the final model obtained with $S_{acc}$ as *CircAcc*, with $S_{ood}$ as *CircOOD*, and with $S_{avg}$ as *CircAvg*. With $S$ specified, we describe below how heads are selected and removed.

**Greedy circuit extraction under a head budget.** Given a head budget $B$, extraction proceeds by turning heads off—i.e., ablating them—*individually* in a greedy loop and measuring the impact on $S$:

1. Initialize the candidate set $\mathcal{R} = \{(l, h)$ for all layers and heads$\}$.
2. For each $(l, h) \in \mathcal{R}$: *temporarily* turn that head off (ablate it), evaluate the model to compute $S$, then restore the head. Record $S$ for that particular $(l, h)$.
3. Identify the single best candidate $(l^\star, h^\star)$—the one whose removal achieves the highest $S$ (i.e., currently the least useful head for the chosen objective)—and *permanently* turn that head off (ablate it). Remove $(l^\star, h^\star)$ from $\mathcal{R}$.
4. Repeat steps 2–3 until $B$ heads have been removed.

This procedure yields a ranking of attention heads across the whole model by their relevance to our chosen objective, this ranking can then be used to create our pruned subnetwork—our extracted *circuit*—optimized for the chosen score $S$ (accuracy, OOD, or their average). Because each step measures the marginal effect of removing exactly one head, the process is stable and easy to implement across frameworks. Please refer to 1 for the full algorithm.

**Details on head scoring and ablation.** At block $l$, let the post–layer-norm input be $\widehat{X}_l \in \mathbb{R}^{T \times D}$. For head $h \in \{0, \dots, H_l - 1\}$ we form

$$Q_l^{(h)} = \widehat{X}_l W_l^{Q(h)}, \qquad K_l^{(h)} = \widehat{X}_l W_l^{K(h)}, \qquad V_l^{(h)} = \widehat{X}_l W_l^{V(h)}.$$

compute attention weights

$$A_l^{(h)} = \text{softmax}\left( \tfrac{1}{\sqrt{d_h}} Q_l^{(h)}(K_l^{(h)})^\top \right) \in \mathbb{R}^{T \times T},$$

and aggregate values

$$Z_l^{(h)} = A_l^{(h)} V_l^{(h)} \in \mathbb{R}^{T \times d_h}.$$

All head outputs are concatenated and passed through the shared output projection:

$$Z_l = \big[\, Z_l^{(0)} \,\|\, \cdots \,\|\, Z_l^{(H_l - 1)} \big] \in \mathbb{R}^{T \times D}, \qquad \text{MHA}_l(X_l) = Z_l W_l^O + \boldsymbol{b}_l^O,$$

with $W_l^O \in \mathbb{R}^{D \times D}$. Because concatenation assigns each head a dedicated channel block, head $h$ corresponds to a *contiguous* slice of $d_h$ columns in $W_l^O$ so to "turn a head off" we just need to change the output projection $W_l^O$, by partitioning $W_l^O$ into per-head column blocks $W_{l,(h)}^O \in \mathbb{R}^{D \times d_h}$:

$$W_l^O = \big[\, W_{l,(0)}^O \,\|\, \cdots \,\|\, W_{l,(h)}^O \,\|\, \cdots \,\|\, W_{l,(H_l - 1)}^O \,\big].$$

The attention output decomposes as

$$Z_l W_l^O = \sum_{j=0}^{H_l - 1} Z_l^{(j)} W_{l,(j)}^O.$$

Setting the specific block $W_{l,(h)}^O = \mathbf{0}$ removes exactly the contribution of a particular head $h$, since the term $Z_l^{(h)} W_{l,(h)}^O$ vanishes while all other head contributions are unchanged. Tensor shapes are preserved: $Z_l$ remains $T \times D$, the projection is still $D \to D$, and the residual update $Y_l = X_l + \text{MHA}_l(X_l)$ is well-defined. Ablating via the $W^O$ slice is reversible and numerically stable: it turns heads off one at a time without modifying the $Q/K/V$ projections or fused attention kernels, which makes it easy to implement across different architectures.

**Ablation operator (non-structural).** Let $s_h = h\, d_h$ and $e_h = (h+1)\, d_h$ be the column range in $W_l^O$ corresponding to head $h$.

**Temporary ablation** (for scoring only):

$$\begin{aligned}
\textsc{AblateTemp}(l, h) : \Delta_{l,h} &\leftarrow W_l^O[:,\, s_h : e_h]\,; \\
W_l^O[:,\, s_h : e_h] &\leftarrow \mathbf{0}; \\
\text{evaluate score } S&; \\
W_l^O[:,\, s_h : e_h] &\leftarrow \Delta_{l,h}.
\end{aligned}$$

**Permanent ablation** (to define the circuit):

$$\textsc{AblatePerm}(l, h) : \; W_l^O[:,\, s_h : e_h] \leftarrow \mathbf{0} \quad \text{(no restore)}.$$

These ablation operators are *non-structural* in the sense that parameter tensors keep their original sizes (no FLOP reduction). This choice makes comparisons across heads and across scores consistent and easily reversible.

Given a budget of $r$ heads to prune, we remove the first $r$ heads given by the circuit algorithm and structurally rebuild the attention as a `MultiHeadAttentionPruned` module by keeping only the selected head blocks and slicing the per-head columns/rows of $W^Q$, $W^K$, $W^V$, and $W^O$ (and their biases).

**OOD sets and fairness.** For circuit extraction on ImageNet-1k we use *OpenImage-O* as OOD; for CIFAR-100 circuits, we use *Tiny-ImageNet*; following OOD validation subsets set by *OpenOOD*. Although these OOD validation subsets splits differ from the ones used during the OpenOOD benchmark we exclude them from our final comparison tables for more fairness; they are used only to define the pruning score in the OOD/average circuits. For SST2 dataset we use the AG's News dataset (Zhang et al., 2015) as OOD validation and we do not include it in our final results.

---

**Algorithm 1** Greedy circuit extraction under budget $B$

---

**Require:** model $f$; ID loader $\mathcal{D}_{\text{ID}}$; OOD loader $\mathcal{D}_{\text{OOD}}$; score type $s \in \{\text{acc}, \text{ood}, \text{avg}\}$; budget $B$
1: $\mathcal{R} \leftarrow \{(l, h)$ for all layers/heads$\}$
2: **for** $t \leftarrow 1$ **to** $B$ **do**
3:     best $\leftarrow -\infty$, $(l^\star, h^\star) \leftarrow$ None
4:     **for all** $(l, h) \in \mathcal{R}$ **do**
5:         ABLATETEMP$(l, h)$
6:         $(\text{acc}, \text{auroc}) \leftarrow$ EVALUATE$(f; \mathcal{D}_{\text{ID}}, \mathcal{D}_{\text{OOD}})$
7:         $S \leftarrow \begin{cases} \text{acc}, & s = \text{acc} \\ \text{auroc}, & s = \text{ood} \\ \frac{1}{2}(\text{acc} + \text{auroc}), & s = \text{avg} \end{cases}$
8:         RESTORE$(l, h)$
9:         **if** $S >$ best **then** best $\leftarrow S$; $(l^\star, h^\star) \leftarrow (l, h)$
10:     ABLATEPERM$(l^\star, h^\star)$; $\mathcal{R} \leftarrow \mathcal{R} \setminus \{(l^\star, h^\star)\}$

---

### D.2 POST-PRUNING FINE-TUNING PROTOCOL.

For each dataset, we fine-tune the pruned transformer using the same recipe, independent of the pruning technique. We freeze most weights, unfreeze a targeted subset (attention modules, MLPs, LayerNorms, and the classifier head), and train with SGD using a per-step warm-up followed by cosine decay to a learning-rate floor. We report validation metrics during training and evaluate on the held-out test split at the end.

#### IMAGENET-1K: FINE-TUNING AFTER PRUNING

**Learning rate.** We select $\eta_{\max}$ via a small grid $\{0.003, 0.01, 0.03, 0.06\}$ (best by validation) and use a *per-step* schedule: linear warm-up for $5{,}000$ steps followed by cosine decay to a floor fraction $f_{\min} = 10^{-5}$; the total number of steps is $120{,}000$.

**Train transforms.** RandomResizedCrop to $224 \times 224$, RandomHorizontalFlip with probability $0.5$, RandAugment (2 ops, magnitude 12) and ImageNet mean/std normalization.

**Regularisation.** Cross-entropy with label smoothing $\epsilon = 0.1$, and dropout $p = 0.1$ inside each transformer layer.

#### CIFAR-100: FINE-TUNING AFTER PRUNING

**Learning rate.** We select $\eta_{\max}$ via a small grid (best by validation; e.g., $\{0.001, 0.003, 0.01, 0.03\}$) and use a *per-step* schedule: linear warm-up for $500$ steps followed by cosine decay to a floor fraction $f_{\min} = 10^{-5}$; the total number of steps is $10{,}000$.

**Train transforms.** RandomResizedCrop to $224 \times 224$, RandomHorizontalFlip with probability $0.5$, and ImageNet mean/std normalization.

**Regularisation.** Cross-entropy with label smoothing $\epsilon = 0.1$.

**Specific details for CIFAR-100.** (1) Partial fine-tuning: only the *last five* Transformer blocks are updated (attention projections, LayerNorms, MLP linear layers); earlier blocks remain frozen. (2) The classifier head is re-initialized to zeros before fine-tuning. Following the observation of Raghu et al. (2021), we find that fine-tuning all layers on small datasets such as CIFAR-100 tends to degrade performance, as the model quickly overfits and loses useful pretrained representations. To mitigate this issue, we fine-tune only the final classification layer when working with CIFAR-100.

#### GLUE/SST-2: FINE-TUNING BERT AFTER PRUNING

**Learning rate.** We set $\eta = 1\text{e}{-5}$ and use a *per-step* linear warm-up over $10\%$ of the total steps followed by linear decay to zero. Optimization is AdamW with decoupled weight decay; gradient clipping $\ell_2 = 1.0$.

Table 12: Results of Hydra Ensembles on ImageNet-1K and CIFAR-100 across 8 seeds

| Dataset | Method | Classification | | | Calibration | | OOD Average | | |
|---|---|---|---|---|---|---|---|---|---|
| | | Acc (↑) | Brier (↓) | NLL (↓) | ECE (↓) | aECE (↓) | AUROC (↑) | FPR95 (↓) | AUPR (↑) |
| **ImageNet-1K** | SINGLE | 80.67 | 0.27 | 0.71 | 0.01 | 0.01 | 84.40 | 50.25 | 60.91 |
| | DEEP ENSEMBLES | 82.19 | 0.25 | 0.65 | 0.01 | 0.01 | 85.48 | 46.93 | 62.76 |
| | HYDRA ENS (CIRC) | $80.88_{\pm0.04}$ | $0.27_{\pm0.00}$ | $0.74_{\pm0.00}$ | $0.01_{\pm0.00}$ | $0.01_{\pm0.00}$ | $86.31_{\pm0.01}$ | $47.62_{\pm0.01}$ | $63.18_{\pm0.05}$ |
| **CIFAR-100** | SINGLE | 92.15 | 0.11 | 0.25 | 0.01 | 0.01 | 85.46 | 40.27 | 93.55 |
| | DEEP ENSEMBLES | 93.52 | 0.09 | 0.22 | 0.01 | 0.01 | 86.08 | 38.67 | 94.26 |
| | HYDRA ENS (CIRC) | $92.05_{\pm0.04}$ | $0.12_{\pm0.01}$ | $0.28_{\pm0.00}$ | $0.07_{\pm0.00}$ | $0.08_{\pm0.00}$ | $89.72_{\pm0.23}$ | $35.78_{\pm0.57}$ | $95.25_{\pm0.06}$ |

**Train details.** `bert-base-uncased` tokenizer with `max_length`=128, truncation, and padding to max length. Deterministic split: the first $K$ examples of GLUE/SST-2 `train` form validation ($K$=3000); the official `validation` set is used for test.

**Regularisation.** Cross-entropy (no label smoothing); default BERT dropout is left unchanged.

### D.3  ZERO SHOT IMAGES CLASSIFICATION

For CLIP-based methods, we prune 24 attention heads jointly from both encoders. When pruning both encoders, we rely exclusively on ImageNet-1k (Deng et al., 2009) and its associated OOD sets. The datasets are the same as those used in Section 5.1, ensuring a fair comparison.

**Train details.** When training CLIP, several design choices are possible. Following Zhai et al. (2022), we lock the image encoder during training. This strategy provides three key benefits: (i) it improves performance compared to finetuning the vision encoder, (ii) it eliminates the need to merge MLPs across circuits, since the pruned vision encoders remain identical and differ only in their attention heads, and (iii) it simplifies training, as the text encoder contains less than half the parameters of the vision encoder. Training is carried out with the OpenCLIP repository (Ilharco et al., 2021). We train for one epoch on 60M samples from LAION-400M (Schuhmann et al., 2021), using a learning rate equal $5\mathrm{e}-5$ selected from a small grid $\{1\mathrm{e}-4, 5\mathrm{e}-5, 1\mathrm{e}-5\}$ on the train loss. We employ 100 warmup steps, a batch size of 1500, and 4 gradient accumulation steps (yielding a virtual batch size of 96,000). All other settings follow the OpenCLIP defaults.

**Hardware.** Training one circuit model requires approximately one hour, for a total of three hours to create the Hydra Ensembles, using 16 A100s GPUs.

## E  COMPLEMENTARY RESULTS

### E.1  SUPERVISED IMAGE CLASSIFICATION

We report additional OOD results for ImageNet-1K and CIFAR-100 in Table 15 and Table 16. More detailed per-dataset results on ImageNet-1K are provided in Tables 21 and 22, and per-dataset OOD results for CIFAR-100 appear in Table 23. (Note that CIFAR-10 is the only near-OOD split for CIFAR-100, so its performance matches the average near-OOD result already shown in Table 16.)

Across both ImageNet-1K and CIFAR-100, Hydra Ensembles exhibits stable behavior over eight different seeds, see Table 12. The mean performance varies only minimally across runs, and the associated standard deviations remain small for all metrics (accuracy, Brier, NLL, calibration, and OOD scores). This indicates that Hydra Ensembles deliver statistically consistent results with no sensitivity to initialization or seed choice while maintaining competitive accuracy and strong OOD robustness.

We also evaluate all ImageNet-1K baselines under distribution shift using corrupted ImageNet-C Hendrycks & Dietterich (2019b) inputs at five severity levels. Table 13 reports top-1 accuracy and Table 14 reports the corresponding Brier scores (lower is better).Deep Ensembles obtain the best average accuracy across severities (57.12%), but Hydra Ensembles (Taylor) remain highly compet-

Table 13: Results under distribution shift (ImageNet-C): top-1 accuracy (%). Columns `sev1-sev5` correspond to corruption severities 1–5; `Avg` is the mean over severities.

| Method | Acc_sev1 | Acc_sev2 | Acc_sev3 | Acc_sev4 | Acc_sev5 | Average |
|---|---|---|---|---|---|---|
| Single | 70.44 | 63.31 | 56.57 | 45.10 | 31.97 | 53.47 |
| Deep Ensembles | **73.11** | **66.59** | **60.43** | 49.50 | 35.97 | **57.12** |
| Packed Ensembles | 68.11 | 60.18 | 52.87 | 41.21 | 28.81 | 50.23 |
| MIMO | 71.49 | 64.88 | 58.78 | 47.56 | 34.40 | 55.42 |
| Batch Ensemble | 70.10 | 62.94 | 56.21 | 44.71 | 31.72 | 53.13 |
| MC Dropout | 70.44 | 63.64 | 57.39 | 46.38 | 33.22 | 54.21 |
| LoRA Ensembles | 70.27 | 63.09 | 56.34 | 44.88 | 31.79 | 53.27 |
| OBA | 69.12 | 62.49 | 56.40 | 47.06 | 35.24 | 54.06 |
| Taylor | 70.63 | 63.81 | 57.84 | 47.84 | 34.92 | 55.00 |
| CircAvg | 70.00 | 62.97 | 56.52 | 45.81 | 32.66 | 53.59 |
| Hydra Ens (Taylor) | 71.65 | 65.10 | 59.35 | **49.62** | **36.79** | 56.50 |
| Hydra Ens (Circ) | 70.81 | 63.87 | 57.62 | 46.96 | 33.73 | 54.59 |

Table 14: Results under distribution shift (ImageNet-C): Brier score (lower is better). Columns `sev1-sev5` correspond to corruption severities 1–5; `Avg` is the mean over severities.

| Method | Brier_sev1 | Brier_sev2 | Brier_sev3 | Brier_sev4 | Brier_sev5 | Average |
|---|---|---|---|---|---|---|
| Single | 0.40 | 0.48 | 0.56 | 0.67 | 0.80 | 0.58 |
| Deep Ensembles | **0.38** | **0.45** | **0.52** | **0.64** | **0.76** | **0.55** |
| Packed Ensembles | 0.43 | 0.52 | 0.60 | 0.71 | 0.83 | 0.61 |
| MIMO | 0.39 | 0.47 | 0.54 | 0.65 | 0.78 | 0.56 |
| Batch Ensemble | 0.41 | 0.49 | 0.56 | 0.68 | 0.80 | 0.58 |
| MC Dropout | 0.40 | 0.48 | 0.55 | 0.66 | 0.79 | 0.57 |
| LoRA Ensembles | 0.40 | 0.49 | 0.56 | 0.68 | 0.80 | 0.58 |
| OBA | 0.43 | 0.50 | 0.57 | 0.67 | 0.79 | 0.59 |
| Taylor | 0.42 | 0.49 | 0.56 | 0.66 | 0.79 | 0.58 |
| CircAvg | 0.43 | 0.50 | 0.57 | 0.68 | 0.80 | 0.59 |
| Hydra Ens (Taylor) | 0.41 | 0.48 | 0.55 | **0.64** | **0.76** | 0.56 |
| Hydra Ens (Circ) | 0.42 | 0.50 | 0.57 | 0.67 | 0.79 | 0.59 |

itive with an average of 56.5% while clearly improving performance at the strongest shifts: Hydra Ensembles (Taylor) achieves the best accuracy at severity 4 and 5 (49.62% and 36.79%), outperforming Deep Ensembles in this regime. Hydra Ensembles (Taylor) also matches the calibration of Deep Ensembles, with very similar Brier scores and identical performance at the highest severities. Compared to alternative efficient ensemble methods (Packed Ensembles, MIMO, Batch Ensemble, MC Dropout, LoRA Ensembles, OBA, Taylor, CircAvg), Hydra Ensembles consistently achieve higher accuracy and equal or better Brier scores across severities, confirming that Hydra Ensembles 's structured diversity transfers to distribution-shift robustness on ImageNet-1K.

## E.2 SUPERVISED TEXT CLASSIFICATION

Table 17 present more results for OOD performance of Bert (Devlin et al., 2019) on SST2 dataset (Socher et al., 2013). Overall, *Hydra Ensembles (Circ)* yields the strongest OOD detection: best Near-OOD (AUROC 70.97, lowest FPR95 69.38) and best overall average (AUROC 77.60, lowest FPR95 55.06). *CircAvg* is the best single-pruned variant (overall AUROC 75.04, FPR95 56.22), while *DeepEns-Downstream* attains the top overall AUPR (84.90). See Table 18 for more detailed results on Near-OOD performance and tables 19 20 for Far-OOD results.

Table 15: ImageNet-1K (ViT-B/16): OOD detection on OpenOOD near/far splits (MSP).

| | | Near-OOD | | | Far-OOD Avg | | | OOD Avg (all OOD) | | |
|---|---|---|---|---|---|---|---|---|---|---|
| Method | Heads | AUROC↑ | FPR95↓ | AUPR↑ | AUROC↑ | FPR95↓ | AUPR↑ | AUROC↑ | FPR95↓ | AUPR↑ |
| SINGLE | 12 | 77.96 | 63.08 | 55.29 | 90.84 | 37.42 | 66.53 | 84.4 | 50.25 | 60.91 |
| DEEP ENSEMBLES | 12 | 78.77 | 61.70 | 56.15 | **92.20** | **32.17** | **69.38** | 85.48 | **46.93** | 62.76 |
| PACKED ENSEMBLE | 12 | 76.30 | 65.59 | 52.60 | 90.21 | 37.70 | 63.74 | 83.26 | 51.65 | 58.17 |
| MIMO | 12 | 78.06 | 62.85 | 55.67 | 89.21 | 42.44 | 62.61 | 83.63 | 52.64 | 59.14 |
| LoRA-SINGLE | 12 | 77.92 | 63.11 | 55.11 | 90.56 | 38.01 | 65.64 | 84.24 | 50.56 | 60.37 |
| BATCH ENSEMBLE | 12 | 77.93 | 63.20 | 55.08 | 90.54 | 37.99 | 65.61 | 84.24 | 50.59 | 60.35 |
| MC DROPOUT | 12 | 77.13 | 64.46 | 53.75 | 90.27 | 38.42 | 64.06 | 83.7 | 51.44 | 58.9 |
| OBA | 12 | 76.81 | 67.43 | 52.69 | 88.42 | 42.37 | 57.09 | 82.61 | 54.9 | 54.89 |
| TAYLOR | 8 | 79.83 | 65.10 | 59.22 | 88.93 | 43.92 | 59.71 | 84.38 | 54.51 | 59.46 |
| CIRCAVG | 8 | 81.00 | 62.00 | 61.33 | 90.43 | 38.46 | 63.49 | 85.71 | 50.23 | 62.41 |
| HYDRA-ENS (TAYLOR) | 8x3 | 80.53 | 62.31 | 59.85 | 90.18 | 38.70 | 61.64 | 85.36 | 50.5 | 60.75 |
| HYDRA-ENS (CIRC) | 8x3 | **81.39** | **60.10** | **61.45** | 91.19 | 35.14 | 64.85 | **86.29** | 47.62 | **63.15** |

Table 16: CIFAR-100 (ViT-B/16): OOD detection on OpenOOD near/far splits (MSP).

| | | Near-OOD (CIFAR-10) | | | Far-OOD Avg | | | OOD Avg (all OOD) | | |
|---|---|---|---|---|---|---|---|---|---|---|
| Method | Heads | AUROC↑ | FPR95↓ | AUPR↑ | AUROC↑ | FPR95↓ | AUPR↑ | AUROC↑ | FPR95↓ | AUPR↑ |
| SINGLE | 12 | 89.63 | 37.96 | 89.71 | 84.41 | 40.85 | 94.51 | 85.46 | 40.27 | 93.55 |
| DEEP ENSEMBLES | 12 | 90.88 | 34.73 | 91.09 | 84.88 | 39.65 | 95.06 | 86.08 | 38.67 | 94.26 |
| LoRA-SINGLE | 12 | 89.32 | 39.18 | 89.39 | 85.07 | 40.24 | 94.57 | 85.92 | 40.03 | 93.53 |
| LoRA-ENS | 12 | 89.47 | 38.56 | 89.52 | 84.85 | 40.58 | 94.53 | 85.77 | 40.18 | 93.53 |
| PACKED | 12 | 88.07 | 40.13 | 87.36 | 86.72 | 38.14 | 95.01 | 86.99 | 38.54 | 93.48 |
| MIMO | 12 | 91.58 | 34.67 | 92.24 | 87.21 | 37.58 | 95.87 | 88.08 | 37.00 | 95.15 |
| BATCH ENSEMBLES | 12 | 89.35 | 39.14 | 89.39 | 85.50 | 39.79 | 94.72 | 86.27 | 39.66 | 93.66 |
| MC DROPOUT | 12 | 88.88 | 41.03 | 88.97 | 83.71 | 42.91 | 94.16 | 84.74 | 42.53 | 93.12 |
| OBA | - | 88.86 | 40.53 | 88.72 | 85.10 | 40.47 | 94.58 | 85.85 | 40.48 | 93.41 |
| TAYLOR | 10 | 91.47 | 33.33 | 91.98 | 88.12 | 42.10 | 95.79 | 88.79 | 40.35 | 95.03 |
| CIRCAVG | 10 | 91.47 | 34.33 | 91.91 | **89.34** | 38.65 | 96.04 | **89.77** | 37.79 | 95.21 |
| HYDRA-ENS (TAYLOR) | 10x3 | **91.96** | 33.11 | **92.51** | 88.13 | 41.19 | **96.20** | 88.89 | 39.57 | **95.46** |
| HYDRA-ENS (CIRC) | 10x3 | 91.62 | **32.92** | 91.84 | 88.89 | **37.32** | 96.00 | 89.43 | **36.44** | 95.17 |

Table 17: SST-2 (BERT-base): OOD detection (near/far) with MSP.

| | | Near-OOD Avg | | | Far-OOD Avg | | | OOD Avg (all OOD) | | |
|---|---|---|---|---|---|---|---|---|---|---|
| Method | Heads | AUROC↑ | FPR95↓ | AUPR↑ | AUROC↑ | FPR95↓ | AUPR↑ | AUROC↑ | FPR95↓ | AUPR↑ |
| SINGLE | 12 | 59.45 | 88.13 | 99.08 | 74.45 | 63.62 | 75.07 | 70.16 | 70.62 | 81.93 |
| DEEPENS-DOWNSTREAM | 12 | 61.42 | 85.89 | 99.16 | 80.16 | 53.41 | **79.20** | 74.81 | 62.69 | **84.9** |
| MC DROPOUT | 12 | 62.31 | 87.21 | 99.19 | 76.20 | 59.42 | 75.07 | 72.23 | 67.36 | 81.96 |
| LoRA-SINGLE | 12 | 59.82 | 86.41 | 99.08 | 75.10 | 61.81 | 75.51 | 70.74 | 68.84 | 82.24 |
| LoRA-ENS | 12 | 59.83 | 86.12 | 99.08 | 75.23 | 61.74 | 75.55 | 70.83 | 68.7 | 82.27 |
| TAYLOR | 6 | 58.64 | 90.54 | 99.07 | 75.88 | 59.79 | 75.32 | 70.95 | 68.57 | 82.11 |
| CIRCAVG | 6 | 63.26 | 75.23 | 99.16 | 79.76 | **48.62** | 78.19 | 75.04 | 56.22 | 84.18 |
| HYDRA-ENS (TAYLOR) | 6x3 | 60.14 | 80.33 | 99.08 | 76.54 | 55.50 | 75.79 | 71.85 | 62.59 | 82.44 |
| HYDRA-ENS (CIRC) | 6x3 | **70.97** | **69.38** | **99.37** | **80.25** | 49.33 | 78.07 | **77.6** | **55.06** | 84.16 |

Table 18: SST-2 (BERT-base): OOD detection on **near-OOD** splits (Yelp Polarity, Amazon Polarity) with MSP.

| | Yelp Polarity | | | Amazon Polarity | | | Near-OOD Avg | | |
|---|---|---|---|---|---|---|---|---|---|
| Method | AUROC↑ | FPR95↓ | AUPR↑ | AUROC↑ | FPR95↓ | AUPR↑ | AUROC↑ | FPR95↓ | AUPR↑ |
| SINGLE | 62.43 | 86.47 | 98.35 | 56.47 | 89.79 | 99.81 | 59.45 | 88.13 | 99.08 |
| DEEP ENSEMBLES | 63.82 | 84.17 | 98.49 | 59.03 | 87.61 | 99.83 | 61.42 | 85.89 | 99.16 |
| LoRA-SINGLE | 62.61 | 85.21 | 98.36 | 57.03 | 87.61 | 99.81 | 59.82 | 86.41 | 99.08 |
| LoRA-ENSEMBLE | 62.60 | 84.98 | 98.36 | 57.07 | 87.27 | 99.81 | 59.83 | 86.12 | 99.08 |
| MC DROPOUT | 65.04 | 86.12 | 98.55 | 59.58 | 88.30 | 99.83 | 62.31 | 87.21 | 99.19 |
| TAYLOR | 60.49 | 90.25 | 98.33 | 56.79 | 90.83 | 99.82 | 58.64 | 90.54 | 99.07 |
| CIRCAVG | 66.26 | 70.41 | 98.50 | 60.27 | 80.05 | 99.83 | 63.26 | 75.23 | 99.16 |
| Hydra Ensembles (Taylor) | 62.10 | 79.01 | 98.35 | 58.18 | 81.65 | 99.82 | 60.14 | 80.33 | 99.08 |
| Hydra Ensembles (Circ) | **73.84** | **65.60** | **98.87** | **68.10** | **73.17** | **99.87** | **70.97** | **69.38** | **99.37** |

Table 19: SST-2 (BERT-base): Far-OOD (**20NG**, **TREC**, **MNLI**) with MSP.

| Method | 20NG | | | TREC | | | MNLI | | |
|---|---|---|---|---|---|---|---|---|---|
| | AUROC↑ | FPR95↓ | AUPR↑ | AUROC↑ | FPR95↓ | AUPR↑ | AUROC↑ | FPR95↓ | AUPR↑ |
| SINGLE | 74.08 | 59.40 | 94.61 | 65.45 | 60.55 | 42.68 | 73.11 | 77.29 | 95.98 |
| DEEP ENSEMBLES | 80.08 | 46.56 | 95.98 | **78.21** | 40.37 | 56.48 | 77.28 | 72.02 | 96.70 |
| LORA-SINGLE | 74.75 | 58.37 | 94.70 | 67.77 | 57.00 | 44.40 | 73.45 | 75.57 | 96.04 |
| LORA-ENSEMBLE | 74.81 | 59.40 | 94.73 | 68.04 | 56.19 | 44.65 | 73.59 | 74.66 | 96.06 |
| MC DROPOUT | 78.68 | 50.11 | 95.47 | 64.57 | 57.57 | 41.38 | 76.14 | 73.85 | **96.46** |
| TAYLOR | 77.66 | 49.20 | 95.22 | 70.47 | 52.64 | 47.09 | 73.71 | 68.92 | 96.14 |
| CIRCAVG | 77.97 | 48.51 | 95.39 | 74.81 | 45.18 | 51.26 | 78.25 | 56.65 | 96.74 |
| Hydra Ensembles (Taylor) | 77.06 | 47.13 | 95.17 | 72.89 | 50.92 | 50.44 | 74.43 | 66.63 | 96.13 |
| Hydra Ensembles (Circ) | **83.69** | **37.50** | **96.51** | 71.99 | 55.16 | 49.69 | 78.07 | 61.58 | 96.69 |

Table 20: SST-2 (BERT-base): Far-OOD (**RTE**, **WMT16**) with MSP.

| Method | RTE | | | WMT16 | | |
|---|---|---|---|---|---|---|
| | AUROC↑ | FPR95↓ | AUPR↑ | AUROC↑ | FPR95↓ | AUPR↑ |
| SINGLE | 84.36 | 51.26 | 57.56 | 75.28 | 69.61 | 84.52 |
| DEEP ENSEMBLES | 86.72 | 45.18 | 60.45 | 78.53 | 62.96 | 86.39 |
| LORA-SINGLE | 84.35 | 50.57 | 58.18 | 75.22 | 67.55 | 84.25 |
| LORA-ENSEMBLE | 84.41 | 50.46 | 57.96 | 75.30 | 68.00 | 84.38 |
| MC DROPOUT | 86.07 | 44.15 | 57.87 | 75.54 | 71.44 | 84.20 |
| TAYLOR | 82.20 | 61.01 | 53.65 | 75.39 | 67.20 | 84.54 |
| CIRCAVG | 87.68 | 40.14 | 60.80 | **80.09** | **52.64** | **86.78** |
| Hydra Ensembles (Taylor) | 82.53 | 50.46 | 52.96 | 75.81 | 62.39 | 84.26 |
| Hydra Ensembles (Circ) | **88.38** | **32.91** | **61.20** | 79.13 | 59.52 | 86.30 |

## E.3 ZERO SHOT IMAGES CLASSIFICATION

In this section we present disaggregated results for both classification and OOD detection relative to Table 3 and the results with pruning applied only on the vision encoder in the zero-shot and fine-tuning setup.

### E.3.1 EXTENSIVE RESULTS

**Classification.** Results are reported in Table 24. Hydra Ensembles achieves accuracy comparable to, or better than, both BayesVLM and the single model, with the only exception of Food101, where it improves performance by a small but significant margin (+0.35%) despite being a training-free method. The gains are even more pronounced in terms of calibration: Hydra Ensembles reduces ECE by 5.19% relative to the single model, 2.59% relative to BayesVLM, and 0.54% compared to Temperature Scaling. In contrast, ViLU does not consistently improve calibration, and even if we disregard its calibration on ImageNet-1k (where it achieves the worst ECE across datasets), its average ECE remains worse than the one achieved by our method (4.17% vs. 3.29%). Hydra Ensembles also improves NLL compared to both BayesVLM and the single model, although it falls behind Temperature Scaling by a small margin. Nevertheless, unlike ViLU and BayesVLM, Hydra Ensembles is fully compatible with Temperature Scaling; when combined, the resulting NLL reaches 0.90, effectively closing the gap. Finally, CircuitAvg outperforms Taylor pruning across all metrics in a training-free setting, highlighting its robustness and practical advantage.

**OOD Detection.** Results are reported in Table 25 for ImageNet-1k and Table 26 for CIFAR100. On ImageNet-1k, Hydra Ensembles consistently outperform all baselines on both Near-OOD and Far-OOD detection. The closest competitor is ViLU, but out method improves upon it by notable margins in AUROC (+1.44%), FPR95 (-3.53%), and AUPR (+4.04%).

On CIFAR100 the picture is more mixed. For Near-OOD, Hydra Ensembles improves AUROC (+1.7%) and FPR95 (-5.17%), while ViLU and BayesVLM significantly underperform. On some Far-OOD datasets, such as Texture and MNIST, Hydra falls short of the single model, with degradation comparable to ViLU and BayesVLM, whereas it shows clear gains on SVHN and Places365. Overall, Hydra Ensembles achieves the best average performance among the training-free meth-

Table 21: ImageNet-1K (ViT-B/16): OOD detection (Near-OOD splits: SSB-Hard, NINCO) with MSP.

| Method | SSB-Hard | | | NINCO | | | Near-OOD Avg | | |
|---|---|---|---|---|---|---|---|---|---|
| | AUROC↑ | FPR95↓ | AUPR↑ | AUROC↑ | FPR95↓ | AUPR↑ | AUROC↑ | FPR95↓ | AUPR↑ |
| SINGLE | 72.44 | 74.76 | 71.76 | 83.49 | 51.40 | 38.82 | 77.96 | 63.08 | 55.29 |
| DEEP ENSEMBLES | 72.74 | 74.86 | 72.02 | 84.80 | 48.54 | 40.28 | 78.77 | 61.70 | 56.15 |
| PACKED ENSEMBLE | 70.50 | 77.98 | 69.72 | 82.11 | 53.21 | 35.49 | 76.30 | 65.59 | 52.60 |
| MIMO | 73.04 | 73.05 | 72.32 | 83.08 | 52.65 | 39.02 | 78.06 | 62.85 | 55.67 |
| OBA | 72.74 | 74.10 | 71.23 | 80.88 | 60.76 | 34.16 | 76.81 | 67.43 | 52.69 |
| LORA-SINGLE | 72.54 | 74.66 | 71.81 | 83.31 | 51.57 | 38.41 | 77.92 | 63.11 | 55.11 |
| LORA-ENSEMBLE | 72.56 | 74.60 | 71.80 | 83.31 | 51.80 | 38.37 | 77.93 | 63.20 | 55.08 |
| BATCH ENSEMBLE | 72.58 | 74.66 | 71.83 | 83.42 | 51.30 | 38.57 | 78.00 | 62.98 | 55.20 |
| MC DROPOUT | 71.29 | 76.21 | 70.46 | 82.98 | 52.72 | 37.04 | 77.13 | 64.46 | 53.75 |
| TAYLOR | 74.95 | 75.58 | 74.86 | 84.72 | 54.63 | 43.59 | 79.83 | 65.10 | 59.22 |
| CIRCAVG | 76.05 | 73.46 | 76.47 | 85.95 | 50.54 | 46.20 | 81.00 | 62.00 | 61.33 |
| Hydra Ensembles (Taylor) | 75.39 | 74.02 | 75.22 | 85.68 | 50.60 | 44.49 | 80.53 | 62.31 | 59.85 |
| Hydra Ensembles (Circ) | **76.25** | **72.94** | **76.57** | **86.53** | **47.26** | **46.33** | **81.39** | **60.10** | **61.45** |

Table 22: ImageNet-1K (ViT-B/16): OOD detection (Far-OOD splits: iNaturalist, Texture) with MSP.

| Method | iNaturalist | | | Texture | | | Far-OOD Avg | | |
|---|---|---|---|---|---|---|---|---|---|
| | AUROC↑ | FPR95↓ | AUPR↑ | AUROC↑ | FPR95↓ | AUPR↑ | AUROC↑ | FPR95↓ | AUPR↑ |
| SINGLE | 95.10 | 24.33 | 83.76 | 86.58 | 50.51 | 49.30 | 90.84 | 37.42 | 66.53 |
| DEEP ENSEMBLES | **96.02** | **19.28** | **86.18** | **88.39** | **45.06** | **52.58** | **92.20** | **32.17** | **69.38** |
| PACKED ENSEMBLE | 94.15 | 26.45 | 80.35 | 86.28 | 48.96 | 47.13 | 90.21 | 37.70 | 63.74 |
| MIMO | 93.02 | 31.46 | 77.27 | 85.41 | 53.42 | 47.95 | 89.21 | 42.44 | 62.61 |
| OBA | 90.51 | 36.26 | 68.00 | 86.34 | 48.49 | 46.18 | 88.42 | 42.37 | 57.09 |
| LORA-SINGLE | 94.86 | 25.40 | 83.07 | 86.27 | 50.62 | 48.22 | 90.56 | 38.01 | 65.64 |
| LORA-ENSEMBLE | 94.82 | 25.52 | 82.92 | 86.27 | 50.47 | 48.31 | 90.54 | 37.99 | 65.61 |
| BATCH ENSEMBLE | 94.99 | 25.00 | 83.55 | 86.40 | 50.59 | 48.90 | 90.69 | 37.79 | 66.22 |
| MC DROPOUT | 94.60 | 25.46 | 81.76 | 85.94 | 51.38 | 46.37 | 90.27 | 38.42 | 64.06 |
| TAYLOR | 92.10 | 33.61 | 74.58 | 85.77 | 54.23 | 44.84 | 88.93 | 43.92 | 59.71 |
| CIRCAVG | 94.33 | 26.13 | 81.04 | 86.53 | 50.80 | 45.94 | 90.43 | 38.46 | 63.49 |
| Hydra Ensembles (Taylor) | 93.09 | 29.80 | 76.41 | 87.28 | 47.60 | 46.88 | 90.18 | 38.70 | 61.64 |
| Hydra Ensembles (Circ) | 94.77 | 23.38 | 81.87 | 87.62 | 46.90 | 47.84 | 91.19 | 35.14 | 64.85 |

ods, offering significant improvements in FPR95 (-3.04%) and only mild reductions in AUROC and AUPR relative to the single model.

Finally, although CircuitAvg attains the highest OOD scores, it suffers from poor ID classification, making it inferior overall.

### E.3.2 EFFECT OF PRUNING ONLY THE VISION ENCODER AND FINE-TUNING

In this section, we present results from two additional settings.

First, we prune only the vision encoder to align with the experiments in Section 5.1, considering a scenario focused on computational efficiency. Second, we evaluate the effect of fine-tuning in this setup.

For these experiments, circuits are extracted using CIFAR-100 with its corresponding OOD set, together with ImageNet-1k, due to observed performance degradation. Consequently, results from vision-encoder-only pruning are not directly comparable to those obtained when pruning both encoders, and we leave further investigation for future work.

Given the high cost of training CLIP and the strong performance of Hydra Ensembles when pruning both encoders, we do not explore fine-tuning in that setting. Finally, since pruning both encoders removes a different number of parameters than pruning only the vision encoder, we also report results with an additional six heads pruned to confirm that the gains are not simply due to a smaller increase in parameters.

Table 23: CIFAR100 (ViT-B/16): OOD detection (Far-OOD splits: MNIST, SVHN, Textures and Places365) with MSP.

| | MNIST | | | SVHN | | | Textures | | | Places365 | | | Far-OOD Avg | | |
|---|---|---|---|---|---|---|---|---|---|---|---|---|---|---|---|
| Method | AUROC↑ | FPR95↓ | AUPR↑ | AUROC↑ | FPR95↓ | AUPR↑ | AUROC↑ | FPR95↓ | AUPR↑ | AUROC↑ | FPR95↓ | AUPR↑ | AUROC↑ | FPR95↓ | AUPR↑ |
| SINGLE | 59.49 | 76.63 | 90.69 | 92.61 | 29.52 | 96.97 | 95.79 | 20.51 | 93.86 | 89.78 | 36.77 | 96.54 | 84.41 | 40.85 | 94.51 |
| DEEP ENSEMBLES | 58.42 | 78.82 | 90.69 | **94.16** | **25.22** | **97.67** | 96.53 | 18.99 | 95.06 | 90.43 | 35.59 | 96.83 | 84.88 | 39.65 | 95.06 |
| PACKED ENSEMBLE | 67.45 | 72.08 | 92.66 | 93.32 | 26.49 | 97.3 | 95.54 | 20.69 | 93.29 | 90.59 | 33.33 | 96.8 | 86.72 | 38.14 | 95.01 |
| MIMO | 61.84 | 74.06 | 91.25 | 92.37 | 30 | 96.86 | 95.65 | 21.18 | 93.6 | 89.56 | 37.11 | 96.44 | 84.85 | 40.58 | 94.53 |
| OBA | 63.03 | 73.49 | 91.79 | 92.6 | 29.36 | 96.95 | 95.4 | 21.43 | 93.24 | 89.4 | 37.61 | 96.36 | 85.10 | 40.47 | 94.58 |
| LoRA-SINGLE | 62.78 | 72.4 | 91.42 | 92.35 | 30.47 | 96.86 | 95.63 | 20.9 | 93.58 | 89.53 | 37.21 | 96.43 | 85.07 | 40.24 | 94.57 |
| LoRA-ENSEMBLE | 61.84 | 74.06 | 91.25 | 92.37 | 30 | 96.86 | 95.65 | 21.18 | 93.6 | 89.56 | 37.11 | 96.44 | 84.85 | 40.58 | 94.53 |
| BATCH ENSEMBLE | 64.54 | 71.51 | 92.22 | 92.33 | 29.31 | 96.8 | 95.54 | 21.54 | 93.44 | 89.61 | 36.83 | 96.45 | 85.50 | 39.79 | 94.72 |
| MC DROPOUT | 63.03 | 73.49 | 91.79 | 92.6 | 29.36 | 96.95 | 95.4 | 21.43 | 93.24 | 89.4 | 37.61 | 96.36 | 85.10 | 40.47 | 94.58 |
| TAYLOR | 75.07 | 69.94 | 94.95 | 90.36 | 41.68 | 96.14 | 96.45 | 18.71 | 95.1 | 90.61 | 38.09 | 97 | 88.12 | 42.10 | 95.79 |
| CIRCAVG | **78.81** | **59.68** | **95.59** | 92 | 34.64 | 96.8 | 96.42 | 17.47 | 94.88 | 90.15 | 42.83 | 96.89 | **89.34** | 38.65 | 96.04 |
| Hydra Ensembles (Taylor) | 71.11 | 82.73 | 94.44 | 92.81 | 31.19 | 97.13 | **97.12** | **15.67** | **95.94** | **91.49** | 35.17 | 97.3 | 88.13 | 41.19 | **96.20** |
| Hydra Ensembles (Circ) | 76.01 | 61.56 | 94.87 | 91.99 | 33.78 | 96.75 | 96.68 | 16.42 | 95.28 | 90.89 | 37.54 | 97.13 | 88.89 | **37.32** | 96.00 |

Table 24: OpenCLIP ViT-B/32 classification results per dataset.

| Metrics | Method | Heads | Train | IN-1K | C100 | C10 | Food | SUN | Pet | DTD | EuroSat | Caltech | **Avg** |
|---|---|---|---|---|---|---|---|---|---|---|---|---|---|
| Acc ↑ | SINGLE | 12 | | 66.13 | 75.59 | 93.68 | 82.02 | 68.10 | 87.27 | 50.48 | 40.95 | **98.62** | 73.65 |
| | BAYESVLM | 12 | ✓ | 65.89 | **75.61** | **93.75** | **82.24** | 67.70 | 86.97 | 50.21 | 41.78 | **98.62** | 73.64 |
| | TAYLOR | 10 | | 58.92 | 37.11 | 64.49 | 67.18 | 63.38 | 82.83 | 49.31 | 25.52 | 97.97 | 60.75 |
| | CIRCAVG | 10 | | 64.29 | 71.79 | 91.75 | 79.63 | 65.67 | 86.21 | 50.90 | 38.94 | 98.05 | 71.91 |
| | HYDRA-ENS (CIRC) | 10x3 | | **66.20** | 74.87 | 93.46 | 80.50 | 68.65 | 87.84 | 53.88 | 41.94 | **98.62** | **74.00** |
| Brier ↓ | SINGLE | 12 | | 0.47 | 0.34 | 0.10 | 0.26 | 0.45 | 0.19 | 0.67 | 0.76 | **0.02** | 0.36 |
| | TEMP. SCALING | 12 | | **0.46** | 0.33 | 0.10 | 0.26 | **0.44** | 0.19 | 0.64 | 0.71 | **0.02** | 0.35 |
| | BAYESVLM | 12 | ✓ | 0.47 | 0.34 | **0.09** | 0.25 | 0.45 | 0.19 | 0.66 | 0.73 | **0.02** | 0.35 |
| | TAYLOR | 10 | | 0.57 | 0.82 | 0.47 | 0.46 | 0.51 | 0.25 | 0.70 | 0.96 | 0.04 | 0.53 |
| | CIRCAVG | 10 | | 0.48 | 0.38 | 0.12 | 0.29 | 0.47 | 0.19 | 0.65 | 0.75 | 0.04 | 0.38 |
| | HYDRA-ENS (CIRC) | 10x3 | | **0.46** | 0.35 | 0.10 | 0.28 | **0.44** | 0.18 | 0.64 | 0.71 | 0.05 | 0.36 |
| NLL ↓ | SINGLE | 12 | | 1.42 | **0.65** | 0.21 | 0.65 | 1.14 | 0.56 | 1.99 | 1.90 | **0.03** | 0.98 |
| | TEMP. SCALING | 12 | | **1.31** | 0.85 | **0.20** | 0.63 | **1.07** | 0.52 | **1.79** | 1.66 | **0.03** | 0.90 |
| | BAYESVLM | 12 | ✓ | 1.35 | 0.86 | **0.20** | 0.62 | 1.10 | 0.54 | 1.89 | 1.73 | **0.03** | 0.93 |
| | TAYLOR | 10 | | 1.82 | 3.38 | 1.10 | 1.33 | 1.36 | 0.70 | 2.18 | 2.85 | 0.08 | 1.65 |
| | CIRCAVG | 10 | | 1.44 | 1.08 | 0.27 | 0.72 | 1.20 | **0.41** | 1.91 | 1.81 | 0.09 | 0.99 |
| | HYDRA-ENS (CIRC) | 10x3 | | 1.36 | 0.94 | 0.22 | 0.70 | 1.08 | 0.42 | 1.80 | 1.78 | 0.10 | 0.93 |
| ECE ↓ | SINGLE | 12 | | 10.00 | 7.18 | 1.65 | 4.00 | 9.88 | **1.73** | 18.21 | 23.93 | 1.52 | 8.68 |
| | TEMP. SCALING | 12 | | **1.65** | 0.85 | 1.04 | 3.12 | **1.07** | 3.92 | 6.78 | 15.49 | 2.38 | 4.03 |
| | BAYESVLM | 12 | ✓ | 5.43 | 4.05 | 0.85 | 0.84 | 5.87 | 1.79 | 13.88 | 18.98 | 1.83 | 5.95 |
| | ViLU | 12 | ✓ | 52.20 | 3.40 | **0.50** | 1.00 | 6.60 | 6.60 | **4.00** | 10.00 | **1.20** | 9.50 |
| | TAYLOR | 10 | | 13.58 | 23.95 | 9.79 | 11.03 | 12.03 | 3.06 | 18.39 | 30.28 | 1.74 | 13.76 |
| | CIRCAVG | 10 | | 4.10 | 1.53 | 0.75 | 2.19 | 4.56 | 2.93 | 7.12 | 9.53 | 3.29 | 4.00 |
| | HYDRA-ENS (CIRC) | 10x3 | | 1.82 | 2.34 | 1.31 | 1.68 | 1.89 | 2.41 | 6.58 | **7.73** | 5.65 | **3.49** |
| aECE ↓ | SINGLE | 12 | | 10.00 | 7.10 | 1.53 | 3.99 | 9.87 | 1.79 | 18.21 | 23.93 | 0.66 | 8.57 |
| | TEMP. SCALING | 12 | | 1.60 | **0.76** | 1.04 | 3.12 | **1.19** | 3.87 | 6.74 | 15.25 | **0.65** | 3.80 |
| | BAYESVLM | 12 | ✓ | 5.42 | 3.88 | 0.78 | **0.72** | 5.86 | **1.01** | 13.88 | 18.97 | **0.65** | 5.68 |
| | ViLU | 12 | ✓ | 49.40 | 2.60 | **0.20** | 0.90 | 6.50 | 5.50 | 3.90 | 9.30 | 1.40 | 8.86 |
| | TAYLOR | 10 | | 13.58 | 23.95 | 9.80 | 11.03 | 12.03 | 2.98 | 18.40 | 30.28 | 1.00 | 13.67 |
| | CIRCAVG | 10 | | 4.13 | 1.60 | 0.88 | 2.17 | 4.55 | 2.89 | 7.40 | 9.51 | 3.88 | 4.11 |
| | HYDRA-ENS (CIRC) | 10x3 | | **1.76** | 2.44 | 1.59 | 1.63 | **1.60** | 2.31 | 6.23 | 7.74 | 4.86 | **3.35** |

**Classification.** We report the most important metrics in Table 27. All approaches reduce NLL and ECE compared to the single model (see Table 24). For Hydra Ensembles pruned only on the vision encoder, remaining competitive with other baselines requires additional training and temperature scaling. Accuracy, nevertheless, remains slightly higher than BayesVLM. When pruning both en-

Table 25: OpenCLIP ViT-B/32 OOD detection with MSP on IN-1K.

| Dataset | Metric | SINGLE | BAYESVLM | VILU | TAYLOR | CIRCAVG | HYDRA-ENS (CIRC) |
|---|---|---|---|---|---|---|---|
| **Near-OOD** | | | | | | | |
| SSB | AUROC ↑ | 61.36 | 61.35 | **66.90** | 60.72 | 66.56 | 66.12 |
| | FPR95 ↓ | 86.14 | 86.60 | 83.92 | 86.47 | 82.69 | **81.90** |
| | AUPR ↑ | 60.64 | 60.62 | 65.69 | 60.28 | **66.26** | 65.57 |
| Ninco | AUROC ↑ | 71.39 | 72.52 | 73.67 | 67.69 | 75.41 | **75.68** |
| | FPR95 ↓ | 74.10 | 72.59 | 74.90 | 77.48 | 69.57 | **69.18** |
| | AUPR ↑ | 22.94 | 23.68 | 25.11 | 19.87 | 27.65 | **28.23** |
| Avg | AUROC ↑ | 66.37 | 66.93 | 70.29 | 64.21 | **70.99** | 70.90 |
| | FPR95 ↓ | 80.12 | 79.59 | 79.41 | 81.97 | 76.13 | **75.54** |
| | AUPR ↑ | 41.79 | 42.15 | 45.40 | 40.07 | **46.95** | 46.90 |
| **Far-OOD** | | | | | | | |
| iNaturalist | AUROC ↑ | 76.42 | 78.98 | 83.10 | 70.56 | **89.14** | 88.82 |
| | FPR95 ↓ | 65.26 | 63.65 | 60.99 | 77.02 | 45.59 | **43.66** |
| | AUPR ↑ | 40.58 | 44.36 | 54.90 | 33.38 | **66.01** | 64.92 |
| Texture | AUROC ↑ | 73.88 | 74.51 | 77.87 | 70.80 | 76.42 | **76.69** |
| | FPR95 ↓ | 75.35 | 75.80 | 66.55 | 76.68 | **75.20** | 77.49 |
| | AUPR ↑ | 26.79 | 27.69 | 29.58 | 21.06 | 30.66 | **32.71** |
| Avg | AUROC ↑ | 75.15 | 76.75 | 80.48 | 70.68 | **82.78** | 82.75 |
| | FPR95 ↓ | 70.30 | 69.72 | 63.77 | 76.85 | **60.39** | 60.57 |
| | AUPR ↑ | 33.69 | 36.03 | 42.24 | 27.22 | 48.34 | **48.81** |
| **OOD** | | | | | | | |
| Avg | AUROC ↑ | 70.76 | 71.84 | 75.38 | 67.44 | **76.88** | 76.82 |
| | FPR95 ↓ | 75.20 | 74.65 | 71.59 | 79.41 | 68.26 | **68.05** |
| | AUPR ↑ | 37.73 | 39.08 | 43.81 | 33.64 | 47.64 | **47.85** |

coders, removing six extra heads leads to only minor drops—and occasional improvements—across metrics, resulting in overall comparable performance.

**OOD Detection.** Results are shown in Table 28. Pruning both encoders in Hydra Ensembles generally yields better OOD metrics on ImageNet-1k and CIFAR-100, with the exception of CIFAR-100 Far-OOD datasets. Removing six additional heads leads to almost identical performance, except for a drop in Near-OOD AUPR. On ImageNet-1k, fine-tuning the pruned model further improves AUROC and AUPR over the dual-encoder pruning setting.

Table 26: OpenCLIP ViT-B/32 OOD detection with MSP on CIFAR-100.

| Dataset | Metric | SINGLE | BAYES | VLM | ViLU | TAYLOR | CIRCAVG | HYDRA-ENS (CIRC) |
|---|---|---|---|---|---|---|---|---|
| **Near-OOD** | | | | | | | | |
| CIFAR-10 | AUROC ↑ | 78.06 | 76.54 | **79.30** | 76.17 | 59.14 | 78.07 | _79.24_ |
| | FPR95 ↓ | 62.80 | 64.74 | 61.00 | 70.49 | 92.39 | **52.77** | _53.48_ |
| | AUPR ↑ | _76.40_ | 74.74 | **77.54** | 74.66 | 60.57 | 72.78 | 74.92 |
| TinyImageNet | AUROC ↑ | 78.10 | 76.39 | 78.80 | 76.82 | 59.38 | _79.61_ | **80.32** |
| | FPR95 ↓ | 58.38 | 61.34 | _57.57_ | 65.37 | 81.42 | 58.57 | **57.37** |
| | AUPR ↑ | 67.01 | 64.23 | 67.75 | 65.24 | 47.18 | _68.05_ | **68.32** |
| Avg | AUROC ↑ | 78.08 | 76.47 | _79.05_ | 76.50 | 59.26 | 78.84 | **79.78** |
| | FPR95 ↓ | 60.59 | 63.04 | 59.28 | 67.93 | 86.91 | _55.67_ | **55.42** |
| | AUPR ↑ | _71.70_ | 69.48 | **72.65** | 69.95 | 53.87 | 70.41 | 71.62 |
| **Far-OOD** | | | | | | | | |
| Texture | AUROC ↑ | **89.71** | 74.66 | 77.19 | 73.67 | 51.79 | _81.85_ | 75.84 |
| | FPR95 ↓ | **36.16** | 65.27 | 63.60 | 68.66 | 86.18 | _54.11_ | 55.97 |
| | AUPR ↑ | **97.98** | 57.45 | 61.48 | _95.49_ | 37.32 | 65.65 | 56.68 |
| MNIST | AUROC ↑ | **97.03** | 87.55 | 91.46 | _92.23_ | 35.87 | 85.43 | 82.80 |
| | FPR95 ↓ | **14.16** | 40.53 | 32.88 | _25.03_ | 86.68 | 29.92 | 37.34 |
| | AUPR ↑ | **98.84** | 97.29 | _98.36_ | 95.83 | 82.28 | 95.51 | 94.64 |
| SVHN | AUROC ↑ | 76.18 | _96.39_ | 97.47 | 86.55 | 59.07 | **96.43** | 95.63 |
| | FPR95 ↓ | 64.46 | 16.37 | **11.37** | 53.87 | 68.88 | _12.91_ | 15.43 |
| | AUPR ↑ | 60.66 | _98.61_ | **99.01** | 81.45 | 74.70 | _98.45_ | 98.01 |
| Places365 | AUROC ↑ | 73.04 | 70.23 | 73.88 | 62.89 | 45.28 | **78.32** | _76.30_ |
| | FPR95 ↓ | 71.77 | 74.96 | 70.80 | 85.88 | 92.63 | **67.46** | _69.90_ |
| | AUPR ↑ | 89.19 | 87.50 | 89.53 | 83.91 | 75.24 | **91.57** | _90.24_ |
| Avg | AUROC ↑ | _83.99_ | 82.21 | 85.00 | 78.84 | 48.00 | **85.51** | 82.64 |
| | FPR95 ↓ | 46.63 | 49.28 | _44.66_ | 58.36 | 83.59 | **41.10** | _44.66_ |
| | AUPR ↑ | 86.67 | 85.21 | _87.09_ | **89.17** | 67.38 | 83.28 | 84.89 |
| **OOD** | | | | | | | | |
| Avg | AUROC ↑ | 82.01 | 80.29 | _83.01_ | 78.05 | 51.75 | **83.28** | 81.68 |
| | FPR95 ↓ | 51.28 | 53.86 | 49.53 | 61.54 | 84.69 | _45.95_ | **48.24** |
| | AUPR ↑ | 81.67 | 79.96 | **82.77** | _82.76_ | 62.87 | 82.00 | 80.46 |

Table 27: OpenCLIP ViT-B/32 classification results per dataset. Comparison between vision encoder pruned, in both zero-shot and fine-tuning setting, and both encoders pruned using Hydra Ensembles with circuit extraction.

| Metric | Method | Pruning | Train | IN-1K | C100 | C10 | Food | SUN | Pet | DTD | EuroSat | Caltech | Avg |
|---|---|---|---|---|---|---|---|---|---|---|---|---|---|
| Acc ↑ | HYDRA-ENS | Vision+Text | | **66.20** | 74.87 | _93.46_ | **80.50** | **68.65** | **87.84** | **53.88** | 41.94 | **98.62** | _74.00_ |
| | HYDRA-ENS (-6 HEADS) | Vision+Text | | _66.11_ | 74.14 | 93.31 | _80.28_ | _68.57_ | _87.54_ | _52.34_ | 45.35 | **98.62** | **74.03** |
| | HYDRA-ENS | Vision | | 64.48 | _75.70_ | **93.53** | 78.61 | 67.25 | 86.73 | 51.28 | 40.06 | **98.62** | 72.92 |
| | HYDRA-ENS | Vision | ✓ | 64.25 | **75.71** | 93.25 | 79.89 | 66.44 | 87.14 | 50.64 | 45.16 | _98.46_ | 73.44 |
| NLL ↓ | HYDRA-ENS | Vision+Text | | **1.36** | 0.94 | 0.22 | **0.70** | **1.08** | **0.42** | **1.80** | 1.78 | 0.10 | _0.93_ |
| | HYDRA-ENS (-6 HEADS) | Vision+Text | | _1.37_ | 0.96 | 0.22 | 0.72 | _1.09_ | _0.43_ | _1.84_ | _1.74_ | 0.11 | **0.94** |
| | HYDRA-ENS | Vision | | 1.47 | _0.86_ | **0.20** | 0.78 | 1.16 | 0.60 | 2.01 | 1.87 | _0.07_ | 1.00 |
| | HYDRA-ENS | Vision | ✓ | 1.47 | **0.85** | _0.21_ | _0.71_ | 1.17 | 0.59 | 2.01 | **1.53** | **0.05** | 0.95 |
| ECE ↓ | HYDRA-ENS | Vision+Text | | **1.82** | **2.34** | 1.31 | 1.68 | _1.89_ | 2.41 | **6.58** | _7.73_ | 5.65 | _3.49_ |
| | HYDRA-ENS (-6 HEADS) | Vision+Text | | _1.83_ | 2.87 | 1.70 | 2.79 | **1.32** | 2.79 | _6.80_ | **5.57** | 5.30 | **3.44** |
| | HYDRA-ENS | Vision | | 6.98 | 3.32 | **0.70** | _0.93_ | 6.46 | **1.62** | 13.78 | 20.76 | **1.37** | 6.21 |
| | HYDRA-ENS | Vision | ✓ | 6.51 | _2.67_ | _0.95_ | **0.90** | 6.78 | _2.07_ | 14.72 | 11.74 | _1.56_ | 5.32 |

Table 28: OpenCLIP ViT-B/32 OOD detection (Near/Far) using MSP across datasets. Comparisons include: pruned vision encoder (zero-shot and fine-tuned), both encoders pruned, and both encoders pruned with 6 fewer heads.

| Dataset | Method | Pruning | Training | Near-OOD Avg | | | Far-OOD Avg | | | OOD Average | | |
|---|---|---|---|---|---|---|---|---|---|---|---|---|
| | | | | AUROC ↑ | FPR95 ↓ | AUPR ↑ | AUROC ↑ | FPR95 ↓ | AUPR ↑ | AUROC ↑ | FPR95 ↓ | AUPR ↑ |
| IN-1K | HYDRA-ENS | Vision+Text | | 70.90 | 75.54 | **46.90** | **82.75** | 60.57 | **48.81** | 76.82 | 68.05 | **47.85** |
| | HYDRA-ENS (-6 HEADS) | Vision+Text | | **70.93** | **75.31** | 46.86 | 82.71 | 60.68 | 48.52 | 76.82 | **67.99** | 47.69 |
| | HYDRA-ENS | Vision | | 66.61 | 80.52 | 41.90 | 78.43 | 63.68 | 37.49 | 72.51 | 72.10 | 39.69 |
| | HYDRA | Vision | ✓ | 65.83 | 81.10 | 41.37 | 78.31 | 66.59 | 38.88 | 72.07 | 73.84 | 40.12 |
| C100 | HYDRA-ENS | Vision+Text | | **79.78** | **55.42** | 71.62 | 82.64 | **44.66** | 84.89 | 81.68 | **48.24** | 80.46 |
| | HYDRA-ENS (-6 HEADS) | Vision+Text | | 78.79 | 56.67 | 70.59 | 82.56 | 45.09 | 84.52 | 81.30 | 48.95 | 79.88 |
| | HYDRA-ENS | Vision | | 77.00 | 63.15 | 70.91 | 83.58 | 48.83 | **86.92** | 81.38 | 53.60 | **81.58** |
| | HYDRA-ENS | Vision | ✓ | 78.15 | 60.76 | **71.64** | **83.67** | 46.76 | 86.48 | **81.82** | 51.42 | 81.53 |

