# OpenReview forum: "Ensembling Pruned Attention Heads For Uncertainty-Aware Efficient Transformers"
_ICLR.cc/2026/Conference — ICLR 2026 Poster_

### Official Review · Reviewer_UeX6 · 2025-10-31

**Soundness:** 3
**Presentation:** 3
**Contribution:** 4
**Rating:** 8
**Confidence:** 3

**Summary:**

This work introduces a method for uncertainty estimation in transformers.
The key idea is producing an ensemble of models via pruning attention
heads - which leads to reduced computational complexity compared
to a full ensemble. Evaluation on image and text classification
benchmarks indicates close to state-of-the-art performance.

**Strengths:**

*Clarity:*
- The paper is well-written and is easy to follow.

*Significance / soundness:*
- This work addresses an important problem of epistemic uncertainty estimation in a widely used transformer architecture.
- Method itself and technical solutions seems well-motivated and simple to implement, and paper provides sufficient details for reproducing the results.
- The fact that this method works w/o retraining the base model is a key benefit of the proposed method - re-training large models is intractable.
- Authors provide multiple variants of the method suitable for different settings (with and w/o uncertainty val set, fine-tuning vs zero-shot).

*Evaluation:*
- Performance of the method is close to state-of-the-art, both on prediction and OOD tasks, while the cost is significantly lower, especially in the practical bfloat16 setting. This makes the method extremely useful in practice.

**Weaknesses:**

*Novelty:*
- MoE models seems to be a very similar approach to what is being proposed in this work. A naive baseline
could be re-using an existing MoE model for getting uncertainty estimates?

*Evaluation:*
- Authors claim that the results do not require additional re-training, but in practice this seems a bit misleading
because for both variants of the model (Taylor and Circ) either needs access to uncertainty validation set or
actually requires fine-tuning.
- Paper provides reasonable argument for pruning attention heads instead of MLP, but does not provide a
quantitative evaluation.

**Questions:**

- Have you considered re-using existing MoE-s for uncertainty estimation directly?
- Evaluation is conducted on prediction / OOD tasks. Would the method also work out-of-the-box for generation tasks, MoE-style?
- I wonder if authors have any intuition on why LoRA ensembles would work worse than fine-tuned pruning?

---

> ### Author Response · Authors · 2025-11-20
> **Rebuttal answer 1**
>
> We thank the reviewer for the positive assessment of the paper’s clarity, significance, and practical relevance. We address the raised concerns below.
>
> **P1. Novelty — “MoE models seem very similar; could they be used as a baseline?”**
>
> Mixture-of-Experts (MoE) models and Hydra Ensembles are fundamentally different in purpose and behavior.  Hydra aims to create diverse predictive subnetworks to estimate _epistemic uncertainty_ through an ensemble of structurally different models.
>
> In contrast, MoE routing mechanisms are designed to maximize predictive accuracy, not uncertainty quantification:
>
> - Experts are jointly trained and typically become _highly correlated_, limiting epistemic diversity.
>
> - A single expert subset is activated per input, which does not produce an ensemble of predictions.
>
> To the best of our knowledge, MoE architectures do not produce ensembles of predictions, even though they ensemble experts internally. However, we checked the literature on MOE and Uncertainty quantification and found MEGAN \[1]. MEGAN is not incompatible with our work; it trains experts specialized in different uncertainty types. In fact, Hydra could be combined with MoE approaches by ensembling outputs from MEGAN-like experts to further enhance robustness.\
> We will add a discussion clarifying the distinction between Hydra and MoE-based methods, as well as cite recent works such as MEGAN\[1].
>
> \[1] Agbelese, Damola, et al. "MEGAN: Mixture of Experts for Robust Uncertainty Estimation in Endoscopy Videos." _International Workshop on Uncertainty for Safe Utilization of Machine Learning in Medical Imaging_. Cham: Springer Nature Switzerland, 2025.
>
> **P2. “No Retraining” vs. Fine-Tuning Requirements**
>
> We thank the reviewer for the clarification request.\
> &#x20;Our intention was not to imply that _no data_ is needed: Hydra requires a training or validation set for pruning-score estimation or threshold selection.
>
> However:
>
> - Circ-Hydra does _not_ require any fine-tuning, as shown in our CLIP experiments.
>
> - Taylor-Hydra requires light fine-tuning, which we already acknowledge.
>
> We will revise the text to remove any ambiguity: the key point is that Hydra does not require _retraining the base model_. When fine-tuning appears, it is optional and limited.
>
> **P3. Quantitative Justification — Why Prune Heads and Not MLP Neurons?**
>
> We appreciate the reviewer’s question. We chose head pruning over MLP pruning for two main reasons:
>
> 1. Compatibility with MoE architectures.\
>    &#x20;Modern foundation models increasingly use MoE layers primarily in the MLP block.\
>    &#x20;If Hydra pruned MLP neurons, this would interact poorly with MoE routing and potentially require guaranteeing that every expert maintains enough neurons.\
>    &#x20;By pruning only attention heads, Hydra becomes directly compatible with MoE-based transformers.
>
> 2. Circuit rationale and computational feasibility.\
>    &#x20;Circuit-discovery methods are defined at the attention-head level or full layer level. There is currently no practical circuit extraction methodology for individual MLP neurons to the best of our knowledge. Even if feasible, it would be computationally prohibitive.\
>    &#x20;For this reason, Circuit-Hydra cannot meaningfully operate at the MLP level.
>
> As requested, we will include quantitative results showing that MLP pruning (without circuit extraction) yields inferior performances. Here is the table :
>
> | Method         | ACC              | Brier            | NLL             | ECE             | aECE            | AUROC            | FPR95            | AUPR             |
> |----------------|------------------|------------------|------------------|------------------|------------------|-------------------|-------------------|-------------------|
> | single         | *80.67*     | **0.27**      | **0.71**      | **0.01**      | **0.01**      | 84.40             | *50.25*      | *60.91*      |
> | single(mlp)    | 79.92            | 0.29             | 0.88             | 0.05             | 0.05             | 81.90             | 59.04             | 55.50             |
> | single(taylor) | **80.68**     | 0.28             | 0.79             | *0.02*      | *0.02*      | 84.38             | 54.51             | 59.46             |
> | single(circ)   | 80.22            | *0.28*      | *0.77*      | *0.02*      | *0.02*      | **85.71**      | **50.23**      | **62.41**      |

---

> > ### Author Response · Authors · 2025-11-20
> > **Rebuttal answer 2**
> >
> > **P4. Applicability to Generation Tasks**
> >
> > Short answer: yes, Hydra Ensemble can be used in generative settings, but its usefulness depends on the definition of uncertainty in LLMs.
> >
> > In generation tasks, such as in LLMs, uncertainty is typically estimated using the inherent stochasticity of sampling, producing multiple outputs per prompt. Many methods, such that for example the semantic uncertainty estimation \[2,3,4], rely on this property rather than explicit ensembles which are almost impossible.
> >
> >  Hydra Ensemble can in principle replace sampling-based diversity with structural ensemble diversity. However, two challenges remain:
> >
> > - The field still lacks a consensus on _how to evaluate correctness and uncertainty_ in open-ended generation.
> >
> > - It is unclear whether _model-structural diversity_ (Hydra Ensemble) provides more meaningful uncertainty than _sampling diversity_ (LLMs’ natural randomness).
> >
> > Thus, while Hydra applies mechanically to generative transformers, the interpretation of uncertainty remains task-dependent, as is often the case in generation.
> >
> > We will mention this nuance in the final version.
> >
> > \[2] Kuhn, Lorenz, Yarin Gal, and Sebastian Farquhar. "Semantic uncertainty: Linguistic invariances for uncertainty estimation in natural language generation." _arXiv preprint arXiv:2302.09664_ (2023).
> >
> > \[3] Nikitin, Alexander, et al. "Kernel language entropy: Fine-grained uncertainty quantification for llms from semantic similarities." _Advances in Neural Information Processing Systems_ 37 (2024): 8901-8929.
> >
> > \[4] Qiu, Xin, and Risto Miikkulainen. "Semantic density: Uncertainty quantification for large language models through confidence measurement in semantic space." _Advances in neural information processing systems_ 37 (2024): 134507-134533.
> >
> > **P5. Why Do LoRA Ensembles Perform Worse Than Hydra Ensembles?**
> >
> > This is an insightful question. We don’t know exactly why we can see this phenomenon. We try to replicate the performances of \[5]. We might have a small hint that the performances from LORA ensemble \[5] are mainly improved due to the overfitting competencies of a single LORA, combined with an ensemble of 16 models. But we are not sure. In addition, \[6] seems to claim that Full finetuning should be preferred to LORA finetuning for performance. Sorry, we don’t have a final answer to that point.
> >
> > \[5] Mühlematter, Dominik J., et al. "LoRA-Ensemble: Efficient Uncertainty Modelling for Self-attention Networks." arXiv preprint arXiv:2405.14438 (2024).
> >
> > \[6] Shuttleworth, Reece, et al. "Lora vs full fine-tuning: An illusion of equivalence." _arXiv preprint arXiv:2410.21228_ (2024).

---

> ### Author Response · Authors · 2025-11-27
> **Official Comment by Authors**
>
> Dear Reviewer UeX6,
>
> Given that the discussion deadline is approaching, we are eager to know if our responses have addressed your concerns. We look forward to further discuss with you. If you have any remaining questions, please feel free to raise them, we will try our best to address your concerns until the end of the discussion period.
>
> Best regards,
>
> The authors

---

### Official Review · Reviewer_J4z9 · 2025-10-31

**Soundness:** 2
**Presentation:** 3
**Contribution:** 2
**Rating:** 6
**Confidence:** 3

**Summary:**

The paper addresses the challenge of uncertainty quantification (UQ) in large-scale transformer models. While Deep Ensembles are known to provide reliable uncertainty estimates, they are computationally expensive due to multiple independently trained models. The authors aim to retain ensemble-level calibration and robustness while significantly reducing computational and memory costs. Hydra Ensembles constructs an efficient transformer ensemble by pruning attention heads in a single pre-trained transformer to generate diverse subnetworks and merging these pruned models into a single architecture using Grouped Fully Connected (GFC) layers, forming a Fused Multi-Head Attention (MHA) and Merged MLP structure.

**Strengths:**

* The paper is well written and well motivated.
* The method can be applied to different MoE architectures.
* It shows good improvements in OOD performance.

**Weaknesses:**

* The paper emphasizes ensemble diversity but does not quantify the resulting predictive diversity of the pruned members or the fused model against standard baselines. Appendix B.5 analyzes sources of diversity (e.g., seeds, batch order) but stops short of reporting diversity magnitude after pruning/fusion (e.g., disagreement rate), leaving it unclear whether Hydra is more or less diverse than alternatives.
* On image classification, in-distribution calibration (Brier, NLL) is roughly on par with a single model, suggesting the method’s gains are concentrated in OOD detection. So the only benefit of the model is for OOD detection, and it does not affect the method's robustness in IND, which is counterintuitive.

**Questions:**

* If each model differs only in the set of surviving heads (line 240), why do you need to average the weights and biases across the M models for the MLP layer (line 247)? Are these not the same?
* In Hydra Ensembles(circuit), the authors use the Headmap method (Wang et al., 2025) to identify which heads matter most for uncertainty, and remove the rest. What would be the impact of optimizing for a different task? Also, if you're always optimising for which heads matter most for uncertainty and removing the rest, how do you get different results across the M models?
* Given the strong performance of Taylor and CircuitAverage in the benchmark (on classification and zero-shot tasks), is it worth including them in the benchmark of inference time?

---

> ### Author Response · Authors · 2025-11-20
> **Rebuttal answer 1**
>
> We thank the reviewer for their thorough and constructive feedback. Below, we provide detailed responses to each point raised.
>
> **P1. Missing Quantification of Predictive Diversity (Weakness)**
>
>  We fully agree with the reviewer that a quantitative analysis of predictive diversity is essential. In the revised version, we will extend Table 8 and include an additional table in the supplementary material reporting the following diversity metrics for both ID and OOD samples:
>
> - Mutual Information ID (ID\_MI) / OOD(OOD\_MI)
>
> - Disagreement  Diversity ID(ID\_DI) / OOD (OOD\_DI)
>
> These metrics collectively capture epistemic disagreement at both the logits and distributional levels. See the following table to see our results.
>
> | Method           | pruning seed | batch order | backpropagation | ID_MI | ID_DI | OOD_MI | OOD_DI |
> |------------------|--------------|-------------|------------------|-------|-------|--------|--------|
> | single           | -            | -           | -                | -     | -     | -      | -      |
> | single pruned    | -            | -           | -                | -     | -     | -      | -      |
> | deep ensemble    | -            | -           | -                | **0.06**  | **0.13**  | **0.23**   | **0.58**   |
> | Hydra Ens (Circ) | V            | X           | X                | *0.05*  | *0.11*  | *0.12*   | *0.48*   |
> | Hydra Ens (Circ) | X            | V           | X                | 0     | 0.02  | 0      | 0.13   |
> | Hydra Ens (Circ) | X            | X           | V                | 0     | 0     | 0      | 0.02   |
> | Hydra Ens (Circ) | X            | V           | V                | 0     | 0.02  | 0      | 0.13   |
> | Hydra Ens (Circ) | V            | V           | V                | 0.04  | *0.11*  | *0.12*   |* 0.48*   |
> | Hydra Ens (Circ) | X            | X           | X                | 0     | 0     | 0      | 0      |
>
>  This table will be added in the supplementary. As expected, we observe that diversity increases substantially under OOD inputs, while remaining only moderately higher than the baseline on ID inputs when using different seeds of pruning. We will incorporate these results into the revised paper. This confirms the results of Table 8.

---

> > ### Author Response · Authors · 2025-11-20
> > **Rebuttal answer 2**
> >
> > **P2. IND Calibration Close to a Single Model (Weakness)**
> >
> > We appreciate this insightful observation. We agree that the ID gains appear limited; however, this is a common phenomenon: even full Deep Ensembles often yield only modest improvements in ID calibration metrics (Brier, NLL) when the underlying transformer is already well-calibrated.
> >
> > Notably, for CLIP, where the base model exhibits poorer calibration, Hydra Ensemble shows a stronger improvement. To address this point more completely, we have run an additional covariate-shift detection experiment, which is closely linked to the aleatoric.  Please see the following table:
> >
> >
> >
> >
> >
> >
> > | Method              | Acc_sev1           | Acc_sev2           | Acc_sev3           | Acc_sev4           | Acc_sev5           | Average            |
> > |---------------------|---------------------|---------------------|---------------------|---------------------|---------------------|---------------------|
> > | Single              | 70.44               | 63.31               | 56.57               | 45.10               | 31.97               | 53.47               |
> > | Deep Ensembles      | **73.11**        | **66.59**        | **60.43**        | *49.50*        | *35.97*        | **57.12**        |
> > | Packed Ensembles    | 68.11               | 60.18               | 52.87               | 41.21               | 28.81               | 50.23               |
> > | MIMO                | 71.49       | 64.88        | 58.78        | 47.56               | 34.40               | 55.42                 |
> > | Btach Ensemble      | 70.10               | 62.94               | 56.21               | 44.71               | 31.72               | 53.13               |
> > | MC Dropout          | 70.44               | 63.64               | 57.39               | 46.38               | 33.22               | 54.21               |
> > | Lora Ensembles      | 70.27               | 63.09               | 56.34               | 44.88               | 31.79               | 53.27               |
> > | OBA                 | 69.12               | 62.49               | 56.40               | 47.06               | 35.24                | 54.06               |
> > | Taylor              | 70.63               | 63.81               | 57.84               | 47.84                | 34.92               | 55.00               |
> > | CircAvg             | 70.00               | 62.97               | 56.52               | 45.81               | 32.66               | 53.59               |
> > | Hydra Ens (Taylor)  | *71.65*         | *65.10*        | *59.35*        | **49.62**        | **36.79**        | *56.50*        |
> > | Hydra Ens (Circ)    | 70.81               | 63.87               | 57.62               | 46.96               | 33.73               | 54.59               |
> >
> >
> >
> >
> >
> >
> >
> > | Method              | Brier_sev1         | Brier_sev2         | Brier_sev3         | Brier_sev4         | Brier_sev5         | Average            |
> > |---------------------|---------------------|---------------------|---------------------|---------------------|---------------------|---------------------|
> > | Single              | 0.40                | 0.48                | 0.56                | 0.67                | 0.80                | 0.58               |
> > | Deep Ensembles      | **0.38**         | **0.45**         | **0.52**         | **0.64**         | **0.76**         | **0.55**        |
> > | Packed Ensembles    | 0.43                | 0.52                | 0.60                | 0.71                | 0.83                | 0.61               |
> > | MIMO                | *0.39*         | *0.47*         | *0.54*         | *0.65*         |*0.78*                | *0.56*        |
> > | Btach Ensemble      | 0.41                | 0.49                | 0.56                | 0.68                | 0.80                | 0.58               |
> > | MC Dropout          | 0.40                | 0.48                | 0.55                | 0.66                | 0.79                | 0.57               |
> > | Lora Ensembles      | 0.40                | 0.49                | 0.56                | 0.68                | 0.80                | 0.58               |
> > | OBA                 | 0.43                | 0.50                | 0.57                | 0.67                | 0.79                | 0.59               |
> > | Taylor              | 0.42                | 0.49                | 0.56                | 0.66                | 0.79                | 0.58               |
> > | CircAvg             | 0.43                | 0.50                | 0.57                | 0.68                | 0.80                | 0.59               |
> > | Hydra Ens (Taylor)  | 0.41                | 0.48                | 0.55                | **0.64**         | **0.76**         | *0.56*        |
> > | Hydra Ens (Circ)    | 0.42                | 0.50                | 0.57                | 0.67                | 0.79                | 0.59               |
> >
> >
> >
> >
> > The previous table shows that Hydra Ensemble delivers clear gains. The corresponding results will be added to the paper

---

> > > ### Author Response · Authors · 2025-11-20
> > > **Rebuttal answer 3**
> > >
> > > **P4. How Does HeadMap Produce Different Pruned Sets Across Members? (Clarification)**
> > >
> > > We agree that the mechanism deserves a clearer explanation. Let us consider that we aim to prune $k1$​ heads per model. The process is the following. First, the HeadMap algorithm computes a score for every attention head.\
> > > This provides us with a ranking of the heads. Then, instead of removing the same  $k1$ worst ​ heads that would collapse diversity, we construct a larger candidate set of size $k2 > k1$​ containing the lowest-scored heads. For each Hydra Ensemble member, we sample a different subset of $k1$​ heads from this pool, yielding partially overlapping but distinct pruning configurations.
> > >
> > > This strategy ensures controlled diversity: small differences in head configurations turn out to be sufficient to generate meaningful epistemic disagreement. We will add this explanation to the appendix.
> > >
> > >
> > > **P5. Should Taylor and CircuitAverage Be Included in the Inference-Time Benchmark?** We thank the reviewer for this helpful suggestion. We agree that including them provides a more complete comparison. We will update the inference-time benchmark table accordingly. Here is the revised table:
> > >
> > >
> > > | Method          | Float16/ whole test (s) | Float16/ 1 batch (ms) | Float32/ whole test (s) | Float32/ 1 batch (ms) |
> > > |-----------------|-------------------------|------------------------|--------------------------|------------------------|
> > > | Single          | 23.07                   | 6.15                   | 31.27                    | 8.34                   |
> > > | Taylor          | **13.18**            | **3.51**            | **28.02**             | **7.47**            |
> > > | CircAvg         | *13.20*            | *3.52*            | *28.47*             | *7.59*            |
> > > | Deep Ensembles  | 69.06                   | 18.42                  | 94.36                    | 25.16                  |
> > > | Hydra Ensembles | 24.55                   | 6.55                   | 82.39                    | 21.99                  |

---

> > > > ### Comment · Reviewer_J4z9 · 2025-11-26
> > > >
> > > > I thank the authors for their discussion. While some of my concerns regarding the results have been addressed, I believe the authors still need to clarify what they mean by uncertainty, especially since it is one of the main motivations of the paper. As it stands, the paper does not explicitly quantify uncertainty and instead relies on experimental results, which, as others have shown in the rebuttal, primarily demonstrate the performance of the method with respect to aleatoric uncertainty. It should be made clear which type(s) of uncertainty the paper addresses, and the different notions of uncertainty should not be used interchangeably.

---

> > > > > ### Author Response · Authors · 2025-11-27
> > > > > **Response to Reviewer J4z9 (Part 1)**
> > > > >
> > > > > We thank the reviewer for their follow-up comments. We agree that clarifying what we mean by _uncertainty_ is essential. Before addressing this point, we want to confirm that our responses to P1, and P4 were clear. We also realized that we forgot to answer P3 and apologies for that, and we now provide additional clarification for P4 as well as a complete answer for P3.
> > > > >
> > > > > ***
> > > > > ### **P2. Clarifying which uncertainties we address**
> > > > >
> > > > >
> > > > > Our method primarily targets **epistemic uncertainty**, i.e., uncertainty arising from the model parameters and limited knowledge, as well as **aleatoric uncertainty**, i.e., uncertainty inherent to the data itself \[1].
> > > > >
> > > > >
> > > > > Hydra Ensembles generate multiple structurally diverse subnetworks. The variance across their predictions modifies the overall predictive distribution, which can be seen as an approximation of marginalizing over model weights $P(Y \mid \theta, X)$. The resulting aggregated predictive distribution $P(Y∣X)$ is what we use to estimate uncertainty.
> > > > >
> > > > >
> > > > > - **Epistemic uncertainty** is mainly evaluated through _OOD detection_, which is a standard approach in the literature \[2].
> > > > >
> > > > >
> > > > > - **Aleatoric uncertainty** is evaluated following the framework of Ovadia et al. \[3], using **covariate shift robustness tests**.
> > > > >
> > > > >
> > > > > In particular, benchmarks such as **ImageNet-C** introduce structured corruptions (e.g., speckle noise, gaussian noise, salt and pepper noise, blur, weather effects, digital distortions) applied to the ImageNet test set with different levels of severity. On these corrupted inputs, we measure accuracy, Brier Score, and Area Under Risk-Coverage (AURC) which measures how well a model can identify and reject uncertain samples. It summarizes the trade-off between risk and coverage across all confidence thresholds, where lower AURC indicates better robustness. These evaluations naturally emphasize aleatoric uncertainty.
> > > > >
> > > > > | Method              | Acc_sev1                | Acc_sev2                | Acc_sev3                | Acc_sev4                | Acc_sev5                | Average                 |
> > > > > |---------------------|--------------------------|--------------------------|--------------------------|--------------------------|--------------------------|--------------------------|
> > > > > | Single              | 70.44                    | 63.31                    | 56.57                    | 45.10                    | 31.97                    | 53.47                    |
> > > > > | Deep Ensembles  | **73.11**                | **66.59**                | **60.43**                | 49.50                    | 35.97                    | **57.12**                |
> > > > > | Packed Ensembles    | 68.11                    | 60.18                    | 52.87                    | 41.21                    | 28.81                    | 50.23                    |
> > > > > | MIMO                | 71.49     | 64.88      | 58.78     | 47.56                    | 34.40                    | 55.42                    |
> > > > > | Btach Ensemble      | 70.10                    | 62.94                    | 56.21                    | 44.71                    | 31.72                    | 53.13                    |
> > > > > | MC Dropout          | 70.44                    | 63.64                    | 57.39                    | 46.38                    | 33.22                    | 54.21                    |
> > > > > | Lora Ensembles      | 70.27                    | 63.09                    | 56.34                    | 44.88                    | 31.79                    | 53.27                    |
> > > > > | OBA                 | 69.12                    | 62.49                    | 56.40                    | 47.06                    | 35.24                  | 54.06                    |
> > > > > | Taylor              | 70.63                    | 63.81                    | 57.84                    | 47.84                    | 34.92                    | 55.00                    |
> > > > > | CircAvg             | 70.00                    | 62.97                    | 56.52                    | 45.81                    | 32.66                    | 53.59                    |
> > > > > | Hydra Ens (Taylor) | $\underline{\text{71.65}}$ | $\underline{\text{65.10}}$ | $\underline{\text{59.35}}$ | **49.62**                | **36.79**                | $\underline{\text{56.50}}$ |
> > > > > | Hydra Ens (Circ)    | 70.81                    | 63.87                    | 57.62                    | 46.96                    | 33.73                    | 54.59                    |

---

> > > > > > ### Author Response · Authors · 2025-11-27
> > > > > > **Response to Reviewer J4z9 (Part 2)**
> > > > > >
> > > > > > | Method              | Brier_sev1              | Brier_sev2              | Brier_sev3              | Brier_sev4              | Brier_sev5              | Average                |
> > > > > > |---------------------|--------------------------|--------------------------|--------------------------|--------------------------|--------------------------|--------------------------|
> > > > > > | Single              | 0.40                     | 0.48                     | 0.56                     | 0.67                     | 0.80                     | 0.58                    |
> > > > > > | Deep Ensembles  | **0.38**                 | **0.45**                 | **0.52**                 | **0.64**                 | **0.76**                 | **0.55**                |
> > > > > > | Packed Ensembles    | 0.43                     | 0.52                     | 0.60                     | 0.71                     | 0.83                     | 0.61                    |
> > > > > > | MIMO | $\underline{\text{0.39}}$ | $\underline{\text{0.47}}$ | $\underline{\text{0.54}}$ | 0.65                     | 0.78                     | $\underline{\text{0.56}}$ |
> > > > > > | Btach Ensemble      | 0.41                     | 0.49                     | 0.56                     | 0.68                     | 0.80                     | 0.58                    |
> > > > > > | MC Dropout          | 0.40                     | 0.48                     | 0.55                     | 0.66                     | 0.79                     | 0.57                    |
> > > > > > | Lora Ensembles      | 0.40                     | 0.49                     | 0.56                     | 0.68                     | 0.80                     | 0.58                    |
> > > > > > | OBA                 | 0.43                     | 0.50                     | 0.57                     | 0.67                     | 0.79                     | 0.59                    |
> > > > > > | Taylor              | 0.42                     | 0.49                     | 0.56                     | 0.66                     | 0.79                     | 0.58                    |
> > > > > > | CircAvg             | 0.43                     | 0.50                     | 0.57                     | 0.68                     | 0.80                     | 0.59                    |
> > > > > > | Hydra Ens (Taylor)  | 0.41                     | 0.48                     | 0.55                     | **0.64**                 | **0.76**                 | $\underline{\text{0.56}}$ |
> > > > > > | Hydra Ens (Circ)    | 0.42                     | 0.50                     | 0.57                     | 0.67                     | 0.79                     | 0.59                    |
> > > > > >
> > > > > >
> > > > > >
> > > > > >
> > > > > >
> > > > > >
> > > > > >
> > > > > >
> > > > > >
> > > > > >
> > > > > > | Method              | AURC_sev1 | AURC_sev2 | AURC_sev3 | AURC_sev4 | AURC_sev5 | Average |
> > > > > > |---------------------|-----------|-----------|-----------|-----------|-----------|---------|
> > > > > > | Single              | $\underline{\text{0.10}}$ | 0.14      | 0.18      | 0.28      | 0.41      | 0.22    |
> > > > > > | Deep Ensembles | **0.09**  | **0.12**  | **0.16**  | **0.24**  | **0.37**  | **0.19** |
> > > > > > | Packed Ensembles    | 0.11      | 0.16      | 0.21      | 0.31      | 0.45      | 0.24    |
> > > > > > | MIMO            | **0.09**  | $\underline{\text{0.13}}$ | **0.16**  | $\underline{\text{0.25}}$ | $\underline{\text{0.38}}$ | $\underline{\text{0.20}}$ |
> > > > > > | Batch Ensemble      | $\underline{\text{0.10}}$ | 0.14      | 0.19      | 0.28      | 0.41      | 0.22    |
> > > > > > | MC Dropout          | $\underline{\text{0.10}}$ | 0.14      | 0.18      | 0.27      | 0.40      | 0.21    |
> > > > > > | Lora Ensembles      | $\underline{\text{0.10}}$ | 0.14      | 0.19      | 0.28      | 0.41      | 0.22    |
> > > > > > | OBA                 | $\underline{\text{0.10}}$ | 0.14      | 0.18      | 0.26      | **0.37** | 0.21    |
> > > > > > | Taylor              | 0.11      | 0.15      | 0.19      | 0.27      | 0.40      | 0.22    |
> > > > > > | CircAvg             | 0.11      | 0.15      | 0.19      | 0.28      | 0.42      | 0.22    |
> > > > > > | Hydra Ens (Taylor)  | $\underline{\text{0.10}}$ | $\underline{\text{0.13}}$ | $\underline{\text{0.17}}$ | $\underline{\text{0.25}}$ | **0.37** | $\underline{\text{0.20}}$ |
> > > > > > | Hydra Ens (Circ)    | $\underline{\text{0.10}}$ | 0.14      | 0.18      | 0.27      | 0.40      | 0.21    |
> > > > > >
> > > > > >
> > > > > > Our approach shows improvements for both types of uncertainty, which we will clarify and make explicit distinction in the revised version.
> > > > > >
> > > > > >
> > > > > > **References**\
> > > > > > [1] Hora, Stephen C. _Aleatory and epistemic uncertainty in probability elicitation with an example from hazardous waste management._ Reliability Engineering & System Safety, 1996.\
> > > > > > [2] Maddox et al. _A Simple Baseline for Bayesian Uncertainty in Deep Learning._ NeurIPS 2019.\
> > > > > > [3] Ovadia et al. _Can You Trust Your Model's Uncertainty? Evaluating Predictive Uncertainty under Dataset Shift._NeurIPS 2019.

---

> > > > > > > ### Author Response · Authors · 2025-11-27
> > > > > > > **Response to Reviewer J4z9 (Part 3)**
> > > > > > >
> > > > > > > ### **P3. Why Average MLP Weights If the MLP Is Shared Across Members? (Clarification)**
> > > > > > > We thank the reviewer for raising this point. The clarification is as follows:
> > > > > > >
> > > > > > >
> > > > > > > - For fine-tuned Hydra Ensemble variants (e.g., Taylor), each pruned subnetwork is fine-tuned independently, which leads to slight but meaningful parameter drift in the MLP layers. Therefore, averaging the MLP weights is necessary during fusion.
> > > > > > >
> > > > > > >
> > > > > > > - For CLIP, we intentionally do not fine-tune the pruned subnetworks in the main paper (we did it in the sup). This choice is deliberate: fine-tuning CLIP often harms zero-shot generalization, as shown in Tables 18 and 19; even without any fine-tuning, Hydra Ensemble achieves performance close to state-of-the-art. In this case, the MLP parameters remain identical, so the averaging operation has no effect.
> > > > > > >
> > > > > > >
> > > > > > > We again thank the reviewer for emphasizing this conceptual point. We will revise the paper and hope that this clarification will strengthen the paper. We hope that our answers clarify the missing points.

---

### Official Review · Reviewer_Ez3D · 2025-11-07

**Soundness:** 4
**Presentation:** 4
**Contribution:** 4
**Rating:** 10
**Confidence:** 5

**Summary:**

This article deals with uncertainty quantification (UQ) in transformer-based architectures. It observes that UQ is difficult to achieve, with the current state of the art being held by Deep Ensembles (DE), a contribution from 2017. Deep ensemble achieve sometime-excellent results on UQ at the expense of very expensive training and inference, since multiple training and inference with different parameters must be performed every time.

The article proposes to use pruning methods on the attention heads of transformer architectures only. They justify this approach by suggesting that such pruning is easier to control and that a combined network can then be put together, performing efficient prediction and UQ at once. This proposal is thoroughly justified on both theoretical and practical grounds, achieving close to the state of the art result at virtually no cost on BF16 precision due to the specialised hardware involved. When using FP32 precision, the computational gains are almost non-existent with respect to DE, however.

**Strengths:**

The article is relatively easy to read, very well justified with complete and detailed theoretical and practical explanations. The source code is promised to be made available upon acceptance.

**Weaknesses:**

The success of an ensemble heavily relies on the diversity of its component models. If all models make similar mistakes, combining them won't lead to much improvement. Strategies like varying training data, model architectures, or initialization are needed to ensure diversity

Adaptive Attacks: Sophisticated attackers can create adaptive adversarial examples that specifically aim to minimize the uncertainty metrics (like variance or entropy) of the ensemble, attempting to make their attack look like a confident, in-distribution prediction.

Machine-learning models can be fooled by adversarial examples, i.e., carefully-crafted input perturbations that force models to output wrong predictions. While uncertainty quantification has been recently proposed to detect adversarial inputs, under the assumption that such attacks exhibit a higher prediction uncertainty than pristine data, it has been shown that adaptive attacks specifically aimed at reducing also the uncertainty estimate can easily bypass this defense mechanism.

https://arxiv.org/abs/2309.10586

**Questions:**

- What would be the cost and gain of using 5 heads in BP16 instead of just 3?
- Can the source code be made available for review? Many contributors promise to publish code that turn out to be unreadable, uncommented or incomplete.

---

> ### Author Response · Authors · 2025-11-20
> **Rebuttal answer 1**
>
> We thank the reviewer for their thorough and constructive feedback. Below, we provide detailed responses to each point raised.
>
> **P1. Missing Quantification of Predictive Diversity (Weakness)**
>
>  We fully agree with the reviewer that ensembles benefit most when we ensure strong diversity between members although all Hydra members originate from the same pretrained backbone, structured pruning creates architectural heterogeneity that leads to significantly different functional behaviours  we provide a quantitative analysis of predictive diversity in  the revised version, we will extend Table 8 and include an additional table in the supplementary material reporting the following diversity metrics for both ID and OOD samples:
>
> - Mutual Information ID (ID\_MI) / OOD(OOD\_MI)
>
> - Disagreement  Diversity ID(ID\_DI) / OOD (OOD\_DI)
>
> These metrics collectively capture epistemic disagreement at both the logits and distributional levels. See the following table to see our results.
>
> | Method           | pruning seed | batch order | backpropagation | ID_MI | ID_DI | OOD_MI | OOD_DI |
> |------------------|--------------|-------------|------------------|-------|-------|--------|--------|
> | single           | -            | -           | -                | -     | -     | -      | -      |
> | single pruned    | -            | -           | -                | -     | -     | -      | -      |
> | deep ensemble    | -            | -           | -                | **0.06**  | **0.13**  | **0.23**   | **0.58**   |
> | Hydra Ens (Circ) | V            | X           | X                | *0.05*  | *0.11*  | *0.12*   | *0.48*   |
> | Hydra Ens (Circ) | X            | V           | X                | 0     | 0.02  | 0      | 0.13   |
> | Hydra Ens (Circ) | X            | X           | V                | 0     | 0     | 0      | 0.02   |
> | Hydra Ens (Circ) | X            | V           | V                | 0     | 0.02  | 0      | 0.13   |
> | Hydra Ens (Circ) | V            | V           | V                | 0.04  | *0.11*  | *0.12*   |*0.48*   |
> | Hydra Ens (Circ) | X            | X           | X                | 0     | 0     | 0      | 0      |
>
>  This table will be added in the supplementary. As expected, we observe that diversity increases substantially under OOD inputs, while remaining only moderately higher than the baseline on ID inputs when using different seeds of pruning. We will incorporate these results into the revised paper. This confirms the results of Table 8.

---

> ### Author Response · Authors · 2025-11-20
> **Rebuttal answer 2**
>
> **P2. Vulnerability to Adaptive Adversarial Attacks (Weakness)**
>
> We appreciate the reviewer’s insight. We acknowledge that, similar to Deep Ensembles and other ensemble-based UQ frameworks, Hydra Ensemble inherits some limitations regarding adaptive adversarial robustness. We did not include experiments on fully adaptive adversarial attacks in the current submission.
>
> However, we note that Hydra does not increase the attack surface beyond that of Deep Ensembles; in fact, because Hydra generates all member predictions through a single unified architecture, it may reduce certain attack vectors that arise from independently trained models.
>
> To address the reviewer’s concern, we will add results on corruption robustness using ImageNet-C \[1]. These benchmarks are widely adopted proxies for natural adversarial perturbations. Please refer to P2 (Point 2) answer for reviewer J4z9 for performance table on covariate shift.
>
> [1] Hendrycks, Dan, and Thomas G. Dietterich. "Benchmarking neural network robustness to common corruptions and surface variations." _arXiv preprint arXiv:1807.01697_ (2018).
>
>
> **P3. Cost/Benefit of Using 5 Members in BF16 Instead of 3 (Question)**
>
> Thank you for raising this question. In BF16, most of our experiments were conducted using 3-member Hydra Ensembles. To address the reviewer’s request, we ran additional experiments with 5 members on ImageNet.
>
> Increasing from 3 to 5 members leads to approximately a 1.6× increase in pre-fusion fine-tuning cost, it also leads to a bit of increase in inference time and slightly better performance overall across metrics
>
> We provide the results in the table below and will add them to the paper.
>
> | Method                  | Float16/ whole test (s)      | Float16/ 1 batch (ms)      | ACC               | Brier            | NLL               | ECE               | aECE              | AUROC             | FPR95             | AUPR               |
> |-------------------------|----------------------|--------------------|--------------------|-------------------|--------------------|--------------------|--------------------|--------------------|--------------------|---------------------|
> | Single                  | **23.07**                | **6.15**        | 80.67              | 0.27              | 0.71               | **0.01**        | **0.01**        | 84.40              | 50.25              | 60.91              |
> | Deep Ensembles (3M)     | 69.06                | 18.42              | **82.19**       | **0.25**       | **0.65**        | **0.01**        | **0.01**        | 85.48              | **46.93**       | *62.76*       |
> | Hydra Ensembles (3M)    | *24.55*         | *6.55*        | 80.88              | 0.27              | 0.74               | **0.01**        | **0.01**        | *86.29*       | *47.62*       | **63.15**       |
> | Hydra Ensembles (5M)    | 41.98                | 11.19              | *81.20*       | 0.27              | *0.73*        | **0.01**        | **0.01**        | **86.33**       | 47.85              | *63.20*       |
>
>
>
> **P4. Request for Code Availability Before Acceptance (Question)**
>
> We confirm our commitment to releasing clean and fully reproducible code. A polished version of the codebase will be publicly available at the beginning of next week.

---

> ### Author Response · Authors · 2025-11-27
> **Official Comment by Authors**
>
> Dear Reviewer Ez3D,
>
> Given that the discussion deadline is approaching, we are eager to know whether our responses so far have addressed your concerns. We remain fully available to continue the discussion.
>
> In addition, we have anonymized our code and included it as a ZIP file in the supplementary material, along with comments and a README to facilitate testing, we will also publicly release all checkpoints after the review process to ensure full reproducibility of our results.
>
> Best regards,
>
> The authors

---

### Official Review · Reviewer_twYJ · 2025-11-09

**Soundness:** 3
**Presentation:** 3
**Contribution:** 2
**Rating:** 6
**Confidence:** 3

**Summary:**

The paper proposes Hydra Ensembles: make a few differently pruned versions of the same transformer (different attention heads kept), fine-tune them, and then fuse them into a single model using fused MHA + grouped FC so you can get “ensemble-like” predictions at roughly single-model cost. It targets efficient uncertainty estimation for transformers.

**Strengths:**

1. The paper tackles efficient uncertainty estimation for large transformers, a problem people actually have. That makes the work naturally interesting.

2. Pruning-as-diversity is a nice angle. Most pruning papers try to preserve one model; here, pruning is used to induce differences between members. That’s a small but genuine conceptual twist.

3. The method is demonstrated on both vision (ViT) and language (BERT-style) models, plus a zero-shot setting, so it doesn’t look tied to one benchmark. That strengthens the claim of generality.

4. The cost story is attractive. Claiming ~1× inference vs ~3× for Deep Ensembles directly supports the motivation; it’s exactly the comparison readers care about

**Weaknesses:**

1. The whole method relies on “different pruned heads = different members,” but the paper doesn’t show simple diversity metrics (e.g. pairwise disagreement/KL) before and after fusion. For this idea, that’s the key missing evidence.

2. Several main tables give single numbers but no std / CI / ± over seeds, even though the paper is about uncertainty/robustness.

3. On SST2, the ensemble seems cheaper than a real, fully fine-tuned deep ensemble, which makes Hydra look better than it might against a “full” baseline.

4. The attention fusion story is clear; the MLP part is basically “we average/group.” Since members were pruned and fine-tuned separately, that choice could wash out diversity; an ablation or a justification can be beneficial.

5. The small theoretical part about pruning hurting more under noisy/OOD inputs is more motivational than general; assumptions aren’t clearly checked on ViT/BERT. Either support it empirically in the main text or de-emphasize it.

6. Because zero-shot UQ is sensitive to prompts/datasets/temperature, more details should be in the main paper, not just the appendix.

**Questions:**

1. When you fuse the pruned members into one model, do you still get separate member outputs so we can measure disagreement, or is it combined into one prediction? A small clarification (and maybe a number) would help.

2. If different members keep different attention heads, does the fused layer run the union of those heads (which could increase compute), or do you share some heads to stay close to 1×?

3. You average / group MLP weights across members. Did you try an alternative (e.g. a small per-member adapter) and it didn’t help, or was this mainly for simplicity?

4. How sensitive is Hydra to how aggressively you prune? A short plot or table for one dataset would clarify how robust the method is.

5. For the different Hydra members, how do you ensure that the pruned head sets are actually different (i.e., not largely overlapping)? Do you use different random seeds for the pruning score, or do you enforce low overlap between members?

---

> ### Author Response · Authors · 2025-11-20
> **Rebuttal answer 1**
>
> Dear Reviewer,
>
> First of all, we sincerely thank you for the time and effort you put into reviewing our work. We fully acknowledge the importance of your comments. We also believe that several of your questions reveal a misunderstanding about what our method actually does. This is extremely valuable for us, because addressing these misunderstandings will allow us to significantly improve the clarity of the paper and help future readers better understand our approach and contributions. We hope that the detailed clarifications below will address your concerns and make the method, its motivation, and its empirical support fully clear.
>
> **P1. Missing Quantification of Predictive Diversity (Weakness)**
>
>  We fully agree with the reviewer that a quantitative analysis of predictive diversity is essential. In the revised version, we will extend Table 8 and include an additional table in the supplementary material reporting the following diversity metrics for both ID and OOD samples:
>
> - Mutual Information ID (ID\_MI) / OOD(OOD\_MI)
>
> - Disagreement  Diversity ID(ID\_DI) / OOD (OOD\_DI)
>
> These metrics collectively capture epistemic disagreement at both the logits and distributional levels. See the following table to see our results.
>
> | Method           | pruning seed | batch order | backpropagation | ID_MI | ID_DI | OOD_MI | OOD_DI |
> |------------------|--------------|-------------|------------------|-------|-------|--------|--------|
> | single           | -            | -           | -                | -     | -     | -      | -      |
> | single pruned    | -            | -           | -                | -     | -     | -      | -      |
> | deep ensemble    | -            | -           | -                | **0.06**  | **0.13**  | **0.23**   | **0.58**   |
> | Hydra Ens (Circ) | V            | X           | X                | *0.05*  | *0.11*  | *0.12*   | *0.48*   |
> | Hydra Ens (Circ) | X            | V           | X                | 0     | 0.02  | 0      | 0.13   |
> | Hydra Ens (Circ) | X            | X           | V                | 0     | 0     | 0      | 0.02   |
> | Hydra Ens (Circ) | X            | V           | V                | 0     | 0.02  | 0      | 0.13   |
> | Hydra Ens (Circ) | V            | V           | V                | 0.04  | *0.11*  | *0.12*   |*0.48*   |
> | Hydra Ens (Circ) | X            | X           | X                | 0     | 0     | 0      | 0      |
>
> **P2. Missing Standard Deviations / Confidence Intervals (Weakness)**
>
> We completely agree with the reviewer that reporting variability metrics such as standard deviations or confidence intervals is crucial to understanding the robustness of our method. To address this, we ran Hydra Ensemble with _8_ different random seeds on ImageNet, and we obtained a mean of 80.89 with a standard deviation of _4e-4_ on Accuracy and a mean of 86.29 with standard deviation _1e-4_ on ood AUROC. We will therefore revise all relevant tables to explicitly report results in the form of mean ± standard deviation for Hydra Ensemble.
>
> However, we also want to emphasize a practical limitation that is not specific to our work but affects the entire field: running multiple full trainings of ViT-ImageNet-21k models for the purpose of reporting Deep Ensemble confidence intervals is computationally infeasible. To the best of our knowledge, our paper is the first to actually train a _full Deep Ensemble_ of Vision Transformers at this scale. Repeating such training several times to compute variability would require enormous computational resources, far beyond what is available in typical academic settings.
>
> **P3. Baseline Comparison on SST-2 Might Be Unfair (Weakness)**
>
> We understand the reviewer’s concern about fairness in comparing to Deep Ensembles on SST-2. Indeed, fully retraining each member of a Deep Ensemble usually yields a stronger baseline. However, we want to highlight the computational implications of such a choice. Training a Deep Ensemble on SST-2 with BERT-base (110M parameters) requires three complete pre-trainings + fine-tunings, which is extremely costly given the dataset size and the model scale. That is why we choose to just perform M training on the SST-2 training dataset.
>
> To remain transparent, we want to insist that the differences in performance on ImageNet, on the full pretraining Deep Ensemble, and the trained Deep Ensemble (denoted fn in the table) on ImageNet 1k were not that big in the accuracy but were important on the UQ metrics. We will discuss that point in the supplementary. Please see the following table :
>
> | Method               | ACC   | Brier | NLL  | ECE  | aECE | AUROC | FPR95 | AUPR  |
> |----------------------|-------|-------|------|------|------|-------|-------|-------|
> | Deep Ensembles       | 82.19 | 0.25  | 0.65 | 0.01 | 0.01 | 85.48 | 46.93 | 62.76 |
> | Deep Ensembles (fn)  | 81.23 | 0.27  | 0.81 | 0.03 | 0.03 | 84.33 | 51.25 | 57.13 |

---

> > ### Author Response · Authors · 2025-11-20
> > **Rebuttal answer 2**
> >
> > **P4. Fusion of MLP Layers May Wash Out Diversity (Weakness)**
> >
> > We politely disagree that fusing the MLP layers could raise concerns about reducing model diversity. As we explain in P7 and P8, the model handles diverse predictions of the different members of the ensemble, even though we perform a fusion of the MLP.   The following table shows the results before and after fusion of   the different members:
> >
> >
> > | Method           | ACC              | Brier           | NLL             | ECE             | aECE            | AUROC            | FPR95            | AUPR             |
> > |------------------|------------------|------------------|------------------|------------------|------------------|-------------------|-------------------|-------------------|
> > | single           | 80.67            | **0.27**      | **0.71**      | **0.01**      | **0.01**      | 84.40             | 50.25             | 60.91             |
> > | hydra(fused)     | *80.88*     | **0.27**      | 0.74             | **0.01**      | **0.01**      | **86.29**      | **47.62**      | *63.15*      |
> > | hydra(non fused) | **81.00**     | **0.27**      | 0.74             | **0.01**      | **0.01**      | *86.26*      | *47.98*      | **63.26**      |
> >
> > This table shows that our fusion of the MLP does not degrade the performance.
> >
> > To avoid any misunderstanding, we will expand the discussion in the main paper to detail exactly why MLP fusion does not degrade ensemble diversity.
> >
> > **P5. Theoretical Argument About Pruning Under OOD Inputs Is Not Empirically Supported (Weakness)**
> >
> > We respectfully disagree with this point. The reviewer suggests that our theoretical discussion is not backed by empirical observations, but in fact, Section B.2 contains direct empirical evidence supporting our argument.
> >
> > These results show that when the model is _not_ fine-tuned after pruning, the performance under OOD shifts decreases dramatically:
> >
> > - A \~10% drop in AUROC on ImageNet (near-OOD)
> >
> > - A \~12% drop in AUROC on SST-2
> >
> > This is a very significant degradation. These empirical findings do not merely provide intuition; they demonstrate that improper pruning can severely harm uncertainty estimation, to the point of being catastrophic.
> >
> > Thus, our theoretical argument is not speculative. It is directly supported by the experimental results included in the supplementary material.
> >
> > **P6. Lack of Zero-Shot UQ Details in Main Paper (Question)**
> >
> > Thank you for this remark. Now that the paper is allowed to extend to 10 pages, we will add more explicit details about the prompts used for zero-shot uncertainty estimation. This will make the zero-shot setting more transparent and easier for readers to reproduce.
> >
> >
> > **P7. Fused Model Output: Do We Still Get Per-Member Predictions? (Clarification)**
> >
> > Yes, Hydra Ensemble fully preserves per-member predictions, even though it is implemented inside a single fused transformer.
> >
> > Lines 275–282 describe the mechanism, but we agree that the idea can be hard. Here is a clearer description:
> >
> > - Before the MHA, we arrange the _M_ ensemble predictions along the embedding dimension. This allows the fused MHA to operate efficiently over the enlarged embedding space while preserving member-specific structure. Since we don’t want to have too many tokens for the MHA that will extend too much the attention matrices.&#x20;
> >
> > - After MHA, we reshape the tensor so that the _M_ predictions appear along the token dimension, which allows the merged MLP to process them efficiently.
> >
> > In this way, the fused architecture preserves the diversity of predictions that would normally arise from _M separate networks_. The fusion is purely an architectural reorganization that keeps the computation low without sacrificing member identity.
> >
> > Please let us know if the explanation is not clear.

---

> > > ### Author Response · Authors · 2025-11-20
> > > **Rebuttal answer 3**
> > >
> > > **P8. Does Fused MHA Contain the Union of Heads? Compute Concerns (Clarification)**
> > >
> > > Yes, the fused MHA includes the union of heads selected across all ensemble members. However, this does _not_ imply a significant compute increase, for two reasons:
> > >
> > > 1. We prune aggressively, so the total number of remaining heads remains very small.
> > >
> > > 2. Attention heads have far fewer parameters than MLP layers, meaning that merging many heads still leads to a model much smaller than merging entire transformer blocks.
> > >
> > > Tables 1–4 clearly show the exact number of heads for each dataset:
> > >
> > > - 24 heads on ImageNet
> > >
> > > - 30 heads for CIFAR and CLIP
> > >
> > > - 18 heads for SST-2
> > >
> > > Moreover, Figures 5 and 6 illustrate the effect of different pruning levels.\
> > > &#x20;Section B.4 explains in detail why the total parameter count increases only slightly even when we merge multiple members.
> > >
> > > We will expand the explanation in the paper to avoid any confusion regarding the computational implications of head fusion.
> > >
> > > Please let us know if the explanation is not clear.
> > >
> > > **P9. Why Use MLP Weight Averaging / Did You Test Alternatives? (Clarification)**
> > >
> > > Weight averaging has been shown in prior work to approximate the behavior of an ensemble when all models originate from the same initialization \[1]. This provides a theoretical foundation that justifies our decision to average the MLP weights: since all our subnetworks begin from the same pretrained model and differ only by the heads that are pruned, weight averaging offers a principled way to combine them while preserving some ensemble-like properties.
> > >
> > > In addition to this theoretical grounding, we emphasize that efficiency is a central objective of Hydra Ensembles. Introducing extra layers, such as adapters, per-member MLPs, or other parameter-heavy modules, would substantially increase both the computational cost and the number of parameters. This would go against the core goal of the method, which is to provide _ensemble-quality uncertainty_ at _nearly single-model cost_.
> > >
> > > If efficiency were not a priority, then indeed the simplest alternative would be to keep three separate MLP layers, one for each ensemble member. However, this would negate the computational benefits of Hydra Ensemble and would essentially reproduce the behavior already discussed in Point P4.
> > >
> > > \[1] Izmailov, Pavel, et al. "Averaging weights leads to wider optima and better generalization." _arXiv preprint arXiv:1803.05407_ (2018).
> > >
> > > **P10. Sensitivity to Pruning Aggressiveness (Clarification)**
> > >
> > > The plots illustrating the sensitivity of performance to pruning aggressiveness are already included in Appendix B.3. If the reviewer feels that this is an important aspect of the method, we can add one of these plots directly to the main paper to make the point more prominent.
> > >
> > > **P11. Ensuring Members Have Different Head Sets (Clarification)**
> > >
> > > Section D.1.2 thoroughly explains how we ensure that each ensemble member receives a different set of pruned attention heads. This mechanism ensures diversity across members. We also address this topic in the response to Reviewer J4z9 (point P4).

---

> ### Author Response · Authors · 2025-11-27
> **Official Comment by Authors**
>
> Dear Reviewer twYj,
>
> Given that the discussion deadline is approaching, we are eager to know if our responses have addressed your concerns. We look forward to further discuss with you. If you have any remaining questions, please feel free to raise them, we will try our best to address your concerns until the end of the discussion period.
>
> Best regards,
>
> The authors

---

### Official Review · Reviewer_e9rh · 2025-11-14

**Soundness:** 3
**Presentation:** 3
**Contribution:** 3
**Rating:** 4
**Confidence:** 4

**Summary:**

This paper introduces Hydra Ensembles, a novel framework designed to achieve uncertainty-aware efficient ensembling for large-scale transformer models such as ViT, BERT, and CLIP. The key motivation is to retain the uncertainty quantification (UQ) robustness of Deep Ensembles while drastically reducing their computational and memory costs.

Hydra Ensembles works by pruning attention heads from a single pre-trained transformer to create diverse submodels, which are then fused into a single network via a Grouped Fully Connected (GFC) fusion of their Multi-Head Attention (MHA) and MLP layers. Unlike conventional ensemble approaches, Hydra Ensembles:
	•	avoids retraining each model from scratch,
	•	allows near-single-model inference cost, and
	•	preserves ensemble diversity for robust uncertainty estimation.

The paper’s contributions are threefold:
	1.	Theoretical Analysis: Demonstrates that naïve pruning can degrade model calibration under noisy data conditions, supported by a formal proposition showing loss gap widening in pruned models.
	2.	Framework Design: Introduces structured head-level pruning and GFC-based fusion that maintain model diversity while minimizing computation.
	3.	Empirical Evaluation: Provides extensive experiments on image classification (ImageNet-1K, CIFAR-100), text classification (SST-2), and zero-shot image classification (OpenCLIP). Results show that Hydra Ensembles achieve comparable or superior uncertainty metrics (AUROC, AUPR, ECE) to Deep Ensembles while being ~3× faster and requiring significantly fewer parameters.

Overall, Hydra Ensembles present a theoretically grounded and practically efficient solution for scalable uncertainty quantification in transformer architectures — bridging the gap between computational efficiency and epistemic robustness.

**Strengths:**

Originality
	•	The paper presents a novel architectural strategy for ensemble diversity by pruning and recombining transformer attention heads rather than training separate models.
	•	The Grouped Fully Connected (GFC) fusion mechanism introduces an efficient ensembling pipeline that avoids the typical cost explosion of Deep Ensembles.
	•	Although it builds on concepts like BatchEnsemble and pruning-based model compression, Hydra’s hybridization of pruning and ensembling is unique in both motivation and execution.
	•	The formal proposition linking random pruning to calibration degradation is a valuable theoretical insight, providing justification for structured pruning instead of purely empirical reasoning.

Quality
	•	The methodology is sound and reproducible, with strong empirical results across both vision (ViT, ImageNet-1K, CIFAR-100) and text (BERT, SST-2) domains.
	•	Extensive ablations (pruning ratios, ensemble sizes, GFC variants) demonstrate that the performance improvements are not cherry-picked but systematic.
	•	The authors carefully balance theoretical analysis, algorithmic description, and experimental validation — showing maturity in both design and evaluation.
	•	Calibration metrics (ECE, AUROC, AUPR) are properly selected for uncertainty quantification, and their consistent improvement validates the main claim.

Clarity
	•	The paper is well-organized and readable, with clear section flow and minimal redundancy.
	•	Figures such as the Hydra architecture schematic and calibration plots effectively support understanding.
	•	The theoretical analysis is concise and mathematically consistent, though dense in presentation — yet the accompanying intuition keeps it accessible to non-specialists.
	•	Notation is consistent, and references to related work are appropriate and fair.

Significance
	•	Practical significance: Hydra Ensembles deliver ensemble-level uncertainty quality at a fraction of the computational cost, which is crucial for large transformer models where full ensembles are infeasible.
	•	Research significance: The framework provides a blueprint for uncertainty-aware model compression, bridging a long-standing gap between trustworthiness and efficiency.
	•	Community impact: The method is relevant to multiple research threads at ICLR — efficient transformers, uncertainty quantification, and scalable AI reliability.
	•	Its applicability to both vision and language domains broadens its potential adoption and demonstrates methodological generality.

**Weaknesses:**

1. Limited Theoretical Grounding of Calibration Claims
	•	The proposed theoretical proposition—that random pruning leads to calibration degradation—is intuitively plausible but lacks rigorous derivation or empirical verification linking the theory to measured uncertainty metrics (ECE, NLL, Brier score).
	•	The argument relies primarily on Fisher Information heuristics rather than a formal probabilistic treatment of epistemic variance or diversity loss.
	•	This makes the theoretical part informative but incomplete, as it doesn’t generalize to nonlinear pruning effects or the stochastic dynamics of self-attention.

Recommendation:
The authors could expand this section by (a) connecting pruning-induced variance loss to epistemic uncertainty via bias–variance decomposition, or (b) empirically validating the proposition using entropy/Fisher metrics before and after pruning.

2. Moderate Conceptual Novelty
	•	Hydra Ensembles’ innovation lies mainly in the engineering combination of known techniques: pruning for efficiency (Michel et al., 2019; Voita et al., 2019) and efficient ensemble fusion (Wen et al., 2020; Havasi et al., 2021).
	•	While the integration is elegant, it does not introduce a fundamentally new learning principle or uncertainty formulation.
	•	Related works like BatchEnsemble (Wen et al., ICLR 2020), MIMO (Havasi et al., NeurIPS 2021), and LayerDrop (Fan et al., ICLR 2020) already explore efficiency–diversity trade-offs; the paper could better clarify where Hydra theoretically diverges from these beyond implementation detail.

Recommendation:
Reframe Hydra as a structured synthesis approach rather than a conceptual breakthrough, emphasizing its engineering elegance and scalability benefits.

3. Scope of Experiments
	•	All evaluations are confined to classification tasks (CIFAR-100, ImageNet-1K, SST-2). No tests on generation or multimodal transformers (e.g., CLIP zero-shot) beyond classification accuracy and calibration.
	•	Without broader task validation, it’s unclear whether Hydra’s uncertainty improvements generalize to tasks requiring sequence modeling, open-ended text generation, or multi-modal reasoning—key frontiers of transformer research.

Recommendation:
Include or discuss pilot results on transformer-based generative tasks (e.g., GPT-style models) or multimodal settings to support claims of generality.

4. Insufficient Discussion of Trade-offs
	•	The paper highlights efficiency gains (3× faster inference, fewer parameters) but lacks quantitative trade-off analysis between pruning ratio, uncertainty calibration, and ensemble diversity.
	•	The reader is left without a clear sense of how much diversity is sacrificed at higher pruning rates or how inference cost scales with ensemble size.
	•	Additionally, calibration–efficiency curves or Pareto plots would strengthen interpretability.

Recommendation:
Provide explicit trade-off visualizations (e.g., efficiency vs. ECE or AUROC) and discuss how practitioners can tune pruning levels for optimal performance.

5. Missing Analysis of Diversity and Correlation Among Pruned Submodels
	•	Since ensemble robustness depends on model diversity, the paper should quantify inter-head diversity or correlation (e.g., using cosine similarity, pairwise prediction disagreement, or mutual information across pruned models).
	•	Without such analysis, it’s difficult to confirm whether Hydra truly achieves “diverse ensembling” or merely benefits from redundancy.

Recommendation:
Add an empirical diversity analysis to support the central claim of maintaining epistemic diversity post-pruning.

6. Limited Robustness and OOD Testing
	•	Although Hydra improves calibration, there are no results under domain shift or corrupted data (e.g., ImageNet-C, CIFAR-C, SST-2 perturbations).
	•	This omission weakens claims about robustness and “uncertainty-awareness,” since true epistemic reliability is best evaluated under distributional drift.

Recommendation:
Include robustness experiments on corrupted or OOD benchmarks to demonstrate Hydra’s behavior under uncertainty-inducing conditions.

7. Presentation and Comparison Clarity
	•	Some mathematical notation is dense and occasionally inconsistent across sections (e.g., subscripts for attention heads and fusion layers).
	•	Related work could be contextualized more critically — especially contrasting Hydra’s scalability and calibration against post-hoc methods like temperature scaling, EMM, or confidence regularization.

Recommendation:
Add a summary table contrasting Hydra with BatchEnsemble, MIMO, and FastGeLU Ensembles, highlighting distinct features, efficiency, and calibration properties.

**Questions:**

1. On the Theoretical Proposition and Calibration Justification
	•	The paper presents a theoretical result suggesting that random pruning widens calibration loss gaps.
	•	Could the authors expand or clarify the assumptions underlying this result — e.g., is the bound derived under linearized model assumptions, or does it generalize to nonlinear self-attention layers?
	•	How does this proposition directly link to empirical calibration metrics (ECE, NLL, AUROC)?
	•	Would it be possible to empirically measure information loss or variance reduction (e.g., via Fisher information or entropy) before and after pruning to validate the theory?

2. On Model Diversity and Ensemble Behavior
	•	Hydra’s design implies that pruning attention heads creates diverse submodels whose combination improves uncertainty calibration.
	•	How do the authors quantify or measure diversity among these pruned heads or submodels?
	•	Have they examined pairwise output correlations or disagreement rates across ensemble members?
	•	Without such evidence, how can we be confident that Hydra’s calibration improvements arise from true epistemic diversity rather than simple regularization effects?

3. On Efficiency–Uncertainty Trade-offs
	•	The paper reports efficiency gains (~3× faster inference) while maintaining or improving uncertainty metrics.
	•	Could the authors provide explicit quantitative trade-off curves between pruning ratio, uncertainty calibration, and accuracy?
	•	For example, what is the marginal drop in accuracy or AUROC per additional pruning step?
	•	This would help practitioners decide optimal pruning thresholds under different compute constraints.

4. On Generalization Beyond Classification Tasks
	•	Hydra Ensembles are evaluated on image and text classification tasks, but transformers are widely used for generation and multimodal learning.
	•	Have the authors explored whether Hydra can be applied to sequence generation tasks (e.g., autoregressive decoding, summarization) or vision–language models like CLIP or BLIP?
	•	If not yet tested, do the authors foresee architectural or stability challenges (e.g., head dependencies in causal self-attention) that might limit its application?

5. On the Fusion Mechanism (Grouped Fully Connected Layers)
	•	The Grouped Fully Connected (GFC) fusion layer is a central component, but its mathematical and practical behavior could be clarified.
	•	How does GFC differ from existing ensemble fusion or parameter-sharing mechanisms (e.g., BatchEnsemble, SplitDense)?
	•	Is there a risk of overfitting or co-adaptation when merging diverse heads via GFC?
	•	How is group size or fusion granularity chosen, and how sensitive is Hydra’s performance to these hyperparameters?

6. On the Scope of Uncertainty Metrics
	•	The experiments primarily report ECE and AUROC, which measure calibration and discrimination, respectively.
	•	Have the authors evaluated additional uncertainty metrics such as Brier score, NLL, or predictive entropy?
	•	Different metrics capture complementary aspects of uncertainty; including them could strengthen claims about “uncertainty-awareness.”

7. On Robustness and Distribution Shift
	•	Hydra’s motivation includes improving reliability and robustness through structured diversity.
	•	Have the authors tested Hydra on corrupted or domain-shift datasets (e.g., CIFAR-C, ImageNet-C, SST-2 with noise)?
	•	If not, could they provide a theoretical argument or empirical proxy suggesting Hydra’s robustness benefits beyond clean test distributions?

8. On Implementation Complexity and Reproducibility
	•	Hydra involves structured pruning, submodel fusion, and fine-tuning stages.
	•	Could the authors comment on the implementation complexity and reproducibility — e.g., how many lines of modification are required for ViT or BERT baselines?
	•	Are pretrained Hydra checkpoints or open-source scripts available (or planned) to facilitate community adoption?

9. On Relation to Other Efficient Ensembles
	•	Hydra’s conceptual overlap with BatchEnsemble (Wen et al., 2020) and MIMO (Havasi et al., 2021) is acknowledged but not deeply dissected.
	•	Could the authors articulate the precise difference in diversity mechanism between Hydra and these methods?
	•	Specifically, how does Hydra’s pruning-induced diversity compare empirically to BatchEnsemble’s multiplicative rank-1 reparameterization or MIMO’s input-sharing scheme?

10. On Interpretability of Pruned Attention Heads
	•	Since Hydra modifies attention structure, there may be implications for interpretability (e.g., loss of certain attention patterns or semantics).
	•	Have the authors analyzed whether the pruned heads correspond to interpretable functions (e.g., positional, syntactic, or semantic attention)?
	•	If not, do they expect pruning to impact model explainability — and could this trade-off affect trustworthiness in downstream deployment?

**Details Of Ethics Concerns:**

This paper does not raise any direct ethical concerns in its methodology, data usage, or potential applications.
It focuses exclusively on architectural efficiency and uncertainty quantification in transformer models — a technical advancement with no involvement of human subjects, personal data, or sensitive social content.

All datasets used (e.g., ImageNet-1K, CIFAR-100, SST-2) are public, well-established research benchmarks with existing licensing and ethical clearances.
The work does not propose methods that could be misused for harmful decision-making, data extraction, or disinformation.

The paper poses no ethical or legal risks and aligns with responsible AI research practices.
Its goal — improving uncertainty-aware efficiency in transformer models — contributes positively to trustworthy and sustainable AI.

---

> ### Author Response · Authors · 2025-11-20
> **Rebuttal answer 1**
>
> We thank the reviewer for their reading of our work and for its comments. Below, we provide detailed responses to each point.
>
>
> **W1. Limited Theoretical Grounding of Calibration Claims**
>
>
> We appreciate the reviewer’s concern regarding the theoretical grounding, but we politely disagree on the absence of a link to measured uncertainty metrics. Our proposition is directly tied to the loss, which happened to be the Negative Log-Likelihood (NLL). Therefore, there exists an intrinsic connection between the proposition and calibration.
>
>
> Empirically, the link is as follows: if pruning increases the clean–noisy loss gap, the model’s predicted probabilities become less reliable on corrupted or uncertain samples. This leads to higher NLL, since both overconfident and underconfident predictions are penalized. Although the proposition is expressed in terms of changes in loss, these changes inevitably affect calibration metrics such as ECE, because they depend on the same probability outputs.
>
>
> We acknowledge that the proposition does not explicitly address diversity nor provide a direct theoretical link to ECE. However, NLL itself is a well-established uncertainty quantification metric, and changes in NLL inherently influence calibration behavior.
>
>
> **W2. Moderate Conceptual Novelty / Positioning Among Related Work**
>
>
> We kindly request clarification from the reviewer about the cited works Michel et al. (2019), Voita et al. (2019) without the full references, it is difficult for us to respond precisely. We would be grateful if the reviewer could provide the bibliographic details.
>
>
> Regarding BatchEnsemble \[1] and MIMO \[2], we believe these approaches are not directly comparable to ours. Both require full pretraining or full fine-tuning of large foundation models, which is computationally costly. In contrast, Hydra Ensemble does not rely on such expensive retraining.
>
>
> Additionally:
>
>
> - MIMO performs an implicit ensemble, resulting in a high memory cost,
>
>
> - BatchEnsemble performs an explicit one, but to our knowledge, applying it directly to transformers is non-trivial, and models are often weaker and require full pretraining, making them a comparatively weak baseline in this context.
>
>
> Our method is therefore orthogonal to both.
>
>
> \[1] Wen, Yeming, Dustin Tran, and Jimmy Ba. "Batchensemble: an alternative approach to efficient ensemble and lifelong learning." _arXiv preprint arXiv:2002.06715_ (2020).
>
>
> \[2] Havasi, Marton, et al. "Training independent subnetworks for robust prediction." _arXiv preprint arXiv:2010.06610_ (2020).
>
> **W3. Limited Task Scope: No Multimodal Validation**
>
>
> We acknowledge that there may have been some misunderstanding. The concern raised does not appear to align with the reviewer’s summary (“\[...] Provides extensive experiments \[...] and zero-shot image classification (OpenCLIP).”), which accurately notes that our evaluation of Hydra Ensemble uses zero-shot image classification with OpenCLIP.).  Multimodal validation is already included:
>
>
> - **Table 3** in the main paper provides multimodal experiments.
>
>
> - **Tables 9 and 10** in the supplementary further expand on these results.
>
>
> We will ensure that this is more clearly highlighted in the revised version.
>
>
> **W4. Missing Trade-off Analysis Between Efficiency and Uncertainty Quality**
>
>
> We believe that what is asked is in Section **B.3** of the supplementary material already analyzes the effect of the number of heads on Hydra Ensemble's performance.

---

> > ### Author Response · Authors · 2025-11-20
> > **Rebuttal answer 2**
> >
> > **W5. Missing Diversity/Correlation Analysis Across Pruned Submodels**
> >
> >
> > We agree that diversity analysis is important. While Table 8 in the supplementary already provides relevant diversity results, we will expand this section by reporting the following diversity metrics for both ID and OOD samples:
> >
> >
> > - Mutual Information ID (ID\_MI) / OOD(OOD\_MI)
> >
> >
> > - Disagreement  Diversity ID(ID\_DI) / OOD (OOD\_DI)
> >
> >
> > These metrics collectively capture epistemic disagreement at both the logits and distributional levels. See the following table to see our results.
> >
> >
> > | Method           | pruning seed | batch order | backpropagation | ID_MI | ID_DI | OOD_MI | OOD_DI |
> > |------------------|--------------|-------------|------------------|-------|-------|--------|--------|
> > | single           | -            | -           | -                | -     | -     | -      | -      |
> > | single pruned    | -            | -           | -                | -     | -     | -      | -      |
> > | deep ensemble    | -            | -           | -                | **0.06**  | **0.13**  | **0.23**   | **0.58**   |
> > | Hydra Ens (Circ) | V            | X           | X                | *0.05*  | *0.11*  | *0.12*   | *0.48*   |
> > | Hydra Ens (Circ) | X            | V           | X                | 0     | 0.02  | 0      | 0.13   |
> > | Hydra Ens (Circ) | X            | X           | V                | 0     | 0     | 0      | 0.02   |
> > | Hydra Ens (Circ) | X            | V           | V                | 0     | 0.02  | 0      | 0.13   |
> > | Hydra Ens (Circ) | V            | V           | V                | 0.04  | *0.11*  | *0.12*   |*0.48*   |
> > | Hydra Ens (Circ) | X            | X           | X                | 0     | 0     | 0      | 0      |

---

> > > ### Author Response · Authors · 2025-11-20
> > > **Rebuttal answer 3**
> > >
> > > **W6. Limited Robustness and Covariate-Shift Evaluations**
> > >
> > > We agree with the reviewer on this point and will include robustness results on ImageNet-C\[3]. These benchmarks provide a standard evaluation of natural covariate shifts.
> > >
> > > The corresponding results are shown below and will be added to the revised manuscript.
> > >
> > >
> > > | Method              | Acc_sev1           | Acc_sev2           | Acc_sev3           | Acc_sev4           | Acc_sev5           | Average            |
> > > |---------------------|---------------------|---------------------|---------------------|---------------------|---------------------|---------------------|
> > > | Single              | 70.44               | 63.31               | 56.57               | 45.10               | 31.97               | 53.47               |
> > > | Deep Ensembles      | **73.11**        | **66.59**        | **60.43**        | *49.50*        | *35.97*        | **57.12**        |
> > > | Packed Ensembles    | 68.11               | 60.18               | 52.87               | 41.21               | 28.81               | 50.23               |
> > > | MIMO                | 71.49       | 64.88        | 58.78        | 47.56               | 34.40               | 55.42                 |
> > > | Btach Ensemble      | 70.10               | 62.94               | 56.21               | 44.71               | 31.72               | 53.13               |
> > > | MC Dropout          | 70.44               | 63.64               | 57.39               | 46.38               | 33.22               | 54.21               |
> > > | Lora Ensembles      | 70.27               | 63.09               | 56.34               | 44.88               | 31.79               | 53.27               |
> > > | OBA                 | 69.12               | 62.49               | 56.40               | 47.06               | 35.24                | 54.06               |
> > > | Taylor              | 70.63               | 63.81               | 57.84               | 47.84                | 34.92               | 55.00               |
> > > | CircAvg             | 70.00               | 62.97               | 56.52               | 45.81               | 32.66               | 53.59               |
> > > | Hydra Ens (Taylor)  | *71.65*         | *65.10*        | *59.35*        | **49.62**        | **36.79**        | *56.50*        |
> > > | Hydra Ens (Circ)    | 70.81               | 63.87               | 57.62               | 46.96               | 33.73               | 54.59               |
> > >
> > >
> > >
> > > | Method              | Brier_sev1         | Brier_sev2         | Brier_sev3         | Brier_sev4         | Brier_sev5         | Average            |
> > > |---------------------|---------------------|---------------------|---------------------|---------------------|---------------------|---------------------|
> > > | Single              | 0.40                | 0.48                | 0.56                | 0.67                | 0.80                | 0.58               |
> > > | Deep Ensembles      | **0.38**         | **0.45**         | **0.52**         | **0.64**         | **0.76**         | **0.55**        |
> > > | Packed Ensembles    | 0.43                | 0.52                | 0.60                | 0.71                | 0.83                | 0.61               |
> > > | MIMO                | *0.39*         | *0.47*         | *0.54*         | *0.65*         |*0.78*                | *0.56*        |
> > > | Btach Ensemble      | 0.41                | 0.49                | 0.56                | 0.68                | 0.80                | 0.58               |
> > > | MC Dropout          | 0.40                | 0.48                | 0.55                | 0.66                | 0.79                | 0.57               |
> > > | Lora Ensembles      | 0.40                | 0.49                | 0.56                | 0.68                | 0.80                | 0.58               |
> > > | OBA                 | 0.43                | 0.50                | 0.57                | 0.67                | 0.79                | 0.59               |
> > > | Taylor              | 0.42                | 0.49                | 0.56                | 0.66                | 0.79                | 0.58               |
> > > | CircAvg             | 0.43                | 0.50                | 0.57                | 0.68                | 0.80                | 0.59               |
> > > | Hydra Ens (Taylor)  | 0.41                | 0.48                | 0.55                | **0.64**         | **0.76**         | *0.56*        |
> > > | Hydra Ens (Circ)    | 0.42                | 0.50                | 0.57                | 0.67                | 0.79                | 0.59               |
> > >
> > >
> > >
> > > \[3] Hendrycks, Dan, etal. "Benchmarking neural network robustness to common corruptions and surface variations." _arXiv preprint arXiv:1807.01697_ (2018).
> > >
> > >
> > > **W7. Missing Related Works Presentation and Comparison Clarity**
> > >
> > > To improve clarity, we will include a notation table and refine the related works section. To the best of our knowledge, we did not find some references mentioned by the reviewer.

---

> > > > ### Author Response · Authors · 2025-11-20
> > > > **Rebuttal answer 4**
> > > >
> > > > **Q1 The paper presents a theoretical result suggesting that random pruning widens calibration loss gaps. Could the authors expand or clarify the assumptions underlying this result — e.g., is the bound derived under linearized model assumptions, or does it generalize to nonlinear self-attention layers?**
> > > >
> > > >
> > > > The proposition we present does not rely on random pruning as claimed by the reviewer, but on gradient-based pruning. It doesn't make linearized-network assumption, nor does it require a specific architecture such as fully connected, convolutional, or self-attention layers. Its only structural ingredients are the standard second-order Taylor expansion and the local-minimum condition used in classical pruning analyses (OBD/OBS). These apply to any differentiable neural network, including models with nonlinear activations, convolutional operations, or attention mechanisms.
> > > >
> > > >
> > > > The assumptions in the proposition concern only the local loss landscape at the trained parameters:
> > > >
> > > >
> > > > 1\. The gradients of the training and clean test losses vanish at the solution (local minimum).
> > > >
> > > >
> > > > 2\. The pruning perturbation is sufficiently small for the second-order approximation to remain accurate.
> > > >
> > > >
> > > > 3\. The Hessian difference is positive definite, capturing the intuition that noisy data induce a locally sharper curvature.
> > > >
> > > >
> > > > None of these assumptions imposes linearity. The Hessians matrices are those of the full nonlinear model; the analysis does not linearize the forward map beyond the Taylor expansion.
> > > >
> > > >
> > > > We verify these assumptions empirically on ViT-B/16 (a self-attention architecture) using ImageNet-1K and CIFAR-100 together with their corrupted variants (see Appendix B.1). The gradient–perturbation alignment is positive across all settings. For the curvature condition, we approximate the Hessian using its diagonal and find that the diagonal difference remains strictly positive on all datasets.
> > > >
> > > >
> > > > **Q2 How does this proposition directly link to empirical calibration metrics (ECE, NLL, AUROC)?**
> > > >
> > > >
> > > > Answered in W1.
> > > >
> > > >
> > > > **Q3 Would it be possible to empirically measure information loss or variance reduction (e.g., via Fisher information or entropy) before and after pruning to validate the theory?**
> > > >
> > > >
> > > > We had some metric in supplementary B.1, and we can try to add the NLL.
> > > >
> > > >
> > > > **Q4 How do the authors quantify or measure diversity among these pruned heads or submodels?**
> > > >
> > > >
> > > > Answered in W5.
> > > >
> > > >
> > > > **Q5 Have they examined pairwise output correlations or disagreement rates across ensemble members?**
> > > >
> > > >
> > > > Answered in W5.
> > > >
> > > >
> > > > **Q6 Without such evidence, how can we be confident that Hydra’s calibration improvements arise from true epistemic diversity rather than simple regularization effects?**
> > > >
> > > >
> > > > Answered in W5
> > > >
> > > >
> > > > **Q7 Could the authors provide explicit quantitative trade-off curves between pruning ratio, uncertainty calibration, and accuracy?**
> > > >
> > > >
> > > >  Already in paper, and see W4
> > > >
> > > >
> > > > **Q8 For example, what is the marginal drop in accuracy or AUROC per additional pruning step?**
> > > >
> > > >
> > > > Already in the paper,, everything is in Figure 5.
> > > >
> > > >
> > > > **Q9 Have the authors explored whether Hydra can be applied to sequence generation tasks (e.g., autoregressive decoding, summarization) or vision–language models like CLIP or BLIP?**
> > > >
> > > >
> > > >  Already in paper, and see W3
> > > >
> > > >
> > > > **Q10 If not yet tested, do the authors foresee architectural or stability challenges (e.g., head dependencies in causal self-attention) that might limit its application?**
> > > >
> > > >
> > > >  Already in paper, and see W3
> > > >
> > > >
> > > > **Q11 How does GFC differ from existing ensemble fusion or parameter-sharing mechanisms (e.g., BatchEnsemble, SplitDense)?**
> > > >
> > > >
> > > > Please, what is the reference for SplitDense? For BatchEnsemble see W2.
> > > >
> > > >
> > > > **Q12 Is there a risk of overfitting or co-adaptation when merging diverse heads via GFC?**
> > > >
> > > >
> > > > There is no risk of overfitting or co-adaptation when merging diverse heads via GFC, because the fusion occurs only at inference time, after all subnetworks have been independently pruned and optionally fine-tuned. No joint training of the fused parameters takes place.
> > > >
> > > >
> > > > **Q13  How is group size or fusion granularity chosen, and how sensitive is Hydra’s performance to these hyperparameters?**
> > > >
> > > >
> > > > The group size is determined automatically by the number of heads remaining in each layer after pruning, rather than being set manually.
> > > >
> > > >
> > > > **Q14 The experiments primarily report ECE and AUROC, which measure calibration and discrimination, respectively. Have the authors evaluated additional uncertainty metrics such as Brier score, NLL, or predictive entropy?**
> > > >
> > > >
> > > > Already in the paper, see tables (1,2, and 3) of the experimental section of the main paper.
> > > >
> > > >
> > > > **Q15  Have the authors tested Hydra on corrupted or domain-shift datasets (e.g., CIFAR-C, ImageNet-C, SST-2 with noise)?**
> > > >
> > > >
> > > > Answered in W6

---

> > > > > ### Author Response · Authors · 2025-11-20
> > > > > **Rebuttal answer 5**
> > > > >
> > > > > **Q16  If not, could they provide a theoretical argument or empirical proxy suggesting Hydra’s robustness benefits beyond clean test distributions?**
> > > > >
> > > > >
> > > > > Already in paper, Hydra Ensembles' performance on the OOD task attests to its robustness. We did an additional experiment on covariate shift, too. Please see W6
> > > > >
> > > > >
> > > > > **Q17 Could the authors comment on the implementation complexity and reproducibility — e.g., how many lines of modification are required for ViT or BERT baselines?**
> > > > >
> > > > >
> > > > > Creating one Hydra Ensembles requires the following steps:
> > > > >
> > > > >
> > > > > - Pruning applying the HeadMap algorithm (find more details at appendix D.1.2)
> > > > >
> > > > >
> > > > > - Standard fine-tuning (optional)
> > > > >
> > > > >
> > > > > - MLP merging (i.e. averaging)
> > > > >
> > > > >
> > > > > - Fusion of MHAs through GFC
> > > > >
> > > > >
> > > > > Overall, the implementation is simple and straightforward. Moreover, we plan to share the code by the beginning of next week. I hope this is considered sufficient.
> > > > >
> > > > >
> > > > > **Q18 Are pretrained Hydra checkpoints or open-source scripts available (or planned) to facilitate community adoption?**
> > > > >
> > > > >
> > > > > Yes, we plan to release the checkpoints on Hugging Face upon acceptance of the paper, and we’ll share the code towards the beginning of next week.
> > > > >
> > > > >
> > > > > **Q19  Hydra’s conceptual overlap with BatchEnsemble (Wen et al., 2020) and MIMO (Havasi et al., 2021) is acknowledged but not deeply dissected. Could the authors articulate the precise difference in diversity mechanism between Hydra and these methods?**
> > > > >
> > > > >
> > > > > We would kindly like to clarify that the review’s statement regarding Hydra Ensemble’s “conceptual overlap” with BatchEnsemble and MIMO is not accurate, as nowhere in the paper do we claim such an overlap. Please see our answer on W2, and also please see our answer on Reviewer J4z9 (point P4).
> > > > >
> > > > >
> > > > > **Q20 Specifically, how does Hydra’s pruning-induced diversity compare empirically to BatchEnsemble’s multiplicative rank-1 reparameterization or MIMO’s input-sharing scheme?**
> > > > >
> > > > >
> > > > > Please see our answer on W2.
> > > > >
> > > > >
> > > > > **Q21 Have the authors analyzed whether the pruned heads correspond to interpretable functions (e.g., positional, syntactic, or semantic attention)?**
> > > > >
> > > > >
> > > > > Appendix B.2 study the pruned heads, but if required, we can explain more.
> > > > >
> > > > >
> > > > > **Q22  If not, do they expect pruning to impact model explainability — and could this trade-off affect trustworthiness in downstream deployment?**
> > > > >
> > > > >
> > > > > Linking model explainability directly to its pruned heads is fundamentally a conceptual and methodological challenge. It is not straightforward to claim that a model becomes more interpretable due to the presence or removal of a specific head, nor is it clear why one head should be inherently more interpretable than another. These questions require deeper theoretical grounding and careful experimental design. While they are important and interesting directions, they fall outside the scope of the current work and are better suited for future research.

---

> ### Author Response · Authors · 2025-11-27
> **Official Comment by Authors**
>
> Dear Reviewer e9rh,
>
> Given that the discussion deadline is approaching, we are eager to know if our responses have addressed your concerns. We look forward to further discuss with you. If you have any remaining questions, please feel free to raise them, we will try our best to address your concerns until the end of the discussion period.
>
> Best regards,
>
> The authors

---

### Author Response · Authors · 2025-11-20
**General Answer**

Dear reviewers and AC,

We would like to express our sincere gratitude for the insightful comments and constructive questions of the reviewers. We value the time and effort the reviewer spent reviewing our paper and appreciate your acknowledgment of the substantial work involved in this work.

We believe it is helpful to provide a brief clarification regarding the core contribution of our paper. Our work introduces Hydra Ensembles, a framework designed to solve the efficiency-reliability trade-off in large Transformers. By pruning attention heads to create diverse sub-networks and fusing them via Grouped Fully-Connected layers, we achieve uncertainty quantification performance comparable to Deep Ensembles with an inference cost close to a single model (especially in bfloat16). We tried our best to answer all the clarification points raised by reviewers.

In response to the reviews, we have conducted extensive new experiments to strengthen our empirical validation:

- Diversity Quantification: We now report Disagreement and Mutual Information metrics to explicitly demonstrate the diversity among Hydra members.
- Statistical Robustness: We trained multiple Hydra Ensembles to provide the mean and standard deviations ($\pm$ std) for our main results, ensuring statistical significance.
- Covariate Shift: We added experiments on covariate shift to further test robustness.
- Ablation on Pruning: We performed an MLP pruning experiment to empirically justify our design choice of focusing on attention head pruning.
- Ensemble Size Analysis: We evaluated Hydra Ensembles with up to 5 members to provide a detailed cost-benefit analysis of scaling the ensemble size.
- Comparing full Deep Ensemble and Finetune Deep Ensemble.

We have made every effort to address all reviewer inquiries and have successfully conducted the vast majority of the requested experiments. However, we respectfully note that we were unable to address certain requests regarding specific baselines and prior works, as we could not locate the papers despite extensive searching. **We believe these references do not exist.**
We hope these additions clarify the strengths of our approach and address the concerns raised.
Thank you once again for your constructive feedback and engagement. We look forward to continuing this conversation with reviewers to further refine our work.

Best regards

The authors

---

### Author Response · Authors · 2025-11-26
**Looking Forward to the Reviewers Reply**

Dear Reviewer and AC,

We have uploaded a revised version of the PDF, where all corrections and updates are highlighted in red. We would also like to note that, until now, we have not been able to interact directly with any reviewers. We hope that we have addressed all comments raised by the reviewers.

If any reviewer has specific questions, requests for clarification, or equations they would like us to elaborate on, we would be very happy to provide additional explanations.

Thank you for your time and consideration.

 Best regards,

The authors

---

### Author Response · Authors · 2025-12-01
**Summary of Discussions**

Dear Area Chair and Reviewers,

We would like to thank you again for your time and feedback. Although the discussion phase was relatively short, we used it to run additional experiments and clarify key aspects of the paper :

- **Diversity quantification:** Reported Disagreement and Mutual Information metrics to explicitly demonstrate the diversity among Hydra members, on both ID and OOD data.


- **Statistical robustness:** Trained multiple Hydra Ensembles and now report mean and standard deviation (± std) for our main results.


- **Covariate shift:** Added experiments on covariate shift to further test robustness.


- **Ablation on pruning:** Performed an MLP pruning experiment to empirically justify our design choice of focusing on attention head pruning.


- **Ensemble size analysis:** Evaluated Hydra Ensembles with up to 5 members to provide a cost–benefit analysis of scaling the ensemble size.


- **Deep Ensemble variants:** Compared a full Deep Ensemble with a fine-tuned Deep Ensemble to add clarity to the strengths and weaknesses of the Bert baseline.

Following **reviewer J4z9**'s important comment, we have also explicitly clarified the types of uncertainty we address and added additional results on the corrupted version of ImagNet.

Finally, we note that **reviewer e9rh**'s review contains several apparent **hallucinations from an LLM** (e.g., methods and references that we could not identify). We requested clarification, but none was provided.

We would have liked to be able to discuss with the reviewers and the AC, but this rebuttal has been shortened. We hope that the lack of response from the reviewers is a positive sign that all of the points have been addressed; otherwise, the reviewers would have tried to contact us and discuss with us...


Best regards,

The authors

---

### Meta-Review · Area_Chair_GUGo · 2026-01-06

**Summary:**

Four out of five reviews (which were already inclined toward acceptance prior to rebuttal) agree that the paper presents a strong and practical approach to efficient uncertainty estimation via pruned-and-fused ensembles. The authors argue that the fifth review relies on LLM-generated hallucinations and is therefore partially discounted. The main decision-critical concerns were limited conceptual novelty, lack of explicit predictive diversity evidence, insufficient statistical reporting, baseline clarity, and clearer articulation of where uncertainty gains are achieved. The rebuttal added substantial new experiments and clarifications addressing many of these issues, while some concerns remain partially open.

**Reviewer Concerns:**

Addressed:
* Added explicit diversity metrics and analysis.
* Added mean ± std over multiple runs.
* Added robustness experiments under covariate shift/corruptions.
* Added pruning ablations and MLP.
* Clarified uncertainty framing and baseline comparisons.

Still partially outstanding:
* The level ID calibration gains remain modest relative to OOD improvements.
* Some concerns about some baseline strength (e.g., SST-2 Deep Ensemble) remain.
* Conceptual novelty relative to the state of the art remains a subjective concern.

**Reviewer Scores:**

While it is difficult to precisely estimate how individual reviewers would have updated their scores given the limited discussion, most reviewers were already positive prior to the rebuttal. Given that several of their main concerns were addressed, it is likely that reviewers would have maintained their scores or, at most, increased them slightly. There is no indication that the scores would have been decreased.

---

### Decision · Program_Chairs · 2026-01-26

Accept (Poster)